# A scoping review of measurement of violence against women and disability

**Sarah R. Meyer**[1], **Heidi Stöckl**[2], **Cecilia Vorfeld**[3], **Kaloyan Kamenov**[3], **Claudia García-Moreno**[1]*

**1** Department of Sexual and Reproductive Health and Research, World Health Organization, Geneva, Switzerland, **2** The Institute for Medical Information Processing, Biometry, and Epidemiology, University of Munich, Munich, Germany, **3** Department of Noncommunicable Diseases, World Health Organization, Geneva, Switzerland

* garciamorenoc@who.int

**Data Availability Statement:** All relevant data are within the manuscript and its Supporting Information files.

**Funding:** This work was funded under the UNWomen-WHO Joint Programme on Violence

## Abstract

### Introduction

Existing evidence indicates that prevalence of violence against women with disability is elevated compared to women without disability. We conducted a scoping review with a focus on measurement to assess the forms of measurement and study design utilized to explore the intersection of violence against women with disabilities, and to identify strengths and limitations in current approaches to measuring violence against women with disabilities. This scoping review is designed to inform current debates and discussions regarding how to generate evidence concerning violence against women with disabilities.

### Methods and results

We conducted systematic searches of the following databases: PubMed, PsycINFO, Embase, CINAHL, PILOTS, ERIC, Social Work Abstracts, International Bibliography of the Social Sciences, Social Services Abstracts, ProQuest Criminal Justice, and Dissertations & Theses Global, and conducted structured searches of national statistics and surveys and grey literature available on-line. We identified 174 manuscripts or reports for inclusion. n = 113 manuscripts or reports utilized acts-specific measurement of violence. In terms of measurement of disability, we found that amongst the included manuscripts and reports, n = 75 utilized measures of functioning limitations (n = 20 of these were Washington Group questions), n = 15 utilized a single question approach and n = 67 defined participants in the research as having a disability based on a diagnosis or self-report of a health condition or impairment.

### Discussion

This scoping review provides a comprehensive overview of measurement of violence against women with disabilities and measurement of violence within disability-focused research. We identified several important gaps in the evidence, including lack of sex and disability disaggregation, limited evidence concerning adaptation of data collection methods to ensure accessibility of research activities for women with disabilities, and limited evidence

against Women Data, funded by the Foreign, Commonwealth and Development Office. The funders had no role in study design, data collection and analysis, decision to publish, or preparation of the manuscript.

**Competing interests:** The authors have declared that no competing interests exist

concerning differential relationships between types of disability and violence exposure. This scoping review provides directions for sub-analyses of the included studies and further research to address gaps in evidence.

## Background and introduction

Violence against women is a global public health challenge and violation of human rights. Recent estimates indicate that one in three women aged 15 and older globally have experienced intimate partner violence [IPV] or non-partner sexual violence in their lifetime [1]. Disability has been found to be a risk factor for exposure to violence amongst women and has been found to influence dynamics and patterns of women's exposure to violence [2]. For example, analyses of the association between IPV and disability from seven violence-prevention programs in low and middle-income countries [LMIC] indicates consistent associations between past-year exposure to IPV and disability, with associations stronger with increasing severity of self-reported disability [3]. Literature reviews, systematic reviews and comparative data analyses focusing on violence and disability have indicated that adults with disability are at greater risk for exposure to violence [4–6]. Yet, significant limitations remain in current understanding of the relationship between disability and violence against women, including that some analyses do not adequately account for gender and its shaping of vulnerability to violence.

Risk factors for violence against women with disabilities may be the same as risk factors for women without disabilities [7], yet there also may be specific pathways through which women's vulnerability to violence is heightened due to disability. Proposed factors explaining heightened vulnerability to violence include social exclusion and isolation, reliance of women with disabilities on partners and/ or carers, and the intersection of disability and lack of economic independence, which can compound issues of reliance on abusers [8–10]. Societal views of disability, and social and economic exclusion of persons with disability, can compound severity and duration of violence and restrict women's ability to report or leave abusive situations [11].

A number of conceptual approaches have been proposed to account for the increased vulnerability of women with disabilities to violence. Namatovu et al. suggest two theoretical approaches for understanding violence against women with disabilities. The first, intersectionality, positions both violence and disability as "interdependent and interconnected," and enables understanding of vulnerability to violence through the lens of how the social identities of gender and ability are socially constructed and reinforced. The second, feminist disability theory, proposes that disability is a social construct, and shapes women's experiences and access to full social participation in ways that intersect with patriarchal norms and practices [2]. These approaches theorize the types and patterns of violence against women with disabilities as grounded within the social construct of disability, which results in marginalization and isolation of persons with disability, and patriarchal systems, which uphold power of men over women in family and other central social institutions [12]. This perspective emphasizes that while a specific research methodology may focus on measuring violence and assessing disability at the individual-level, both disability and violence are embedded in social institutions, practices and norms that drive use of violence against women with disabilities. Curry et al. (2001) proposed an ecological model for understanding violence against women with disabilities, identifying environmental and cultural factors that impact prevalence, type and severity of violence against women with disabilities including the intersection of patriarchy and ableist

perspectives resulting in marginalization of women with disabilities, discrimination in health systems, and exclusion from economic opportunities [13]. These conceptual frameworks provide insights into the complex intersections and pathways between disability and violence, yet are limited by their focus on disability in Western and high-income contexts and lack grounding in data based on different forms of disability and a range of contexts and types of violence.

Commentaries and analyses have consistently indicated challenges in estimating prevalence of violence against women with disabilities given lack of comparable data on both disability and violence against women [3]. As outlined in the protocol for this scoping review, multiple measurement issues exist concerning violence against women with disabilities [14]. Challenges specific to measurement of violence include that women with disabilities may experience forms of violence that are not captured in traditional measures, for example, denial of care, physical neglect, and lack of control over medications. Lack of inclusion of these types of violence within standard violence measurement instruments may result in under-estimating violence against women with disabilities. In addition, some evidence indicates that different types of perpetrators may be responsible for violence against women with disabilities [5, 15, 16]. Where specific contexts (i.e. care, institutions) or perpetrators (i.e. carers, assistants) are not included in measurement of violence, violence perpetrated in these contexts and by these perpetrators may be missed [11]. Over-arching principles in measurement of violence against women also apply. Gold standard measurement of violence against women includes asking a series of behavior-specific questions to capture violence, rather than a single question such as "have you ever experienced violence?" Behavior-specific acts questions yield higher levels of disclosure and are less prone to bias based on what different women define as abuse or violence [17].

Beyond challenges with violence measurement, there is considerable debate concerning measurement of disability. Disability is defined as "the interaction between individuals with a health condition. . .with personal and environmental factors including negative attitudes, inaccessible transportation and public buildings, and limited social support," [18]; the Convention on the Rights of Persons with Disabilities also emphasizes social participation, such that "disability results from the interaction between persons with impairments and attitudinal and environmental barriers that hinders their full and effective participation in society on an equal basis with others" [19]. Studies of disability globally employ vastly different outcome measures, definitions of disability or cut-offs to determine disability status across studies, impacting prevalence estimates and comparability of data sources [20]. Some studies apply a single item to identify those with disability, whereas other studies determine disability by asking a set of functional ability questions or use a medical diagnosis as a definition of disability [21]. These distinct approaches lead to major differences in the prevalence estimates of disability impeding comparability. Questions such as "do you have a disability"–likely identify fewer persons with disabilities. For example, in a study in South Africa, there was wide variation in response depending on if a question asked about disability or about difficulty doing things; many respondents indicated difficulty completing a range of daily activities and yet did not define themselves as 'disabled' [22]. Prevalence estimates based on medical diagnosis can also lead to under-reporting as those without access to health services may not have been diagnosed by a professional, and also people with the same health conditions can have different levels of functioning limitations. The Washington Group Questions, which have been widely used in censuses and studies, have also been found to not reliably identify individuals who screen positive clinically for moderate or greater impairment [23]. The use of the Washington Group Questions for screening for disability has been found to define individuals with mild to moderate clinical impairments as non-disabled [24]. A rapid disability assessment in the Philippines found a higher prevalence of disability using a functioning measure assessing eight domains

than a census which had utilized the Washington Group questions [25]. Selection of a particular definition and approach to measurement of disability has a significant impact on conclusions regarding the relationship between disability and violence against women. As such, exploration of how disability is defined and operationalized within this body is literature is essential to furthering evidence in this field.

There is increasing interest in how to develop and implement effective policy and programmatic response for women with disabilities affected by violence [26], yet the current availability of reliable and valid data on disability and violence against women does not adequately match this interest. There is a need for a strengthened evidence-base to inform violence prevention and policy response for women with disabilities subjected to violence and to ensure effective design and implementation of policies, services and programs [27, 28]. Spurred by policy, donor and programmatic interest in addressing violence against women with disabilities, incorporating a measure of disability within a national violence against women survey is increasingly promoted. This may be one effective avenue to increasing availability and improving quality of data on the intersection of violence against women and disability, however integrating violence against women questions in disability surveys may be even more fruitful. This scoping review identifies a starting point for building a stronger evidence-base, namely, improving understanding of appropriate, feasible and valid measures and methodologies to shed light on the intersection between disability and violence against women. As such, the aims of this scoping review are to map definitions, measures and methodologies in quantitative literature on violence against women with disabilities. This scoping review is designed to inform current debates and discussions regarding how to generate evidence concerning violence against women with disabilities.

We cover three bodies of literature: i) measurement of violence within the context of disability-focused research, ii) measurement in research focused on the intersection of disability and violence, and iii) measurement of disability in the context of research focused on violence against women. We focus on studies utilizing quantitative methodologies, given our scoping review emerges from data requirements for the Sustainable Development Goals [SDGs] and seeks to strengthen quantitative population-based surveys of violence against women. We aimed towards a comprehensive snapshot of the existing literature, and in this manuscript present a descriptive analysis of the results, which includes disparate bodies of literature. Future sub-analyses will provide more in-depth exploration of specific topics within the literature, for example, comparison of types of perpetrator and locations of violence experiences.

## Methods

Based on our understanding of the state of knowledge on measurement of disability and violence against women, and the needs of researchers, program and policy makers, and advocates, we selected a scoping review as the most appropriate methodological approach for our broad, exploratory research question [29]. We followed Arksey and O'Malley's framework for design of a scoping review [30]. Further details on definitions employed in this scoping review are available in the published protocol, where there is also further discussion of the rationale for selecting a scoping review [14]. Our reporting of methods and findings follows the PRISMA Checklist [S1 Appendix] and scoping review specific reporting guidelines [31, 32].

For the purpose of our review, we define disability-focused research as quantitative research seeking to estimate the prevalence of disability or identify associations between disability and other health outcomes. We define research focused on the intersection of disability and violence as research that focuses on associations between disability and violence, without being solely focused on either disability or violence as an outcome. We define measurement of

disability in VAW research as research that focuses on questions of prevalence of violence that measure disability as a specific risk factor or variable within study objectives focusing on understanding VAW in a population or specific group.

## Search strategy

**Peer-reviewed literature.** The following electronic databases were included in the systematic search of the peer-reviewed literature: PubMed, PsycINFO, Embase, CINAHL, Web of Science, PILOTS, Sociological Abstracts, ERIC, AgeLine, Social Work Abstracts, International Bibliography of the Social Sciences, Social Services Abstracts, ProQuest Criminal Justice, ASSIA, Dissertations & Theses Full Text, and Dissertations & Theses Global. We developed a search strategy for these databases based on the following domains of the research question: disability; women; violence; and quantitative research. For each of these domains, we identified the relevant keywords and search terms, which varied by database [see Table 1, PubMed Search Strategy]. The search strategy was appropriately modified for each database, including syntax and specific terms, topics and/ or headings. The search was not limited by year of publication or type of publication. The reference lists of all systematic, scoping or other literature reviews identified in the search of electronic databases were hand searched, and any potentially relevant titles added to the review and subjected to the same screening and inclusion/ exclusion criteria as articles identified in database searches.

**Grey literature, including national disability or violence studies.** The grey literature search was implemented by one author (SRM), who conducted structured google searches: "Country X disability survey," "Country X disability study" and "Country X disability statistics," for each country, reviewing 10 pages of results per search. We also searched the websites

**Table 1. PubMed search strategy.**

| | |
|---|---|
| 1 | "Intellectual disability"[MeSH] OR "Communication disorders"[MeSH] OR "Developmental disabilities"[MeSH] OR "Mentally Disabled persons"[MeSH] OR "Disabled persons"[MeSH] OR "physical disabilit*"[TIAB] OR "physically disabled"[TIAB] OR "intellectual disabilit*"[TIAB] OR "handicap"[TIAB] OR "functional impairment"[TIAB] OR "mental disorder*"[TIAB] OR "mentally disabled"[TIAB] OR "mental disability*"[TIAB] |
| 2 | Women[MeSH] OR female[MeSH] OR wife[TIAB] OR spouses[MeSH] OR wives[TIAB] OR "female partner*"[TIAB] OR spouse*[TIAB] |
| 3 | "Elder abuse"[MeSH] OR "domestic violence"[MeSH] OR "Intimate Partner Violence"[MeSH] OR "battered women"[MeSH] OR "violence"[MeSH] OR "aggression"[MeSH] OR "spouse abuse"[MeSH] OR "Physical Abuse"[MeSH] OR Rape [MeSH] OR "elder neglect"[TIAB] OR "elder mistreatment"[TIAB] OR "elder maltreatment"[TIAB] OR "assault"[TIAB] OR "sexual abuse"[TIAB] OR "sexual assault"[TIAB] OR "rape"[TIAB] OR "psychological abuse"[TIAB] OR "psychological violence"[TIAB] OR "emotional abuse"[TIAB] OR "emotional violence"[TIAB] OR "neglect"[TIAB] OR "economic abuse"[TIAB] OR "verbal abuse"[TIAB] OR "violence against women"[TIAB] OR "abused women"[tiab] OR "intimate terrorism"[tiab] OR "marital rape"[tiab] OR "wife beating"[tiab] OR "relationship aggression"[tiab] |
| 4 | "epidemiologic methods"[MeSH] OR "Comparative Study"[Publication Type] OR "outcome and process assessment (health care)"[Mesh] OR "statistics and numerical data"[Subheading] OR "Evaluation Studies"[Publication Type] OR "meta analysis"[Publication Type] OR "multicenter study"[Publication Type] OR "incidence"[TIAB] OR "surveillance"[TIAB] OR "prevalence"[TIAB] OR "epidemiology"[subheading] OR "Health Care Evaluation Mechanisms"[Mesh] OR "morbidity"[TIAB] OR "burden"[TW] OR "Cross sectional study"[MeSH] OR "case-control studies"[MeSH] OR "Cohort studies"[MeSH] OR "Surveys and questionnaires"[MeSH] OR "cross-sectional stud*"[TIAB] OR "quantitative survey"[TIAB] OR "survey"[TIAB] |
| 1 AND 2 AND 3 AND 4 | |

of National Statistics Offices for all countries to identify any national or sub-national disability research, as well as identifying national violence against women studies and Demographic and Health Surveys that have included both disability and violence against women modules, through consultation with experts. We contacted three experts in the field of research on violence and/ or disability measurement and requested that they provide any relevant literature to be considered for inclusion in the review.

## Inclusion and exclusion criteria

Studies were eligible for inclusion in the scoping review if the study:

i. Utilized a quantitative methodology; mixed methods studies were included if the quantitative data were reported separately; and

ii. Compared women with disability to women without disability (studies including men and women with disability were included if sex-specific analyses were included) OR included only women with disability; and

iii. Assessed exposure to any form of violence; and

iv. Examined violence experienced as an adult, aged 15 and older (studies including violence experienced before the age of 15 were included if violence experienced above 15 was also measured).

Studies were excluded if the study:

i. Focused only on common mental health disorders (depression, anxiety, post-traumatic stress disorder [PTSD]); or

ii. Focused only on violence experienced before the age of 15; or

iii. Only utilized data from case studies or client files; or

iv. Was only based on caregiver report and/ or forensic exam; or

v. Focused only validity/ reliability of the measure or scale development.

These exclusion criteria were developed to ensure that the identified literature addresses the specific study aims. Common mental disorders were excluded as there is a robust evidence-base on VAW and common mental disorders. This evidence-base includes several systematic reviews and meta-analyses [33–35], and therefore we focused this review on an area with less well-developed measurement and methodology.

One of the exclusion criteria listed in our protocol was that studies should be excluded if they "only compared women with disability to men with disability" [14]. As we conducted initial title and abstract review and full text review, we identified several studies that would have been excluded based on these criteria that appeared to have important implications for our research question. As such, based on discussion with the research team, we adjusted this and removed this from the exclusion criteria. We included studies published in English, French and Spanish.

## Study selection

We utilized EndNote V.X7 as our bibliographic software management platform. We removed duplicates using EndNote, prior to exporting titles and abstracts to an Excel spreadsheet for review. For all peer-reviewed literature, two authors [SRM and HS] independently reviewed titles and abstracts, to determine which should be included for full text review. Where there

was a discrepancy, the study was included for full text review. SRM and HS reviewed all studies selected for full text review against the inclusion and exclusion criteria. Any discrepancies were resolved through discussion and reasons for excluding articles were recorded. One author [SRM] reviewed the grey literature for possible inclusion; given the volume of grey literature identified, double screening of all grey literature considered for inclusion was not feasible. A flow diagram presenting the process of identification of all included studies is represented in Fig 1, Identification of Included Studies.

### Data extraction

A data extraction template was developed specifically for the purposes of the review, and included the following three over-arching categories: study characteristics (such as study setting and data collection method); measurement of violence (including types of violence measured, scale utilized and specific items); and measurement of disability (including type(s) of disability(ies), scale utilized and specific items). The full list of data extraction variables is included in S2 Appendix.

The data extracted for these three over-arching categories directly responded to our research questions and formed the basis of our analysis. We also extracted data on the findings of each study, data analysis methods and reported study limitations, which are not reported in this review. These data may be used to inform future research questions for subsequent reviews, based on the findings of this scoping review.

While the original protocol specified that one author [SRM] would conduct all data extraction, the volume of studies identified as meeting inclusion criteria necessitated adding a second data extractor [CV] and data for the Spanish-language studies was extracted by AO [see Acknowledgments]. The accuracy of all data extraction was reviewed by SRM.

We did not conduct quality assessment, given that the objective of the scoping review was to map the current measurement approaches, and not to ascertain bias of findings or quality of existing studies overall [36].

### Data analysis

We present the findings narratively under the following sub-headings: study characteristics, violence measurement and disability measurement. The aim of our scoping review is primarily to describe measurement approaches to disability and violence against women, however, description and discussion of study characteristics, in particular research questions, is relevant in understanding the approaches to and framing of definitions and measurement of violence against women. We sought to identify typologies of research questions within the included studies, and provide illustrative examples. Within our description and analysis of measurement approaches to violence, we present the types of violence measured, types of measurement, including names of scales and items (where described and included), and assessed quality of measurement by indicating whether acts-specific items were included for all, some or none of the types of violence measured. We describe whether disability-specific forms of violence are measured and how. For measurement approaches to disability, we categorise the types of disability assessed into four categories, which mirror the categories of disability included in the Convention on the Rights of Persons with Disability–physical, mental, intellectual or sensory impairments [37]. We categorise the measurement of disability into three approaches: assessments measuring functioning limitations; single items (binary yes/ no self-report of disability) and assessment based on a health condition or impairment.

## Identification of Included Studies

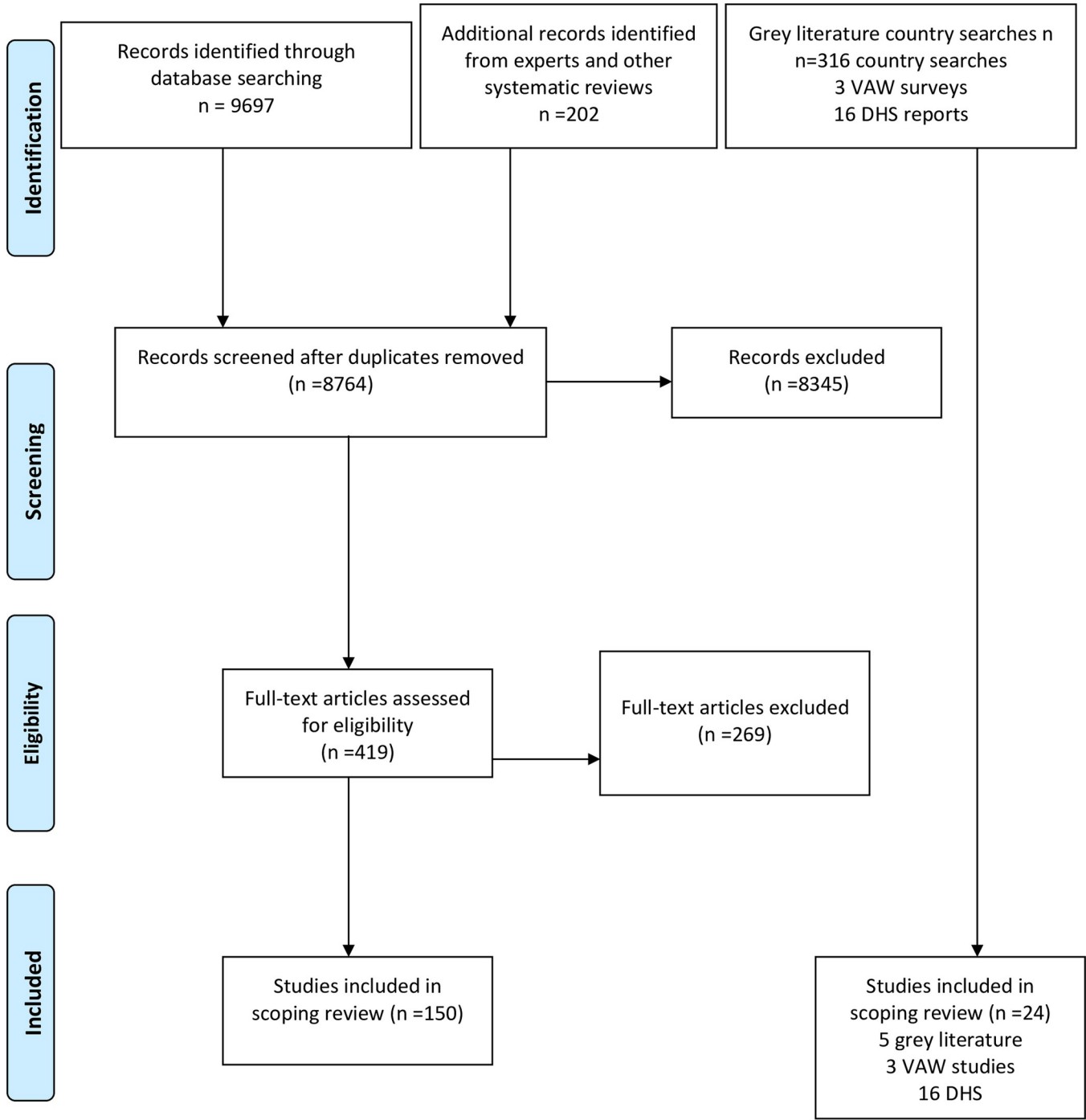

*From:* Moher D, Liberati A, Tetzlaff J, Altman DG, The PRISMA Group (2009). *P*referred *R*eporting *I*tems for *S*ystematic Reviews and *M*eta-*A*nalyses: The PRISMA Statement. PLoS Med 6(6): e1000097. doi:10.1371/journal.pmed1000097

**For more information, visit www.prisma-statement.org.**

**Fig 1. Identification of included studies.**

## Patient and public involvement

Patients were not involved in the development of this scoping review. Members of the public were not consulted specifically for the development of the research questions, however, previous research and consultations with experts has indicated that this is a relevant and important area of enquiry in the field of violence against women research.

# Results

## Literature search

The search of 16 electronic databases yielded 9697 articles and review of reference lists of existing systematic or other literature reviews or input from experts yielded an additional 202 articles. After duplicates were removed, 8764 titles and abstracts were included for initial screening, and 419 selected for full-text review. Following full-text review, an additional 269 were excluded, for a total of 150 peer-reviewed manuscripts

The grey literature search was conducted separately, and identified 316 reports, of which 5 were selected for inclusion. In addition, 16 Demographic and Health Surveys and 3 reports of national violence against women studies met the inclusion criteria. With the 150 peer-reviewed manuscripts, a final 174 reports or manuscripts met the inclusion criteria.

## Study characteristics

The study characteristics, including country and region of focus, research question, study sample, data collection methods and type of comparison(s) included, are shown in Table 2, Characteristics of Included Studies. Among the 174 reports or manuscripts included, n = 97 were conducted in the WHO Americas region [AMRO], n = 42 were conducted in the WHO European region [EURO], n = 5 in the WHO Eastern Mediterranean region [EMRO], n = 15 in WHO Western Pacific region [WPRO], n = 8 in WHO South East Asia region [SEARO] and n = 17 in WHO Africa region [AFRO] (numbers add up to more than 174 as some studies were conducted in multiple countries/ regions). A total of 52 of the studies were conducted in low or middle-income country contexts. The vast majority of studies were cross-sectional surveys. A small number of studies (n = 8) reported results from a wave or multiple waves of a longitudinal survey [38–45] and some studies employed a case-control study design [46–59].

## Typologies of studies

We identified over-arching typologies of studies, based on research question and/ or sampling approach. The first typology were studies where all respondents identified or self-identified as having a disability. These studies looked at prevalence of, risk factors for or impacts of violence exposure amongst persons with disability. Within this typology, we identified n = 42 studies which included only women with disabilities. Amongst these studies, research questions included a focus on prevalence, socio-demographic correlates and risk factors (for example, [43, 60–72]), as well as some focus on other themes, for example, barriers to help-seeking [73] and perceived social support [74]. A small number of these studies included comparison of violence experiences by type of disability (for example, Walker (1997) [54] and Friedman et al. (2011) [44].

Within this first typology, there were n = 22 studies which included only men and women with disabilities. Four of these studies explicitly included within the research objectives the aim of exploring the gender dimensions of the association between disability and violence victimization [75–78]. In other studies, while comparison of male and female experiences is not part of the research question, the differences in exposure to violence between male and female

**Table 2. Characteristics of included studies.**

| Article | Country and region | Research question(s) | Sample–N, % women, % with disability | Methods–study design, data collection methods | Comparison of violence experiences–men vs. women; women with disabilities vs. women without disabilities |
|---|---|---|---|---|---|
| Pandey et al., 2012 [96] | India [SEARO] | 1. What is the effect of women's empowerment on developing blindness during pregnancy? 2. Testing the hypothesis that women's experience of domestic abuse (control, humiliation, and physical violence) from their spouses increases the risk for blindness during pregnancy. | N = 35,248 women in full study; 100% women; 11.87% reported night blindness. n = 19,902 included in analysis of associations between night blindness and domestic violence as 44% of full sample were missing data on domestic violence | Cross-sectional; face-to-face standardized questionnaire | Women with disabilities vs. women without disabilities |
| Valera and Kucyi, 2017 [125] | USA [AMRO] | What are the associations among brain injury severity, cognitive functions and right anterior insula [rAI]–Default Mode Network [DMN] structural and functional connectivity? | N = 20; 100% women, all had experience of Traumatic Brain Injury | Cross-sectional; face-to-face standardized questionnaire and brain imaging scans | No comparisons |
| Valera et al., 2019 [162] | USA [AMRO] | 1. What are the possible microstructural alterations associated with IPV-related moderate Traumatic Brain Injury [mTBI]? 2. What is the association between IPV-related mTBI and diffusion within select ROIs [regions of interest] in the brain, and what are possible associations between FA of our ROIs and several facets of cognitive functioning? | N = 20; 100% women, all had experience of Traumatic Brain Injury | Cross-sectional; face-to-face standardized questionnaire and brain imaging scans | No comparisons |
| Slayter, 2009 [163] | USA [AMRO] | 1. What is the prevalence of past-year IPV and its subtypes amongst women with and without disabilities? 2. To test the hypothesis that: women with disabilities would be more likely to experience any IPV or any particular subtype of IPV (i.e., control tactics consisting of verbal abuse and coercion as well as threats or physical violence). | N = 822, 100% women; n = 141 women with disabilities | Cross-sectional; face-to-face standardized questionnaire | Women with disabilities vs. women without disabilities |
| Powers, 2002 [60] | USA [AMRO] | 1. To what extent do women with disabilities experience abuse, including abuse by personal assistance service providers? 2. What forms of personal assistance abuse do women with disabilities consider most hurtful? 3. What are the most critical barriers that impede women's handling of personal assistance abuse? 4. What are the most important strategies that women and others can use to prevent and/ or stop personal assistance abuse? | N = 200; 100% women, 100% with disability | Cross-sectional; standardized questionnaire–mainly telephone and self-administered (mailed in), some face-to-face to accommodate cognitive disabilities | No comparisons |

(*Continued*)

**Table 2.** (Continued)

| Article | Country and region | Research question(s) | Sample–N, % women, % with disability | Methods–study design, data collection methods | Comparison of violence experiences–men vs. women; women with disabilities vs. women without disabilities |
|---------|-------------------|---------------------|------------------------------------|-----------------------------------------------|---------------------------------------------------------------------------------------------------------|
| Alangea et al., 2018 [164] | Ghana [AFRO] | Descriptive exploratory analysis of baseline survey for evaluation of Rural Response System, Violence against Women prevention program in four rural districts in Ghana | N = 2000; 100% women; n = 140 reported disability | Cross-sectional; face-to-face standardized questionnaire, recorded with personal digital assistant tablet | No comparisons; disability utilized as variable |
| Astbury and Walji, 2014 [46] | Cambodia [WPRO] | 1. Whether and to what extent women with disabilities [WWDs] differed from non-disabled women on socio-demographic factors that function as proxy variables for human rights violations such as being denied the right to education, in lifetime prevalence rates of different types of household violence, coercive control, and injury and in the extent of psychological distress associated with such violence. 2. To identify whether WWDs compared with non-disabled women exhibited different patterns of disclosure and access to health care following injury as a result of household violence. | N = 354; 100% women; n = 177 women with disabilities | Cross-sectional–case-control; face-to-face standardized questionnaire | Women with disabilities vs. women without disabilities |
| Cannell et al., 2015 [38] | USA [AMRO] | 1. To investigate the association between physical and verbal abuse and physical function in a large cohort of postmenopausal women, aged 50–79 (at baseline). 2. To explore the hypothesis that women who experienced physical and/or verbal abuse in the year prior to baseline would have lower levels of physical function at baseline compared to women who did not experience abuse 3. To explore the hypothesis that women with baseline abuse exposure would have a greater rate of decline in physical functioning over time. | N = 154,902, 100% women. Overall disability for whole sample at baseline not reported | Longitudinal; clinical interviews and structured questionnaire; combination of face-to-face and self-administered | Women with disabilities vs. women without disabilities |
| Coston, 2019 [111] | USA [AMRO] | 1. Are there significant differences between heterosexual disabled women and bisexual disabled women in their experiences of intimate partner violence victimization? 2. Are there significant differences between heterosexual disabled women and bisexual disabled women in the mental health-related outcomes of intimate partner violence? | N = 3542; 100% women. n = 137 women (3.9%) with disabilities | Cross-sectional; structured questionnaire; administered over telephone | Women with disabilities vs. women without disabilities |

(*Continued*)

**Table 2.** (Continued)

| Article | Country and region | Research question(s) | Sample–N, % women, % with disability | Methods–study design, data collection methods | Comparison of violence experiences–men vs. women; women with disabilities vs. women without disabilities |
|---|---|---|---|---|---|
| Dembo et al., 2018 [39] | USA [AMRO] | 1. To examine the extent to which violence victimization is associated with mental health disparities between people with and without disabilities, by comparing the psychological consequences of violence experienced by adults with disabilities in the U.S., and comparing findings to the psychological outcomes reported by adults without disabilities. 2. To examine differential effects of violence by gender by stratifying analyses by gender | N = 8070; 4115 women (51%); n = 877 women with disability (21% of women) | Cross-sectional (pooled responses from several waves of survey); data collected as part of National Crime Victimization Survey; methods not described in this manuscript | Women with disabilities vs. women without disabilities; Men with disabilities and women with disabilities |
| Emerson et al., 2016 [40] | UK [EURO] | 1. To examine perceptions of safety and exposure to violence in public places among working age adults with and without disabilities in the UK 2. To assess the extent to which any between-group differences may be moderated by gender and socio-economic situation. | N = 3,454 men and women; approx. 50% women; n = 490 women with disability | Cross-sectional (data drawn from one wave of longitudinal study); standardized interview; face-to-face CAPI | Women with disabilities vs. women without disabilities |
| Gibbs et al., 2018 [165] | Afghanistan [EMRO] | 1. To describe the factors associated with recent IPV amongst a group of currently married women in Afghanistan 2. To describe whether IPV is independently associated with health outcomes amongst these married women. | N = 935; 100% women; proportion reporting disability not stated | Cross-sectional; structured questionnaire; face-to-face | No comparisons, disability addressed as a variable |
| Guedes et al., 2016 [87] | Canada, Albania, Colombia, Brazil [EURO, EMRO] | 1. To examine associations between the experiences of childhood abuse and domestic violence throughout the life course and mobility disability in old age 2. To explore the significance of possible pathways for these associations 3. To examine differences between men and women in these associations. | N = 1995; 1040 women. Overall (across 4 sites), 48.2% of women reported mobility disability and 19.1% of women had poor physical performance | Cross-sectional; structured questionnaire, video-based and direct measurement of disability; face-to-face computer assisted | Women with disabilities vs. women without disabilities; Men with disabilities and women with disabilities |
| Kutin et al., 2017 [133] | Australia [WPRO] | To identify lifetime prevalence of economic abuse in Australia by age and gender, and associated risk factors | N = 17050; 13,307 women; % of women with disability not reported | Cross-sectional; structured questionnaire; face-to-face | No comparisons, disability addressed as a variable |
| Le et al., 2016 [166] | Vietnam [WPRO] | 1. To examine associations between exposure to individual forms of victimisation and health-related quality of life of adolescents in Vietnam 2. To examine associations between exposure to poly-victimisation and the health-related quality of life of adolescents in Vietnam. | N = 1616 total; 729 female; % of women with disability not reported | Cross-sectional; structured questionnaire; self-administered | No comparisons, disability addressed as a variable |

(*Continued*)

**Table 2.** (Continued)

| Article | Country and region | Research question(s) | Sample–N, % women, % with disability | Methods–study design, data collection methods | Comparison of violence experiences–men vs. women; women with disabilities vs. women without disabilities |
|---|---|---|---|---|---|
| Platt et al., 2017 [75] | USA [AMRO] | 1. Explore impact of gender on associations between violence and health among people with developmental disabilities 2. To examine gender similarities and differences in lifetime prevalence of various forms of violence, perpetrator identities and behaviors, and health status associated with abuse. | N = 350; 177 women; 100% with developmental disabilities | Cross-sectional; structured questionnaire; ACASI self-administered | Men with disabilities and women with disabilities |
| Puri et al., 2015 [61] | Nepal [SEARO] | What are the prevalence and risk factors of violence amongst women with disabilities? | N = 475; all women, all with disabilities | Cross-sectional; structured questionnaire; face-to-face | No comparisons |
| Slayter et al., 2017 [112] | USA [AMRO] | 1. What is the nature of IPV among transitional-aged adult women (aged 18–21) with disabilities using population-based data 2. How do risk factors for IPV for transitional-aged women with disabilities compare to risk factors for IPV for transitional-aged women without disabilities? 3. What is the prevalence of past-year IPV among transitional-aged women with disabilities? 4. How does the prevalence of past-year IPV among transitional-aged women with disabilities compare to the prevalence rate of past-year IPV among transitional-aged women without disabilities? | N = 9,170,271, all women; n = 1,616,207 with self-reported disability | Cross-sectional; structured questionnaire; computer-assisted telephone interview | Women with disabilities vs. women without disabilities |
| Valentine et al., 2019 [97] | Uganda [AFRO] | What is the prevalence and consequences of IPV exposure among Ugandan women with disabilities? | N = 8592; all women; n = 299 reported disability | Cross-sectional; structured questionnaire; face-to-face | Women with disabilities vs. women without disabilities |
| Wall et al., 2018 [134] | USA [AMRO] | 1. To test previously reported research suggesting an increased rate of violence-related traumatic brain injury [TBI] among justice-involved women. 2. To describe the characteristics of justice-involved women who sustain these injuries; and to identify additional vulnerabilities associated with TBIs among justice-involved women by investigating the physical health, mental health, and criminal behavior characteristics that may be associated with violence-related TBIs in these women. | N = 409; 135 women. % with disability not reported | Cross-sectional; screening for TBI upon entry into jail or probation, then semi-structured clinical interview; face-to-face | No comparisons |

(*Continued*)

**Table 2.** (Continued)

| Article | Country and region | Research question(s) | Sample–N, % women, % with disability | Methods–study design, data collection methods | Comparison of violence experiences–men vs. women; women with disabilities vs. women without disabilities |
|---|---|---|---|---|---|
| Mirindi 2018 [145] | Democratic Republic of Congo [AFRO] | What is the association between fistulas, other sexual rape-related injuries, post-traumatic stress disorder (PTSD), feelings of worthlessness, social rejection, support from family/friends, and chronic pain and depression among women victims of rape in eastern Democratic Republic of Congo. | N = 156; all women, n = 97 experiencing fistula | Cross-sectional; structured questionnaire, face-to-face | No comparisons |
| Anderson et al., 2012 [64] | USA [AMRO] | 1. What is the prevalence of violent behaviors experienced by Deaf female undergraduates in their past-year relationships? 2. What proportion of these relationships are identified as "abuse"? 3. What scripts and strategies do Deaf female undergraduates utilize to label their experiences of partner violence? | N = 97 women; all deaf | Cross-sectional; structured questionnaire; self-administered | No comparisons |
| Anderson et al., 2014 [63] | USA [AMRO] | 1. To investigate the prevalence, correlates, and characteristics of intimate partner violence victimization in hearing–Deaf and Deaf–Deaf relationships 2. What is the prevalence of intimate partner violence in hearing–Deaf versus Deaf–Deaf relationships? 3. What are the correlates and characteristics of intimate partner violence in hearing–Deaf versus Deaf–Deaf relationships? | N = 97 women; all deaf | Cross-sectional; structured questionnaire; self-administered | No comparisons |
| Anderson et al., 2011 [62] | USA [AMRO] | To ascertain the prevalence and nature of IPV victimization in a sample of Deaf female college students. | N = 100 women; all deaf | Cross-sectional; structured questionnaire; self-administered | No comparisons |
| Barrett et al., 2009 [114] | USA [AMRO] | 1. To describe the prevalence of IPV among women with disabilities and compare IPV prevalence among women with and without disabilities 2. To examine whether health status and health care access differ between women with disabilities experiencing IPV and those who have not 3. To examine the association between IPV, health status and health care access among women with disabilities. | N = 23,154, all women; n = 6,309 reported having a disability | Cross-sectional; structured questionnaire; administered over telephone | Women with disabilities vs. women without disabilities |

(*Continued*)

**Table 2.** (Continued)

| Article | Country and region | Research question(s) | Sample–N, % women, % with disability | Methods–study design, data collection methods | Comparison of violence experiences–men vs. women; women with disabilities vs. women without disabilities |
|---|---|---|---|---|---|
| Brownridge 2006 [98] | Canada [AMRO] | 1. To examine the risk for partner violence against women with disabilities relative to women without disabilities. 2. To identify whether Canadian women with disabilities report an elevated risk for partner violence compared to their counterparts without disabilities and, if so 3. To examine the extent to which disabled women's risk is elevated 4. To examine risk markers derived from potential explanations in terms of their impact on, and the extent to which they account for an elevated risk of, partner violence against women with disabilities. | N = 7,207; all women; n = 1,092 were disabled [15.5%] | Cross-sectional; structured quantitative survey; administered over telephone | Women with disabilities vs. women without disabilities |
| Brownridge 2008 [99] | Canada [AMRO] | 1. To examine the elevated risk for male-female IPV against women with disabilities compared to women without disabilities across three large-scale Canadian surveys. 2. To test an explanatory framework that organized risk markers based on whether they referred to the context of the relationship between the couple (relationship factors), the victim (victim-related characteristics), or the perpetrator (perpetrator-related characteristics). | Samples came from three surveys: i. N = 8,417 women (1,268 with disabilities and 7,149 without disabilities) ii. N = 7,027 women (1,092 with disabilities and 5,935 without disabilities) iii. 6,769 women (748 with disabilities and 5,866 without disabilities) Total N = 22,213 (3,108 with disabilities) | Cross-sectional; structured quantitative survey; administered over telephone | Women with disabilities vs. women without disabilities |
| Brunnberg et al., 2012 [88] | Sweden [EURO] | 1. To compare the incidence of force on the first occasion of sexual intercourse reported by participants with disabilities to that of students without disabilities 2. To determine whether there are significant differences in mental health, substance abuse, and school performance as reported by participants forced into their sexual debut as opposed to those who were not forced, analysed by gender 3. To identify the significant variables that predict girls reporting force at sexual debut as opposed to girls not reporting force, as well as to identify similar variables within the male group. | N = 4748; 2377 female; 330 (14%) with one or multiple disabilities | Cross-sectional; structured quantitative survey; self-administered | Women with disabilities vs. women without disabilities |

(*Continued*)

**Table 2.** (Continued)

| Article | Country and region | Research question(s) | Sample–N, % women, % with disability | Methods–study design, data collection methods | Comparison of violence experiences–men vs. women; women with disabilities vs. women without disabilities |
|---|---|---|---|---|---|
| Curry et al., 2009 [139] | USA [AMRO] | To describe the development and psychometric evaluation of questions designed to assess violence perpetrator risk characteristics experienced by women with disabilities and deaf women. | N = 305; all women, all with disabilities | Cross-sectional; structured quantitative survey; self-administered using ACASI | No comparison |
| Du Mont et al., 2013 [113] | USA [AMRO] | 1. To determine whether women with disabilities were satisfied with the services provided and differed from women with no disabilities in their opinions of the process and provider 2. To compare women with and without disabilities in terms of their (a) presentation characteristics, (b) socio-demographic characteristics, c) social supports, and (d) assault characteristics. | 920; all women, 21% reported some disability | Cross-sectional; structured quantitative survey; self-administered | Women with disabilities vs. women without disabilities |
| Pollard et al., 2014 [159] | USA [AMRO] | To obtain IPV prevalence rate data reported by Deaf community samples and compare them, where possible, to data from general population samples. | Two samples for this study: N = 308, 53.9% female; all deaf N = 162, 49.7% female; all deaf Compared to national surveys [N = 1906, N = 16,000] | Cross-sectional; structured quantitative interview; Self-administered, interactive touch-screen computer interface | Comparison of men and women, with and without disabilities |
| Powers et al., 2009 [73] | USA [AMRO] | 1. What types of safety promoting behaviors do women with disabilities and deaf women report using to address interpersonal violence?2. How do women's use of safety promoting behaviors relate to their experience of interpersonal violence and their exposure to perpetrator risk characteristics? | N = 305; all women, all with disabilities | Cross-sectional; structured quantitative survey; self-administered using ACASI | No comparison |
| Smith et al., 2008 [115] | USA [AMRO] | To examine the relationship between the employment status of women with disabilities and the incidence of physical and sexual abuse in the United States | N = 219,911 women; n = 4574 reported activity limitation | Cross-sectional; structured quantitative survey; telephone | Women with disabilities vs. women without disabilities |
| Alriksson-Schmidt et al., 2010 [100] | USA [AMRO] | To investigate whether US female adolescents who self-reported having a physical disability or long-term health problem were more likely to report having been physically forced to have sexual intercourse than US female adolescents without a physical disability or long-term health problem. | 7193 female adolescents; 12.5% reported having a disability or long-term health problem | Cross-sectional; structured quantitative survey; not specified | Women with disabilities vs. women without disabilities |

(*Continued*)

**Table 2.** (Continued)

| Article | Country and region | Research question(s) | Sample–N, % women, % with disability | Methods–study design, data collection methods | Comparison of violence experiences–men vs. women; women with disabilities vs. women without disabilities |
|---|---|---|---|---|---|
| Carbone-López et al., 2006 [89] | USA [AMRO] | 1. To model patterns of IPV separately for males and females 2. To assess the unique contributions of different types of violence exposure to health related outcomes. | N = 111,858; 55,991 female; 8% of women reported disability due to injury, 11% reported chronic disease | Cross-sectional; structured quantitative survey; telephone | No comparison |
| Eberhard-Gran et al., 2007 [101] | Norway [EURO] | To study the associations between recent and repetitive exposure to violence and presence of somatic symptoms and diseases in women. | N = 2730, all women; overall prevalence of disability not reported | Cross-sectional; structured quantitative survey; self-administered | No comparison |
| Haydon et al., 2011 [41] | USA [AMRO] | To determine the association between unwanted sex and both physical disability and cognitive performance in a nationally representative sample | N = 11,878; 6450 female. 5.8% of women reported physical disability | Longitudinal; structured quantitative survey; not specified | Women with disabilities vs. women without disabilities |
| Morris et al., 2019 [65] | UK [EURO] | 1. To explore the history of experienced and perpetrated IPV in women detained to secure specialist intellectual disability [ID] forensic service 2. To explore the prevalence and different types of IPV that were experienced and perpetrated in a detained female forensic ID population; and 3. To explore if there are differences in the levels of experienced and perpetrated IPV in a detained female forensic ID population. | N = 16; all women, all with intellectual disability | Cross-sectional; structured quantitative survey; face to face | No comparison |
| Rasoulian et al., 2014 [126] | Iran [EMRO] | To determine lifetime and past-year prevalence of exposure to physical violence among married women in the city of Tehran and urban and rural areas of Hashtgerd. | N = 1000, all women; 22.6% reported illness or disability | Cross-sectional; structured quantitative survey; face to face | No comparison |
| Stöckl et al., [102] | Germany [EURO] | To explore if there are risk and protective factors that, if identified and understood, could help inform the design of interventions to prevent partner violence | N = 3866; all women; 14% reported disability | Cross-sectional; structured quantitative survey; face to face and self-administered (two separate components of survey) | No comparison |
| Brownridge et al., 2016 [90] | Canada [AMRO] | To compare the risk of IPV against men and women with and without activity limitations | N = 15,010; 56% women; 20% of women reported some activity limitation | Cross-sectional; structured quantitative survey; telephone | Women with disabilities vs. women without disabilities; comparison of men and women, with and without disabilities |
| Casteel et al., 2008 [42] | USA [AMRO] | To examine the association between the level of disability impairment and physical and sexual assault in a sample of US women at least 18 years of age. | N = 6273 women; 20.2% reported severe or moderate disability | Retrospective longitudinal; structured quantitative survey; telephone | Women with disabilities vs. women without disabilities |

(*Continued*)

**Table 2.** (Continued)

| Article | Country and region | Research question(s) | Sample–N, % women, % with disability | Methods–study design, data collection methods | Comparison of violence experiences–men vs. women; women with disabilities vs. women without disabilities |
|---|---|---|---|---|---|
| Cimino et al., 2019 [167] | USA [AMRO] | . To determine the prevalence of probable TBI and its association with comorbid PTSD and depression among Black women 2. To examine the relationship between past experiences of violence and probable TBI 3. To assess the effect of past violence and probable TBI on mental health disorders (i.e., comorbid depression and PTSD). | N = 95 women; n = 33 with probable TBI | Retrospective cohort; structured quantitative survey; self-administered with ACASI | Women with disabilities vs. women without disabilities |
| Cohen et al., 2005 [168] | Canada [AMRO] | To determine the prevalence of IPV in the previous five years among women reporting activity limitations | N = 8771 women; n = 1483 reported activity limitation | Cross-sectional; structured quantitative interview; telephone (computer-assisted) | Women with disabilities vs. women without disabilities |
| Cohen et al., 2006 [91] | Canada [AMRO] | To compare IPV against men and women with activity limitations | N = 16,216; 8771 women. n = 1521 women (17.3%) reported activity limitations | Cross-sectional; structured quantitative interview; telephone (computer-assisted) | Women with disabilities vs. women without disabilities; and men and women with disabilities |
| Curry et al., 2011 [140] | USA [AMRO] | 1. To describe the facilitators and barriers to abuse disclosure endorsed by a sample of women with diverse disabilities who have a history of abuse 2. To examine the extent to which health professionals have discussed violence and personal safety to women with disabilities 3. To explore demographic, disability, and abuse-related characteristics in relationship to facilitators and barriers, disclosure, and discussion of violence by a health provider | N = 305; all women, all with disabilities | Cross-sectional; structured quantitative survey; self-administered with ACASI | No comparison |
| Elliott Smith et al., 2015 [66] | USA [AMRO] | 1. To explore the prevalence rate of sexual assault among deaf female undergraduate students and their acknowledgment of assault 2. To examine the demographic and background characteristics of both the survivors and the assailants. | N = 70 women, all deaf | Cross-sectional; structured quantitative survey; self-administered on computer | No comparison |
| Hahn et al., 2014 [67] | USA [AMRO] | 1. To explore the prevalence rate of sexual assault among deaf female undergraduate students and their acknowledgment of assault 2. To examine the demographic and background characteristics of both the survivors and the assailants. | N = 70 women, all deaf | Cross-sectional; structured quantitative survey; self-administered on computer | No comparison |

(*Continued*)

**Table 2.** (Continued)

| Article | Country and region | Research question(s) | Sample–N, % women, % with disability | Methods–study design, data collection methods | Comparison of violence experiences–men vs. women; women with disabilities vs. women without disabilities |
|---|---|---|---|---|---|
| Hasan et al., 2014 [169] | Bangladesh [SEARO] | 1. To estimate the prevalence of IPV in a sample of 226 women with disabilities living in four different districts of Bangladesh 2. To explore the physical and psychological suffering of women experiencing violence and their various coping strategies. | N = 226 women; all with disabilities | Cross-sectional; structured quantitative survey; face-to-face | No comparison |
| Johnston-McCabe et al., 2011 [74] | USA [AMRO] | 1. To examine domestic violence and perceived social support in a clinical sample of Deaf and Hard of Hearing women 2. To explore the nature of the abuse between Deaf and Hard of Hearing women and their partners to ascertain possible distinctive characteristics of abuse within this unique subgroup 3. To explore perceived social support as it related to domestic violence in these relationships. | N = 46 women, all deaf | Cross-sectional; structured quantitative survey; different options for mode of administration: i) use of a sign language interpreter to translate the measures (21.7%); ii) self-administered (34.8%); or iii) paper and pencil method with the option of having support staff/interpreter available to help with translation of items 43.5%). | No comparison |
| Krnjacki et al., 2016 [92] | Australia [WPRO] | 1. To estimate the prevalence of violence for men and women according to disability status 2. To compare the risk of violence among women and men with disabilities to their same-sex non-disabled counterparts 3. To compare the risk of violence between women and men with disabilities. | N = 17,750; 13,307 women; population-weighted prevalence of disability or long-term health condition for women was 32% | Cross-sectional; structured quantitative survey; face to face | Women with disabilities vs. women without disabilities; and men and women with disabilities |
| Martin et al., 2006 [151] | USA [AMRO] | To compare women with and without disabilities in terms of the prevalence of physical and sexual assault perpetrated by a variety of individuals in a representative sample of non-institutionalized women | N = 5,326 women, prevalence of some type of disability 26% | Cross-sectional; structured quantitative survey; telephone | Women with disabilities vs. women without disabilities |
| McFarlane et al., 2001 [141] | USA [AMRO] | To determine the frequency, type, and perpetrator of abuse toward women with physical disabilities. | N = 511 women, 100% with physical disability | Cross-sectional; structured quantitative survey; face to face | No comparison |
| Mitra et al., 2012 [103] | USA [AMRO] | To describe the prevalence of physical abuse before and during pregnancy among a representative sample of Massachusetts women with and without disabilities. | N = 2,876 all women, 4.9% disability | Cross-sectional; structured quantitative survey; not specified | Women with disabilities vs. women without disabilities |

(*Continued*)

**Table 2.** (Continued)

| Article | Country and region | Research question(s) | Sample–N, % women, % with disability | Methods–study design, data collection methods | Comparison of violence experiences–men vs. women; women with disabilities vs. women without disabilities |
|---|---|---|---|---|---|
| Nosek et al., 2006 [142] | USA [AMRO] | 1. To explore the demographic, disability, and psychosocial characteristics that distinguish women with disabilities who have experienced abuse within the past 12 months from those who have not<br>2. To what extent is a model including demographic, disability, and psychosocial variables sensitive in identifying abuse among women with disabilities? | N = 511 women; 100% with disability | Cross-sectional; structured quantitative survey; face to face | No comparison |
| Rees et al., 2011 [170] | Australia [WPRO] | To assess the association of gender-based violence and mental disorder, its severity and comorbidity, and psychosocial functioning among women. | N = 4451 women; prevalence of disability not reported | Cross-sectional; structured quantitative survey; face to face computer assisted | No comparison |
| Smith et al., 2008 [171] | USA [AMRO] | 1. To examine intimate partner sexual and physical abuse experienced by women with disabilities compared to women without disabilities and men with and without disabilities<br>2. To test the hypotheses that women with disabilities experience physical and sexual violence at a significantly higher rate than women without disabilities and men with and without disabilities<br>3. To test the hypothesis that being a woman with a disability increases the likelihood that a person will experience interpersonal violence. | N = 356,112, 219,911 women. n = 49,756 of the women identified themselves as having an activity limitation or disability. | Cross-sectional; structured quantitative survey; phone interview | Women with disabilities vs. women without disabilities; and men and women with disabilities |
| Sumilo et al., 2012 [138] | UK [EURO] | To explore prevalence of disability in women giving birth in the UK as measured by the presence of a limiting longstanding illness | N = 18,231 women, 9.4% reported limiting longstanding illness | Cross-sectional; structured quantitative survey; face to face | Women with disabilities vs. women without disabilities |
| Ward et al., 2010 [172] | USA [AMRO] | 1. To explore dating and romantic relationships among these adults<br>2. To identify the nature and extent of interpersonal violence in their relationships. | N = 47, 22 women. All with developmental disabilities | Cross-sectional; structured quantitative survey; face to face | Men with disabilities and women with disabilities |
| Yoshida et al., 2011 [68] | Canada [AMRO] | To examine victimization data from a Canadian survey of 1,095 women with disabilities to determine: i) who experienced abuse, ii) the forms of abuse, and iii) the factors associated with abuse. | N = 1095, all women, all with disabilities | Cross-sectional; structured quantitative survey; self-administered, mailed in | No comparison |

(*Continued*)

**Table 2.** (Continued)

| Article | Country and region | Research question(s) | Sample–N, % women, % with disability | Methods–study design, data collection methods | Comparison of violence experiences–men vs. women; women with disabilities vs. women without disabilities |
|---|---|---|---|---|---|
| Young et al., 1997 [104] | USA [AMRO] | 1. To examine if significantly more women with physical disabilities experience emotional, physical, or sexual abuse than women without physical disabilities<br>2. To explore if significantly more women with physical disabilities experience abuse by certain categories of perpetrators after onset of disability than women without physical disabilities<br>3. To explore if for women who experience abuse that lasts longer than a single incident, does duration of abuse differ significantly for women with and without physical disabilities? | N = 860 women, n = 439 with disabilities | Cross-sectional–case control; structured quantitative survey; all options possibilities—each woman with a disability was offered her choice of hard copy, computerized, or audio cassette versions of the survey, or the option to complete the survey over the telephone with one of the project staff, in order to permit women with severe disabilities to complete the survey privately without assistance from family or attendants. | Women with disabilities vs. women without disabilities |
| Scolese et al., 2020 [105] | Democratic Republic of Congo [AFRO] | To quantitatively explore IPV, including exploration of age, disability status, and the interaction of age and disability | N = 98 women; n = 74 reported mild or severe disability | Cross-sectional; structured quantitative survey; face to face interviews, recorded with personal digital assistant tablet | Women with disabilities vs. women without disabilities |
| Leskosek et al., 2013 [146] | Slovenia [EURO] | To calculate the incidence and prevalence of violence against women in Slovenia | N = 752 women; prevalence of disability not reported | Cross-sectional; structured quantitative survey; self-administered, mailed in | No comparison |
| Grossi et al., 2018 [47] | Brazil [AMRO] | To compare prevalence of physical and sexual abuse in women with temporomandibular disorders [TMD] to women without [TMD] | N = 80 women; n = 40 with TMD, 40 without TMD | Cross-sectional–case-control; structured quantitative survey; face to face, medical exam and dental exam | Women with disabilities vs. women without disabilities |
| Li et al., 2000 [76] | USA [AMRO] | 1. To examine the relationships between victimization as a result of violence, substance abuse, disability, and gender.<br>2. To test the hypothesis that women with disabilities will be more likely to be victims of substance abuse-related violence than will their male counterparts<br>3. To test the hypothesis that status of victimization from violence will vary across disability conditions for both men and women<br>4. To test the hypothesis that those victims of substance abuse-related violence will be more likely to report their own substance abuse than will people with disabilities who have not been victims of substance abuse-related violence. | N = 1,876, 48% women; all with disabilities | Cross-sectional; structured quantitative survey; self-administered (mailed in), self-administered (in-person) and face to face | Men with disabilities and women with disabilities |

**Table 2.** (Continued)

| Article | Country and region | Research question(s) | Sample–N, % women, % with disability | Methods–study design, data collection methods | Comparison of violence experiences–men vs. women; women with disabilities vs. women without disabilities |
|---|---|---|---|---|---|
| Zilkens et al., 2017 [173] | Australia [WPRO] | 1. To describe the frequency and severity of general body injury in women alleging recent sexual assault<br>2. To identify demographic and assault characteristics associated with injury severity | N = 1163 women; prevalence of disability not reported | Cross-sectional; structured interview and forensic exam; face to face | No comparison |
| Gunduz et al., 2019 [48] | Turkey [EURO] | 1. To compare the prevalence of IPV and comorbid psychiatric disorders among patients with fibromyalgia syndrome (FMS) and healthy controls<br>2. To investigate the relationship of intimate partner violence with psychiatric disorders and severity of pain in FMS patients. | N = 136 women; n = 68 with FMS, 68 without FMS | Cross-sectional–case control; structured quantitative interview; face to face | Women with disabilities vs. women without disabilities |
| Leserman et al., 1998 [106] | USA [AMRO] | 1. To test hypothesis that severity of sexual and physical abuse history, lifetime losses and traumas, turmoil in childhood family, and recent stressful life events will be related to current health status<br>2. To examine the interrelationships among these various stress measures, and determine whether social support acts as a buffer to decrease the negative impact of such stressors on health. | N = 239 women; prevalence of disability not reported | Cross-sectional; structured quantitative interview; self-administered and face-to-face (administered by interviewers) | Women with disabilities vs. women without disabilities |
| Martin et al., 2008 [107] | USA [AMRO] | 1. To examine the prevalence of physical and/or sexual violence experienced by North Carolina women during adulthood<br>2. To examine the socio-demographic characteristics of the women who have and have not experienced physical and/or sexual violence during adulthood<br>3. To examine the potential associations between women's experiences of physical and/or sexual violence during adulthood and their current physical health, mental health, and functional status | N = 9,830, all women; prevalence of functional impairment not reported | Cross-sectional; structured quantitative interview; telephone | Women with disabilities vs. women without disabilities |
| Cascardi et al., 1996 [84] | USA [AMRO] | What is the self-reported rate of, perceptions about, and symptoms associated with violence against newly admitted psychiatric inpatients by partners and family members in the past year? | N = 69, 51.5% women, 100% with psychiatric disability | Cross-sectional; structured quantitative interview; face-to-face | No comparisons |
| Chapple et al., 2004 [174] | Australia [WPRO] | 1. What is the level of victimisation amongst participants in a prevalence study of psychosis?<br>2. What are the demographic and clinical correlates of victimization? | N = 962, 387 women; 100% with disability | Cross-sectional; structured quantitative interview; face-to-face | Men with disabilities and women with disabilities |

(*Continued*)

**Table 2.** (Continued)

| Article | Country and region | Research question(s) | Sample–N, % women, % with disability | Methods–study design, data collection methods | Comparison of violence experiences–men vs. women; women with disabilities vs. women without disabilities |
|---|---|---|---|---|---|
| Goodman et al., 2001 [80] | USA [AMRO] | What is the prevalence and correlates of past year physical and sexual assault among a large sample of women and men with severe mental illness (SMI) drawn from inpatient and outpatient settings across 4 states? | N = 782, 321 women; 100% with disability | Cross-sectional; structured quantitative interview; face-to-face | Men with disabilities and women with disabilities |
| Hodgins et al., 2007 [81] | UK [EURO] | What is the prevalence of aggressive behaviour, victimisation and aggressive behaviour, victimisation and criminality among people receiving inpatient treatment for severe mental inpatient treatment illness in an inner-city area? | N = 205, 85 women; 100% with disability | Cross-sectional; structured clinical interview; face-to-face | Men with disabilities and women with disabilities |
| Teplin et al., 2005 [82] | USA [AMRO] | To determine the prevalence and incidence of crime victimization among persons with SMI by sex, race/ethnicity, and age, and to compare rates with general population data (the National Crime Victimization Survey), controlling for income and demographic differences between the samples | N = 936, 453 women; 100% with disability | Cross-sectional; structured quantitative interview; face-to-face | Men with disabilities and women with disabilities |
| Walsh et al., 2003 [83] | UK [EURO] | 1. To establish the 1-year prevalence of violent victimisation in community dwelling patients with psychosis 2. To identify the socio-demographic and clinical correlates of violent victimisation. | N = 691, 294 women; 100% with disability | Cross-sectional; structured quantitative interview; face-to-face | Men with disabilities and women with disabilities |
| Nosek et al., 2001 [160] | USA [AMRO] | What types of abuse experienced by women with physical disabilities are directly related to their disability? | N = 946 women; n = 504 women with disabilities | Cross-sectional; structured quantitative interview; face to face or mailed self-administered questionnaire | Women with disabilities vs. women without disabilities |
| Majeed-Ariss et al., 2020 [122] | UK [EURO] | 1. What is the prevalence of learning disabilities amongst adult clients attending Saint Mary's Sexual Assault Referral Center for a forensic medical examination? 2. What are the demographics of clients with learning disabilities as compared to clients without? 3. Are there similarities/ differences in the medical history of clients with learning disabilities as compared to clients without? 4. Are there similarities/ differences in the context of the sexual assault for clients with learning disabilities as compared to clients without? | N = 679; 634 women. n = 56 total with learning disability, 50 of those women | Cross-sectional; Questionnaire and forensic medical examination; questionnaire self-administered | Men and women with disability vs. men and women without disability |

(*Continued*)

**Table 2.** (Continued)

| Article | Country and region | Research question(s) | Sample–N, % women, % with disability | Methods–study design, data collection methods | Comparison of violence experiences–men vs. women; women with disabilities vs. women without disabilities |
|---|---|---|---|---|---|
| Acharya 2019 [147] | Mexico [AMRO] | To explore the prevalence of violence against them and its implications on physical injuries and disabilities | N = 68; 100% women, single prevalence of disability not reported | Cross-sectional; open-ended study guide; face-to-face | No comparisons |
| Akyazi et al., 2018 [175] | Turkey [EURO] | To explore the relationship between mental disorders, childhood trauma and sociodemographic characteristics was evaluated in women staying in shelters due to domestic violence | N = 59; 100% women, 76.3% diagnosed with at least one psychiatric disorder | Cross-sectional; structured clinical interview; face-to-face | Women with disabilities vs. women without disabilities |
| Basile et al., 2016 [93] | USA [AMRO] | To examine the relative prevalence of recent (past 12 months) penetrative and non-penetrative sexual violence comparing men and women with and without a disability | N = 16507 total; 9086 women, 3847 total with disability, 2286 women with disability | Cross-sectional; structured quantitative interview; telephone | Women with disabilities vs. women without disabilities |
| Breiding et al., 2015 [94] | USA [AMRO] | To examine the link between disability and IPV in a nationally representative sample of U.S. women and men | N = 16507 total; 9086 women, n = 3847 total with disability, 2286 women with disability | Cross-sectional; structured quantitative interview; telephone | Women with disabilities vs. women without disabilities; Women with disability vs. men with disability |
| De Waal et al., 2017 [176] | Netherlands [EURO] | To identify factors associated with violent and property victimization in male and female dual diagnosis patients in order to identify targets for prevention | N = 243; 72 women. All with psychiatric dual diagnosis | Cross-sectional; structured quantitative interview; face-to-face | Women with disability vs. men with disability |
| Del rio Ferres et al., 2013 [135] | Spain [EURO] | 1. To determine the prevalence of violence and its possible relations with socio-economic, socio-demographic and disability-related factors. 2. To determine the annual and lifetime prevalence of abuse in two groups of women with physical and visual disabilities 3. To explore the possible role of level of education, family and job status, dependence on a caregiver and financial status as risk or vulnerability factors for violence: 4. To analyze the specific impacts of violence on the well-being and health of women with disabilities | N = 96; all women, all with some type of disability | Cross-sectional; structured quantitative interview; face-to-face | No comparisons |
| Jonas et al., 2013 [177] | UK [EURO] | To assess the extent to which being a victim of IPV is associated with psychiatric disorders in men and women | N = 7047; 3850 women; single baseline prevalence of disability not reported | Cross-sectional; structured quantitative interview; CAPI and CASI (for violence-related questions) | Women with disability vs. men with disability |
| Lacey et al., 2016 [116] | USA [AMRO] | 1. To examine the mental and physical health of U.S. Caribbean Black women with and without a history of severe partner IPV 2. To explore the role of generational status—first, second, or third—in association with the physical and mental health of abused Caribbean Black women | N = 6082; all women; single baseline prevalence of disability not reported | Cross-sectional; structured quantitative interview; majority of interviews conducted face-to-face with computer-assisted instrument; smaller proportion (14%) conducted by phone | Women with disabilities vs. women without disabilities |

*(Continued)*

**Table 2.** (Continued)

| Article | Country and region | Research question(s) | Sample–N, % women, % with disability | Methods–study design, data collection methods | Comparison of violence experiences–men vs. women; women with disabilities vs. women without disabilities |
|---|---|---|---|---|---|
| Le et al., 2015 [178] | Vietnam [SEARO] | To establish the prevalence of lifetime exposure to poly-victimisation and demographic characteristics of victims among high school students in Vietnam | N = 1616 total; 729 female; % of women with disability not reported | Cross-sectional; structured questionnaire; self-administered | No comparisons, disability addressed as a variable |
| Macdowall et al., 2013 [123] | UK [EURO] | 1. To determine the prevalence of attempted and completed non-volitional sex in women and men since the age of 13 years 2. To determine the associations between ever having experienced completed non-volitional sex and several socio-demographic, behavioural, and health factors | N = 15162, 8869 women; % of women with disability not reported | Cross-sectional; structured quantitative interview; CAPI and CASI (for violence-related questions) | Women with disabilities vs. women without disabilities |
| New, 2019 [59] | Australia [WPRO] | To determine the prevalence and impact of sexual abuse in people with spinal cord damage | N = 356 total (136 people with spinal cord damage; 220 controls); n = 211 women total | Cross-sectional; case-control, structured questionnaire; self-administered | Men and women with disabilities vs. men and women without disabilities |
| Olofsson et al., 2015 [95] | Sweden [EURO] | To investigate the prevalence and risk for exposure to physical violence, psychological offence, or threats of violence in men and women with physical and/or sensory disabilities, compared to men and women with no such disabilities | N = 47,006; 25,461 women; single baseline % of women with disability not reported | Cross-sectional; structured quantitative survey; self-administered | Women with disabilities vs. women without disabilities; Men and women with disability vs. men and women without disability |
| Salahi et al., 2018 [152] | Iran [EMRO] | To assess the accuracy and psychometric characteristics of Women Abuse Screening Tool [WAST] and its 2-item short form (WAST-SF) for identifying IPV compared to the revised conflict tactics scale (CTS-2) | N = 400; all women; all with mental disorders | Cross-sectional; structured questionnaire; face-to-face | No comparison |
| Owens 2007 [179] | USA [AMRO] | To explore the relationship between the perception by female abuse victims of patient-centeredness in health care providers in a psychiatric emergency setting and the resulting frequency of disclosure of IPV history. | N = 216; all women; all with psychiatric disorder | Cross-sectional; structured quantitative survey; self-administered | No comparisons |
| Coker et al., 2002 [180] | USA [AMRO] | To estimate IPV prevalence by type (physical, sexual, and psychological) and associated physical and mental health consequences among women and men. | N = 13912; 6790 women; n = 518 women with chronic disease that occurred after first IPV or age 25 | Cross-sectional; structured quantitative questionnaire; telephone | Men vs. women; women with disabilities vs. women without disabilities |
| Dammeyer et al., 2018 [181] | Denmark [EURO] | To quantify levels of violence and discrimination among people with disabilities and analyze the effects of gender and the type and degree of disability. | N = 18,019; 53.3% women; n = 4519 (of the total) reported a physical disability and 1398 reported a mental disability | Cross-sectional; structured quantitative questionnaire; self-administered questionnaire online (81%); phone interview (19%) | Men and women with disabilities vs. men and women without disabilities; women with disabilities vs. women without disabilities |

(*Continued*)

**Table 2.** (*Continued*)

| Article | Country and region | Research question(s) | Sample–N, % women, % with disability | Methods–study design, data collection methods | Comparison of violence experiences–men vs. women; women with disabilities vs. women without disabilities |
|---------|-------------------|---------------------|-------------------------------------|----------------------------------------------|----------------------------------------------------------------------------------------------------------|
| Gibbs et al., 2017 [154] | South Africa [AFRO] | 1. To assess the prevalence and factors associated with recent IPV amongst post-partum women in one clinic in eThekwini Municipality, South Africa 2. To explore the relationship between IPV, depression and functional limitations/disabilities | N = 275; all women; single baseline % of women with disability not reported | Cross-sectional; structured quantitative questionnaire; face-to-face | Women with disabilities vs. women without disabilities |
| Khalifeh et al., 2015 [49] | UK [EURO] | To assess the prevalence and impact of crime among people with several mental illness compared with the general population | N = 3499; 1851 women total; n = 361 psychiatric patients, of which 158 women | Cross-sectional; case-control; structured quantitative questionnaire; cases interviewed face-to-face; controls drawn from survey with face-to-face and self-completion module | Men and women with disabilities vs. men and women without disabilities; women with disabilities vs. women without disabilities |
| Milberger et al., 2003 [153] | USA [AMRO] | 1. What is the prevalence of violence among a sample of women with physical disabilities? 2. What risk factors for violence exist among women with physical disabilities? 3. What types of actions do women with physical disabilities engage in to escape abusive situations? | N = 177; all women; all with disabilities | Cross-sectional; structured quantitative questionnaire; choice of phone, on-line or by mail for screening and choice to face-to-face, phone, on-line or by mail for in-depth interview | No comparisons |
| Nannini 2006 [118] | USA [AMRO] | 1. How do sexual assault patterns differ for women with disabilities as compared with women without disabilities and 2. How do patterns differ among women with different disabilities? | N = 16,672, all women; single baseline % of women with disability not reported | Cross-sectional; structured quantitative survey; face-to-face | Women with disabilities vs. women without disabilities |
| Weiner et al., 2013 [50] | USA [AMRO] | To examine perspectives among deaf and hard of hearing students in residential and large day schools regarding bullying, and compare these perspectives with those of a national database of hearing students | N = 812; 392 female, all with disability  Full sample characteristics of control comparison group not included | Cross-sectional; case-control; self-administered | Women with disabilities vs. women without disabilities |
| Nunes de Oliviera et al., 2013 [77] | Brazil [AMRO] | To assess factors associated with lifetime physical violence against patients with mental illness stratified by sex in Brazil | N = 2475; 1,277 (51.6%) were women; all utilizing health services for psychiatric disorder | Cross-sectional; structured quantitative survey; face-to-face | No comparisons |
| Ferraro et al., 2017 [117] | Brazil [AMRO] | To determine if there is a measurable association between combined psychosocial factors, specifically domestic violence and mental disorders, and birth outcomes, specifically birth nutritional status and preterm delivery | N = 775; all women; N = 296 with disability | Cross-sectional; structured quantitative survey; face-to-face | Women with disabilities vs. women without disabilities |
| Gilchrist et al., 2012 [182] | Spain [EURO] | To examine the relationship between intimate partner violence, childhood abuse and psychiatric disorders among 118 female drug users in treatment in Barcelona, Spain | N = 118; all women; single baseline % of women with disability not reported | Cross-sectional; structured quantitative survey; face-to-face | Women with disabilities vs. women without disabilities |

(*Continued*)

**Table 2.** (Continued)

| Article | Country and region | Research question(s) | Sample–N, % women, % with disability | Methods–study design, data collection methods | Comparison of violence experiences–men vs. women; women with disabilities vs. women without disabilities |
|---------|--------------------|--------------------|------------------------------------|-----------------------------------------------|---------------------------------------------------------------------------------------------------------|
| Golding 1996 [108] | USA [AMRO] | To evaluate the functional impact of sexual assault history in two general population survey | N = 6024; 52.9% women; single baseline % of women with disability not reported | Cross-sectional; structured clinical survey; face-to-face | Women with disabilities vs. women without disabilities |
| Siqueira-Campos et al., 2019 [51] | Brazil [AMRO] | To investigate the prevalence of anxiety, depression and mixed anxiety and depressive disorder and factors associated with these conditions in women with chronic pelvic pain compared to a pain-free control group | N = 200; all women; n = 100 with chronic pelvic pain | Cross-sectional case-control study; structured quantitative survey; self-administered | Women with disabilities vs. women without disabilities |
| Sturup et al., 2011 [52] | Sweden [EURO] | To report the rate of violent victimisation of Psychiatric patients 1 year before interview and to examine the relative rate in comparison to the general population | N = 1560; 812 women; n = 203 with psychiatric diagnosis | Cross-sectional case-control study; structured quantitative interview and medical records; face-to-face | Women with disabilities vs. women without disabilities |
| Thompson et al., 2019 [53] | Australia [WPRO] | To investigate aspects of the sexuality and sexual health among female youth (aged 15 to 24 years old) with bi-polar disorder [BPD] pathology (including sub-syndromal BPD, ie, 3 to 9 criteria), and to compare this with a matched healthy population sample | N = 254; all women; n = 50 with BPD diagnosis | Cross-sectional case-control study; Structured diagnostic interview for patient group and comprehensive quantitative survey for both groups; face-to-face and CAPI phone interview | Women with disabilities vs. women without disabilities |
| Walker et al., 1997 [54] | USA [AMRO] | 1. To test the hypothesis that, compared with a group of women with rheumatoid arthritis (RA), women with fibromyalgia would have significantly higher lifetime prevalence rates of adult sexual and physical assault, as well as higher rates and severity of childhood sexual, physical, and emotional abuse and neglect. 2. To test the hypothesis that the association with lifetime victimization would not be limited specifically to sexual and physical abuse alone, but would be associated with a more general lifetime history of distressing interpersonal trauma. 3. To test the hypothesis that levels of dissociation and functional disability would be significantly higher in the patients with FM | N = 69; all women; 36 with fibromyalgia and 33 with rheumatoid arthritis | Cross-sectional case-control study; | Women with one type of disability vs. women with another type of disability |
| Afe et al., 2017 [183] | Nigeria [AFRO] | To explore socio- demographic and socio-economic characteristics of IPV victims with schizophrenia and partners | N = 79; all women; all with disability | Cross-sectional; structured quantitative survey and clinical interview; face-to-face | No comparison |

(*Continued*)

**Table 2.** (Continued)

| Article | Country and region | Research question(s) | Sample–N, % women, % with disability | Methods–study design, data collection methods | Comparison of violence experiences–men vs. women; women with disabilities vs. women without disabilities |
|---|---|---|---|---|---|
| Anderson et al., 2016 [78] | UK [EURO] | 1. To investigate the association between childhood maltreatment and adulthood domestic and sexual violence victimisation among people with severe mental illness (SMI) 2. To explore this association in terms of gender differences and potential mediators | N = 318; 137 women; all with mental health diagnoses | Cross-sectional; structured quantitative survey and clinical interview; face-to-face component and self-administered component | Men with disability vs. women with disability |
| Beydoun et al., 2017 [127] | USA [AMRO] | To examine associations of physical intimate partner violence (PIPV) with selected mental health disorders using a nationally representative sample of emergency department discharges | N reported is number of discharges from emergency department | Cross-sectional; structured quantitative survey; face-to-face | Women with disability vs. women without disability |
| Du Mont et al., 2014 [184] | Canada [AMRO] | To examine the risk of IPV among a representative sample of non-institutionalized women with activity limitations due to a mental health condition | N = 6851; all women of whom 322 (4.7%) reported a mental health-related activity limitation always/often or sometimes | Cross-sectional; structured quantitative survey; phone interview | Women with disability vs. women without disability |
| Gil-Llario et al., 2019 [85] | Spain [EURO] | 1. To determine the prevalence of self-reported and documented sexual abuse in people with mild or moderate intellectual disability 2. To analyse the sequelae that such experiences can have on their psychosocial health | N = 360; 180 women; all with some intellectual disability | Cross-sectional study; structured quantitative survey; face-to-face or self-administered, depending on results of a reading test | No comparison |
| Gold et al., 2012 [136] | USA [AMRO] | To compare victims of suicides during pregnancy, suicides up to 1 year postpartum and non-pregnancy-associated suicides and to compare psychiatric history, substance use, methods of suicide, intimate partner problems and precipitating circumstances among these groups | N = 2083; all women; 56% with mental health disorder diagnosis | Retrospective analysis of national violence death reporting system; Review of death certificates, coroner and medical examiner information, toxicology data and law enforcement reports | No comparison |
| Gonzalez Cases et al., 2014 [69] | Spain [EURO] | To examine the prevalence and characteristics of IPV towards women with a severe mental illness (SMI). | N = 142; all women; all with disability | Cross-sectional; structured quantitative survey; face-to-face | No comparison |
| Helfrich et al., 2008 [137] | USA [AMRO] | To investigate the presence of mental health symptoms and disorders reported by 74 women in a domestic violence shelter and the impact of those symptoms on function in work, school, and social encounters | N = 74; all women; (compared with national sample of N = 65,096,167; single baseline % of women with disability not reported | Cross-sectional; structured quantitative survey; face-to-face | No comparison |
| Khalifeh et al., 2015 [55] | UK [EURO] | To compare the prevalence and impact of violence against SMI patients and the general population | N = 303 psychiatric patients compared to 22606 controls drawn from national survey. Of the 303, 43.9% women; of the controls, 54.4% women | Cross-sectional case control; Structured quantitative survey; review of clinical records; CAPI face to face; opt-in CASI questionnaire which focused on more sensitive topics | Women with disability vs. women without disability |

(*Continued*)

**Table 2.** (Continued)

| Article | Country and region | Research question(s) | Sample–N, % women, % with disability | Methods–study design, data collection methods | Comparison of violence experiences–men vs. women; women with disabilities vs. women without disabilities |
|---|---|---|---|---|---|
| Kmett et al., 2018 [86] | USA [AMRO] | To examine the prevalence rate and characterize the nature of sexual assault among individuals with severe mental illness who were under psychiatric care in three different inpatient facilities | N = 1,136; 455 women; all with disability | Cross-sectional survey Structured quantitative survey and diagnostic interview; face-to-face and chart review | Men with disability vs. women with disability |
| Lundberg et al., 2015 [56] | Uganda [AFRO] | 1. To investigate prevalence of past-year sexual risk behavior and sexual violence exposure in persons with severe mental illness (SMI) in Uganda, and compared results to general population estimates. 2. To investigate whether persons with SMI reporting sexual risk behavior and sexual violence exposure were more likely to be HIV-infected. | N = 602 persons consecutively discharged from psychiatric ward, compared to 9211 from DHS; women: SMI (n = 343) and general population (n = 7413) | Cross-sectional case-control; structured quantitative survey; face-to-face | Women with disability vs. women without disability |
| McPherson et al., 2007 [43] | USA [AMRO] | To investigate the demographic and clinical correlates of intimate partner violence in a sample of 324 mothers with severe mental illness | N = 324; all women; all with disability | Longitudinal study; structured quantitative survey; face-to-face | No comparison |
| Meekers et al., 2013 [57] | Bolivia [AMRO] | To examine the relationship between Bolivian women's experiences with physical, psychological, and sexual intimate partner violence and mental health outcomes | N = 10,119; all women; single baseline % of women with disability not reported | Cross-sectional case-control; structured quantitative survey; self-administered | No comparison |
| Nguyen et al., 2017 [58] | Australia [WPRO] | 1. To explore the sexual and reproductive trends and behaviors in women who attended community mental health clinics in Western Australia 2. To assess the self-reported rate of sexual assault, sexual health seeking behaviors such as engagement with Pap smear, seeking advice for contraception with a general practitioner (GP) and sexually transmitted infection (STI) screening, as well as the rate of unplanned pregnancies and lifestyle factors. 3. To explore the relationship between sexual trauma and sexual health seeking behaviors in this vulnerable group of women | N = 220; all women; all with disability | Cross-sectional case-control; structured quantitative survey; self-administered | No comparison |

(*Continued*)

**Table 2.** (Continued)

| Article | Country and region | Research question(s) | Sample–N, % women, % with disability | Methods–study design, data collection methods | Comparison of violence experiences–men vs. women; women with disabilities vs. women without disabilities |
|---|---|---|---|---|---|
| Racic et al., 2006 [185] | Bosnia and Herzegovina [EURO] | 1. What is the prevalence of mistreatment among elderly patients suffering from mental disorders? 2. What types of elder mistreatment are present? 3. Could the Hwalek-Sengstock Elder Abuse Screening Test be used as a screening tool in primary care settings? 4. What are the contributing factors for mistreatment? | N = 184; 112 women; all with disability | Cross-sectional; Structured quantitative survey; Self-administered; and then following, face to face | No comparison |
| Riley et al., 2014 [186] | USA [AMRO] | What are the associations between co-occurring psychiatric conditions and violence against homeless and unstably housed women | N = 291; all women; 97% screened positive for 1 or more psychiatric disorder | Cross-sectional; Structured quantitative survey; face-to-face and CAPI | No comparison |
| Santaularia et al., 2014 [109] | USA [AMRO] | To identify associations between sexual violence and health risk behaviors, chronic health conditions and mental health conditions utilizing population-based data in Kansas. | N = 4886; all women; single baseline % of women with disability not reported | Cross-sectional; Structured quantitative survey; telephone interview | Women with disability vs. women without disability |
| Schofield et al., 2013 [187] | Australia [WPRO] | To determine whether elder abuse can predict mortality and disability over the ensuing 12 years | N = 12066; all women; single baseline % of women with disability not reported | Prospective cohort study; structured quantitative survey; self-administered | No comparison |
| Shah et al., 2018 [188] | USA [AMRO] | To investigate the association between IPV and psychotic experiences in U.S. cities | 1615 participants in four US cities; women—932 (57.7%); single baseline % of women with disability not reported | Cross-sectional; Structured quantitative survey; face-to-face | Women with disability vs. women without disability vs. men with disability vs. men without disability |
| Anderson et al., 1993 [124] | USA [AMRO] | To identify dissociative experiences and disorders among women who are survivors of sexual abuse | N = 1615; 932 women; single baseline % of women with disability not reported | Cross-sectional study; structured quantitative survey and diagnostic interview; face-to-face | No comparison |
| Institute of Statistics, et al., 2018 [129] | Albania [EUR] | To provide estimates of basic socio-demographic and health indicators for the country as a whole and the twelve prefectures | N = 10,970 women age 15–49, 4,030 women age 50–59 in the overall sample; 3% of women aged 15–59 reported chronic disability; DV module asked to 10,970 women aged 15–49 | Cross-sectional study; structured quantitative survey; CAPI | No comparison |
| National Institute of Statistics et al., 2015 [189] | Cambodia [WPRO] | To provide up-to-date estimates of basic demographic and health indicators | N = 17,578 women and 5,190 men in the overall sample; % of women above 15 with disability not reported; n = 4,307 women asked the DV module | Cross-sectional study; structured quantitative survey; face-to-face | No comparison |
| Institut National de la Statistique et al., 2012 [190] | Cameroon [AFRO] | To provide up-to-date estimates of basic demographic and health indicators | 15,426 women and 7,191 men; n = 5043 women participated in DV module | Cross-sectional study; structured quantitative survey; face-to-face | No comparison |
| Institut National de la Statistique et al., 2015 [191] | Chad [AFRO] | To provide up-to-date estimates of basic demographic and health indicators | 17,719 women, 5,248 men in overall sample; n = 4283 women participated in DV module | Cross-sectional study; structured quantitative survey; face-to-face | No comparison |

*(Continued)*

Table 2. (Continued)

| Article | Country and region | Research question(s) | Sample–N, % women, % with disability | Methods–study design, data collection methods | Comparison of violence experiences–men vs. women; women with disabilities vs. women without disabilities |
|---|---|---|---|---|---|
| Institut Haïtien de l'Enfance (IHE) et al., 2018 [192] | Haiti [AMRO] | To provide up-to-date estimates of basic demographic and health indicators | Women aged 15–49: 14,371; Women aged 50–64: 1,142; Men aged 15–64: 9,795; n = 3816 women participated in DV module | Cross-sectional study; structured quantitative survey; CAPI | No comparison |
| Institut National de la Statistique (INSTAT) et al., 2019 [193] | Mali [AFRO] | To provide up-to-date estimates of basic demographic and health indicators | 10,519 women and 4,618 men in overall sample; n = 3784 women participated in DV module | Cross-sectional study; structured quantitative survey; face-to-face | No comparison |
| Agence Nationale de la Statistique et de la Démographie (ANSD) et al., 2019 [194] | Senegal [AFRO] | To provide up-to-date estimates of basic demographic and health indicators | 8,649 women and 3,365 men in overall sample; n women participating in DV module not reported | Cross-sectional study; structured quantitative survey; face-to-face | No comparison |
| National Department of Health et al., 2016 [195] | South Africa [AFRO] | To provide up -to -date estimates of basic demographic and health indicators | 8514 women and 3618 men in overall sample; n = 5865 women participated in DV module | Cross-sectional study; structured quantitative survey; face-to-face | No comparison |
| Uganda Bureau of Statistics et al., 2018 [196] | Uganda [AFRO] | To provide up-to-date estimates of basic demographic and health indicators | 18,506 women and 5,336 men in overall sample; n = 9232 women participated in DV module | Cross-sectional study; structured quantitative survey; face-to-face | No comparison |
| General Directorate of Statistics et al., 2018 [197] | Timor-Leste [SEARO] | To provide up-to-date estimates of basic demographic and health indicators | 12,607 women; 4,622 men in overall sample; n = 5122 women participated in DV module | Cross-sectional study; structured quantitative survey; CAPI | No comparison |
| The Gambia Bureau of Statistics et al., 2014 [198] | The Gambia [AFRO] | To provide up-to-date estimates of basic demographic and health indicators | 10,233 women and 3,821 men in overall sample; n = 4525 women participated in DV module | Cross-sectional study; structured quantitative survey; face-to-face | No comparison |
| National Institute of Population Studies et al., 2019 [199] | Pakistan [EMRO] | To provide up-to-date estimates of basic demographic and health indicators | 12,364 women and 3145 men; n = 3303 women participated in DV module | Cross-sectional study; structured quantitative survey; face-to-face | No comparison |
| Rand et al., 2007 [200] | USA [AMRO] | To generate first estimates of crime against people with disabilities measured by the National Crime Victimization Survey, administered by the Bureau of Justice Statistics | Not included | Cross-sectional study; structured quantitative survey; mode of interview not full described– 70% of surveys in full National Crime Victimization Survey were by telephone | Men with disabilities vs. women with disabilities (by type of disability); women with disabilities vs. women without disabilities |
| Carlile 1991 [70] | Canada [AMRO] | 1. To assess the prevalence of spousal violence in married, female, psychiatric patients 2. To determine whether or not there was a relationship between such violence and admission to a psychiatric unit | N = 152; all women; all in psychiatric care (in-patient or outpatient) | Cross-sectional study; structured quantitative survey; face-to-face interview | Comparisons between types of psychiatric diagnoses |
| Post et al., 1980 [130] | USA [AMRO] | 1. To determine the prevalence of domestic violence in a population of psychiatric inpatients 2. To discover how the personality characteristics and life experiences of victims and perpetrators of domestic violence differed from each other and from patients without a history of domestic violence | N = 60; 38 women; all psychiatric in-patients | Cross-sectional study; structured quantitative survey; face-to-face interview | Men with disabilities vs. women with disabilities |

(*Continued*)

**Table 2.** (Continued)

| Article | Country and region | Research question(s) | Sample–N, % women, % with disability | Methods–study design, data collection methods | Comparison of violence experiences–men vs. women; women with disabilities vs. women without disabilities |
|---------|-------------------|---------------------|--------------------------------------|-----------------------------------------------|---------------------------------------------------------------------------------------------------------|
| Sansone et al., 2007 [71] | USA [AMRO] | 1. To explore the correlation between intimate partner violence and self-harm behavior. 2. Does physical mistreatment reported by victims correlate with histories of bodily self-harm behavior by victims? | N = 113; all women; all psychiatric in-patients | Cross-sectional study; structured quantitative survey; self-completed by hand | No comparison |
| Zanarini et al., 1999 [201] | USA [AMRO] | 1. To report on the prevalence of adult experiences of physical assault and/or rape in carefully diagnosed borderline patients (N = 290) and axis II comparison subjects (N = 72) 2. To examine the relationship between such experiences of violence and a variety of potential risk factors. | N = 362; % of women not stated; all with some psychiatric disability | Cross-sectional study; structured quantitative survey; face-to-face interview | Men with specific disability vs. women with specific disability (compared male and female borderline patients; and male and female borderline patients with male and female non-borderline patients); women with borderline vs. women with different psychiatric disability |
| Bengtsson-Tops et al., 2012 [79] | Sweden [EURO] | 1. To investigate the prevalence of self-reported adulthood and last-year victimization in male and female outpatients with psychosis 2. To investigate relationships to perpetrators, whether drugs or alcohol were involved in the victimization situation and places where victimization occurred. | N = 174; 99 women; all participants diagnosed with psychosis | Cross-sectional study; structured quantitative survey; face-to-face interview | Men with disability vs. women with disability |
| Ford 2008 [72] | USA [AMRO] | 1. To test the hypothesis that women with severe mental illness would be more likely to meet criteria for current PTSD or Disorders of Extreme Stress Not Otherwise Specified [DESNOS] if they reported past victimization (i.e., abuse, assault, violence—replicating prior studies) or other psychological traumas (i.e., loss, illness, accidents—extending prior studies) than if they did not report this. 2. To test the hypothesis that women with severe mental illness Would be most likely to report a history of exposure to psychological trauma and, independent of trauma history, to meet criteria for current PTSD or DESNOS if they (a) were of color (vs. White) or (b) had a poverty-level income (vs. low but not poverty-level income). | N = 38; all women; all with severe mental illness | Cross-sectional study; structured quantitative survey; face-to-face interview | No comparison |

(*Continued*)

**Table 2.** (Continued)

| Article | Country and region | Research question(s) | Sample–N, % women, % with disability | Methods–study design, data collection methods | Comparison of violence experiences–men vs. women; women with disabilities vs. women without disabilities |
|---|---|---|---|---|---|
| Friedman et al., 2011 [44] | USA [AMRO] | 1. To assess the levels of both victimization and IPV perpetration by Puerto Rican women with serious mental illness.<br>2. To explore the hypothesis that higher rates of perpetration would be detected among women with bipolar disorder (due to irritability) and schizophrenia (due to disorganization and psychosis) than women with major depression. | N = 53; all women, all with severe mental illness | Mixed methods; quantitative component–longitudinal; baseline survey and then survey each year for two years; structured quantitative survey; face-to-face | Comparison between different types of disability–major depression vs. bipolar vs. schizophrenia |
| Goodman et al., 1995 [202] | USA [AMRO] | To examine the prevalence of violent victimization amongst women with severe mental illness who are homeless | N = 99; all women; all with severe mental illness | Cross-sectional study; structured quantitative survey and semi-structured questions for sexual violence items; face-to-face interview | No comparison |
| Leithner et al., 2009 [203] | Austria [EURO] | 1. To assess the prevalence of physical, sexual, or psychological violence in a cohort of patients with gynecological symptoms who presented at a psychosomatic outpatient clinic<br>2. To assess differences in prevalence rates of gynecological symptoms and mental health problems in women with and without a history of experiencing violence. | N = 424; all women; all attending gynecological-psychosomatic outpatient clinic | Cross-sectional study; semi-structured interview face-to-face | No comparison |
| Lipschitz et al., 2009 [204] | USA [AMRO] | To examine rates and characteristics of childhood abuse and adult assaults in a large general outpatient population using a detailed self-report questionnaire | N = 120; 84 women; all psychiatric outpatients | Cross-sectional study; structured quantitative interview; self-completed | Men with disability vs. women with disability |
| Morgan et al., 2010 [205] | UK [EURO] | To establish prevalence of domestic violence among female psychiatric patients, including risk factors, health professional attention and acceptability of routine enquiry. | N = 70; all women; all psychiatric patients | Cross-sectional; structured quantitative instrument; self-completed | No comparison |

(*Continued*)

**Table 2.** (Continued)

| Article | Country and region | Research question(s) | Sample–N, % women, % with disability | Methods–study design, data collection methods | Comparison of violence experiences–men vs. women; women with disabilities vs. women without disabilities |
|---|---|---|---|---|---|
| Surrey et al., 1990 [206] | USA [AMRO] | 1. To determine the prevalence and effects of the reported history of physical and sexual abuse among women outpatients, and to assess factors associated with such a history.<br>2. To test the hypothesis that women outpatients would have high rates of reported abuse, but lower rates than would inpatients<br>3. To test the hypothesis that the severity of overall symptomatology would be greater for outpatients who reported a history of abuse than for those reporting no history of abuse, but would be lower among outpatients than among inpatients<br>4. To test the hypothesis that the patterning of symptoms and diagnosis could be used to identify patients with a reported history of abuse. | N = 140; all women; all psychiatric patients | Cross-sectional; structured quantitative instrument; self-completed | No comparison |
| Swett et al., 1991 [207] | USA [AMRO] | 1. To determine if there was an association between a history of physical and sexual abuse and alcoholic drinking behavior.<br>2. To explore the hypothesis that that there would be a relatively high rate of alcoholic drinking in this patient population and that those with a history of abuse would tend to drink more than the others | N = 188; all women; all psychiatric patients | Cross-sectional; structured quantitative instrument; self-completed | No comparison |
| Briere et al., 1997 [148] | USA [AMRO] | 1. To clarify potential connections between psychological or psychosocial difficulties, previous victimization experiences, and relevant demographic factors<br>2. To explore hypotheses including that adult victimization experiences would be associated with clinical outcome variables even after demographics and childhood victimization experiences were taken into account. | N = 93; all women; all psychiatric patients | Cross-sectional; structured quantitative instrument; face-to-face | No comparison |
| Bengtsson-Tops et al., 2005 [208] | Sweden [EURO] | To investigate self-reported physical, sexual, emotional and economical abuse in Swedish female users of psychiatric services, who the perpetrators were and in which places abuse occurred. | N = 1382; all women; all psychiatric patients | Cross-sectional study; structured quantitative survey; self-administered | No comparison |

(*Continued*)

**Table 2.** (Continued)

| Article | Country and region | Research question(s) | Sample–N, % women, % with disability | Methods–study design, data collection methods | Comparison of violence experiences–men vs. women; women with disabilities vs. women without disabilities |
|---------|--------------------|--------------------|----------------------------------|---------------------------------------------|----------------------------------------------------------------------------------------------------|
| Yellowlees et al., 1994 [149] | Australia [WPRO] | 1. To detail the prevalence of a series of life problems across a full range of psychiatric disorders.<br>2. To test the hypothesis that there is a greater prevalence of life problems in patients presenting with neurotic psychiatric disorders than in patients presenting with psychotic disorders<br>3. To test the hypothesis that patients presenting with personality disorders will have higher levels of life problems than patients presenting with DSM-III Axis 1 diagnoses<br>4. To test the hypothesis that psychiatric patients presenting with a DSM-III Axis 1 diagnosis other than adjustment disorders will have more life problems than "control" patients presenting with diagnoses of adjustment disorders or V codes, or who were not given a psychiatric diagnosis. | N = 707; % women not reported; all with psychiatric diagnosis | Cross-sectional study; structured quantitative survey; face-to-face | Men with specific disabilities vs. women with specific disabilities; men and women with specific disabilities vs. men and women with other disabilities (i.e. compares psychoses depression or mania vs. neuroses-depressive) |
| Coker et al., 2005 [110] | USA [AMRO] | 1. To estimate the frequency and type of disabilities preventing work among those experiencing IPV compared with those never experiencing IPV<br>2. To explore the association between IPV by type (physical, sexual, and psychological), timing (current or past), and disabilities preventing work as reported in a clinical population of women attending primary health clinics. | N = 1152; all women; overall baseline reported disability of 19.2% | Cross-sectional survey; structured quantitative instrument; 10 minute face-to-face interview to screen for IPV, followed by 30–45 minute telephone interview | No comparison |
| Brownlie et al., 2007 [45] | Canada [AMRO] | 1. To examine the prevalence of sexual assault by age 25 among participants with language impairment and participants with unimpaired language<br>2. To examine the relationship between language impairment and sexual assault, controlling for socio-economic status. | N = 268; Of these, 112 (44 women and 68 men) had language or speech impairment at age 5, and 132 (49 women and 83 men) were controls. | Longitudinal–study of children from age 5 followed to age 25; data represents interview at age 25; structured quantitative instrument; face-to-face | Women with disability vs. women without disability |
| Diaz-Olavarrieta et al., 1999 [209] | Mexico [AMRO] | 1.To determine the prevalence of domestic violence among female patients with chronic neurological disorders<br>2. To identify possible diagnoses associated with the domestic violence | N = 1000; all women; all chronic neurological disorders | Cross-sectional; structured quantitative survey; self-administered | Comparison of different types of neurological disorders (structural vs. functional) |

(*Continued*)

**Table 2.** (Continued)

| Article | Country and region | Research question(s) | Sample–N, % women, % with disability | Methods–study design, data collection methods | Comparison of violence experiences–men vs. women; women with disabilities vs. women without disabilities |
|---|---|---|---|---|---|
| Findley et al., 2016 [143] | USA [AMRO] | 1. To learn about the experience of abuse among university students who have identified as having a disability, and how and to whom they reached out for assistance<br>2. To explore the hypothesis that rates of abuse would be higher among students with disabilities who live on campus, and if they experience abuse, they would not necessarily reach out for assistance. | N = 101; n = 65 female; all with disability | Cross-sectional; structured quantitative survey; self-administered online | Men with disability vs. women with disability |
| Giraldo-Rodriguez et al., 2015 [210] | Mexico [AMRO] | 1. To determine the prevalence of self-reported abuse in elderly Mexican adults with long-term disabilities<br>2. To identify associated risk factors. | N = 1089; 50.3% women; all with long-term disabilities | Cross-sectional; structured quantitative survey; face-to-face | Men with disability vs. women with disability |
| Longobardi et al., 2018 [144] | Italy [EURO] | 1. To analyse the extent of abuse amongst persons with disability in an Italian context<br>2. To uncover the relationship between types of disability and the abuse of adults (sexual, physical, or based on negligence or rejection). | N = 237; 50.2% women; all with disabilities | Cross-sectional; structured quantitative survey; self-administered on paper (electronically if needed due to disability) | Men with disability vs. women with disability |
| Ministry of Labor, Invalids and Social Affairs et al., 2020 [120] | Viet Nam [WPRO] | 1. To measure and assess the prevalence and frequencies of different forms of violence against women aged 15 to 64 caused by a current or former husband/partner, including violence against women with disabilities<br>2. To measure and assess the prevalence, frequency and place of occurrence of physical and sexual violence by non-partners against women since the age of 15 years<br>3. To measure and assess the prevalence of physical and sexual violence caused by any perpetrators to women during their childhood (aged under 15 years) | N = 5967; all women; 9% of women reported some form of disability | Cross-sectional; structured quantitative survey; face-to-face and CAPI | Women with disability vs. women without disability |
| National Statistics Office et al., 2018 [121] | Mongolia [WPRO] | 1. To obtain information about reliable estimates of the prevalence and incidence of different forms of violence against women in a way that is comparable with other studies around the world using the World Health Organization methodology<br>2. To assess the extent to which intimate partner violence is associated with a range of health and other outcomes. | N = 7290; all women; 28.9% with some form of disability | Cross-sectional; structured quantitative survey; face-to-face | Women with disability vs. women without disability |

(*Continued*)

**Table 2.** (Continued)

| Article | Country and region | Research question(s) | Sample–N, % women, % with disability | Methods–study design, data collection methods | Comparison of violence experiences–men vs. women; women with disabilities vs. women without disabilities |
|---|---|---|---|---|---|
| National Commission for Women and Children, 2017 [119] | Bhutan [SEARO] | 1. To measure of the prevalence of different forms of violence against women and girls comparable with other studies around the world<br>2. To assess associations between IPV and a range of health and other outcomes. | N = 2184; all women; 5.8% of women with some form of disability | Cross-sectional; structured quantitative survey; face-to-face | Women with disability vs. women without disability |
| CREA 2012 [211] | Bangladesh, India and Nepal [SEARO] | 1. To investigate the hypothesis that women who are outside the mainstream of the South Asian society suffer high rates of violence and are frequently unable to seek and receive protection from State agencies<br>2. To quantify levels of violence suffered by marginalised women in different settings in South Asia<br>3. To catalogue the experiences of women (who have suffered violence) in their help-seeking behaviours | N = 816; all women; all with disability | Cross-sectional; structured quantitative survey; face-to-face | No comparison |
| SINTEF 2016a [131] | Botswana [AFRO] | To conduct a national, representative study on living conditions among persons with disability in Botswana | N = 9904; n = 5280 women; 2830 with disability | Cross-sectional; structured quantitative survey; face-to-face | Men with disability vs. women with disability |
| Bureau of Justice Statistics 2017 [150] | USA [AMRO] | To present the rates of nonfatal violent victimization against persons with and without disabilities, describe types of disabilities, and compare victim characteristics. | N not included in report | Cross-sectional; structured quantitative survey; face-to-face | Men with disability vs. women with disability; women with disability vs. women without disability |
| SINTEF 2016b [132] | Nepal [SEARO] | 1. To carry out a representative nation-wide study on physical, social, economic and living conditions of persons with disability in Nepal<br>2. To generate a complete representative data set on living conditions among the persons with disability | 4123 respondents; 2123 with disability and 2000 without disability; 51.6% women overall; 1023 women with disability | Cross-sectional; structured quantitative survey; face-to-face | Men with disability vs. women with disability |
| Uganda Bureau of Statistics 2018 [212] | Uganda [AFRO] | 1. To provide data at the national and subnational level that will allow the computation of disability rates by selected characteristics<br>2. To undertake a test of response options to the Washington Group short set of questions<br>3. To provide information for monitoring disability-related SDG indicators | 14,617 adults; 42.1% women; % women with disability not reported | Cross-sectional; structured quantitative survey; face-to-face and CAPI | Men with disability vs. women with disability |

(*Continued*)

**Table 2.** (Continued)

| Article | Country and region | Research question(s) | Sample–N, % women, % with disability | Methods–study design, data collection methods | Comparison of violence experiences–men vs. women; women with disabilities vs. women without disabilities |
|---|---|---|---|---|---|
| Schröttle et al 2013 [156] | Germany [EURO] | 1. To explore the experiences of women with disabilities in comparison to women without disabilities<br>2. To determine to what extent can similarities and/or differences be found, for example, in terms of prevalence and forms of violence | 3 samples:<br>1. N = 800; all women; all with disabilities living in private households;<br>2. N = 420; all women; all living in residential institutions<br>3. N = 341 all women; blind, deaf or severely disabled, purposively selected<br>Compared with nationally representative general population survey (N = 8445 women) for purposes of data analysis | Cross-sectional; structured quantitative survey; face-to-face | Women with specific disabilities and/ or living conditions vs. women with other types of disabilities and/ or living conditions, i.e. women with disabilities in households vs. women with disabilities in institutions vs. deaf women vs. blind women vs. women with severe disability |
| Instituto Nacional de Estadistica e Informatica, 2014 [213] | Peru [AMRO] | Provide information on reproductive health and maternal and child health, prevalence of use of contraceptive methods, pregnancy and delivery; immunizations; prevalence of certain diseases in children; weight and height of children under five years of age, weight and height for pregnant women, knowledge and ways to prevent AIDS and domestic violence. | 29941 households, including 24872 women 15–49.<br>n = 13,278 participated in DV module | Cross-sectional study; structured quantitative survey; face-to-face | No comparison |
| Minsalud, Profamilia 2015 [214] | Colombia [AMRO] | To provide information to establish the demographic changes that have occurred in the country; to collect information on the main components of demographic dynamics: fertility and infant mortality and related subjects, such as the size and composition of households and female headship. | 92,779 men and women; 52479 women; 20,855 women participated in DV module | Cross-sectional study; structured quantitative survey; face-to-face | No comparison |
| Centro de Estudios Sociales y Demográficos (CESDEM) et al., 2014 [215] | Dominican Republic [AMRO] | To provide updated information on the levels of fertility and infant mortality; fertility preferences; knowledge and use of planning methods family; maternal and child health; knowledge and attitudes towards HIV/AIDS and other infections of sexual transmission (STI); HIV prevalence among the adult population; violence against women, expenses of household health, among others. | 20,261 men and women; 9,955 women total; 6.996 participated in DV module | Cross-sectional study; structured quantitative survey; face-to-face | No comparison |

*(Continued)*

**Table 2.** (Continued)

| Article | Country and region | Research question(s) | Sample–N, % women, % with disability | Methods–study design, data collection methods | Comparison of violence experiences–men vs. women; women with disabilities vs. women without disabilities |
|---|---|---|---|---|---|
| Secretaría de Salud [Honduras] et al., 2013 [216] | Honduras [AMRO] | Providing updated information on basic health indicators, which allow the Ministry of Health to visualize the progress achieved in the period that includes the study, and the National Institute of Statistics of Honduras to generate the information expressed in the data | 29,877 men and women; 22,757 women total; n = 11,302 women participated in the DV module | Cross-sectional study; structured quantitative survey; face-to-face | No comparison |

respondents is highlighted within the analyses [79]. Other studies compared types of violence or levels of exposure experienced by men and women with disabilities [80–83], with gender as one aspect of the comparisons or variables explored. In other studies included within this category, the research questions did not explicitly focus on gender and there was limited sex-disaggregated reporting of violence exposure [84–86].

The second typology were studies that included respondents with and without disabilities, and included a number of different research foci that primarily sought to examine the association between disability and violence, with n = 36 studies which included men and women with and without disabilities, and n = 75 studies which included only women, with and without disabilities. This final category includes DHS surveys which, while including men in the larger sample, only include women in the violence modules.

Amongst the studies that included men and women, with and without disabilities, research objectives focused on the extent to which disability operated as a risk factor for violence victimization, through focusing on a comparison of the risk of violence between persons with and without disability. Many of these studies explicitly included gender analysis within their research objectives, comparing men with disabilities and women with disabilities, and exploring the role of gender in prevalence, types or perpetrators of violence [39, 40, 87–95].

Studies that included only women, with and without disabilities, similarly focused on comparisons of violence exposure between women with and without disabilities [38, 96–110]. Some studies focused in specific sub-groups of women; for example, Coston et al. (2019) compared experiences of heterosexual women with and without disabilities and bisexual women with and without disabilities [111], while Slayter et al. (2017) focused on women aged 18–21 to explore prevalence and risk factors for past year IPV [112]. Other specific foci of studies included the question of satisfaction with health service providers [113], health care access [114], employment status and its association with disability and violence [115], comparisons of mental and physical health outcomes related to IPV [116] and birth outcomes of children of women affected by violence [117]. Only one study included comparison of different types of disability and violence exposure [118]. Three recent national violence against women surveys–conducted in Viet Nam, Bhutan and Mongolia–explored associations between disability and violence exposure amongst women alongside other primary research objectives of identifying prevalence of violence and health impacts [119–121].

A sub-set of this typology was case-control studies, where a sample of respondents with disabilities was matched and compared with a sample of respondents without disabilities or with different disabilities, to identify patterns in prevalence, patterns or outcomes of violence

exposure [46–58]. These studies explicitly focused on questions of comparison of violence exposure between women, or men and women, with and without disabilities. Some case-control studies focused on specific disabilities, for example fibromyalgia [48], spinal cord injury [59], chronic pelvic pain [51] and deafness [50].

## Measurement of violence

Table 3 displays measurement of violence within the 174 included manuscripts and reports. Of the included manuscripts and reports, n = 134 measured multiple types of violence, whereas n = 18 focused only on sexual violence [41, 45, 53, 56, 58, 59, 66, 85, 86, 88, 93, 100, 108, 109, 118, 122–124], n = 13 focused only physical violence [76, 77, 84, 103, 112, 116, 125–132] and one study focused only on economic abuse [133]. Physical violence was assessed in n = 146 manuscripts or reports, sexual violence was assessed in n = 144, psychological violence (including items defined by authors as emotional abuse or controlling behaviors, verbal abuse and threats) was assessed n = 87 and economic violence was assessed in n = 17 manuscripts or reports. Description or definition of the forms of violence assessed was not specified at all in n = 7 manuscripts or reports [70, 83, 134–138].

Disability-specific types of violence were measured in n = 11 manuscripts or reports. Being prevented from using an assistive device and refusal by an abuser to provide for basic needs required by a woman with disabilities were commonly operationalized forms of violence. Examples of these items are "As an adult, has anyone you know ignored or refused to help you with an important personal need such as using the bathroom, banking, dressing, eating, communicating, or going out in the community?" [75] and, "In the last year, has anyone you know broken or kept you from using important things such as a Phone; Wheelchair; Cane; Walker; Respirator; Communication device; Service animal; and other assistive devices" [139]. A full list of disability-specific violence items is included in Fig 2, Disability-specific items n = 61 manuscripts or reports focused only on intimate partner violence, whereas the other reports and manuscripts either specified that the context of violence included IPV and other contexts (for example, violence perpetrated by an acquaintance) or did not specify the context of violence. n = 9 studies explicitly included perpetrators specific to women with disabilities, for example, personal carers or staff at institutions [60, 77, 118, 139–144].

The level of detail of description of the violence measurement instrument utilized and reported in manuscripts varied widely. n = 82 reports or manuscripts named the scale utilized to measure violence and included some or all items used to measure violence; n = 20 included the name of the scale but did not include any items; n = 71 did not name the scale but did include some or all items, and n = 18 included no description of violence measurement, either the name of the scale or examples of items [44, 55, 70, 73, 85, 92, 113, 118, 122, 125, 136, 137, 145–150].

Violence was assessed using a range of measurement instruments. n = 27 used the Conflicts Tactics Scale in its original or revised version, n = 13 included the WHO Domestic Violence questionnaire or a sub-set of questions from the questionnaire and n = 19 utilized the DHS Domestic Violence module. The majority of reports or manuscripts used acts-specific measures for all forms of violence (n = 113). n = 7 studies utilized an acts-specific measure for some forms of violence but only a single item for other forms of violence [60, 87, 110, 114, 115, 151–153]. Several of these studies used acts-based measures to assess physical violence and a single item to assess sexual violence; for example, Barrett et al. asked "Has an intimate partner ever hit, slapped, pushed, kicked, or physically hurt you in any way?" to assess physical violence, and measured sexual violence with a single item: "Have you ever experienced any unwanted sex by a current or former intimate partner?" [114]. n = 3 studies utilized a single

**Table 3. Measurement of violence.**

| Article | Type(s) of violence | Context/ Perpetrator(s) | Scale, measure, specific items | Time frame(s) | Violence measure(s) specific to women with disabilities (Y/N) If yes, specific items/ measures |
|---|---|---|---|---|---|
| Pandey et al., 2012 [96] | Control, humiliation, physical, sexual (included within physical abuse) | Relationship; Husband | Adapted from Demographic and Health Surveys Domestic Violence module Controls: (1) He is jealous or angry if you talk to other men. (2) He frequently accuses you of being unfaithful. (3) He does not permit you to meet your female friends. (4) He tries to limit your contact with your family. (5) He insists on knowing where you are at all times. (6) He does not trust you with any money. Humiliates: Does your husband (1) Say or do something to humiliate you in front of others? (2) Threaten to hurt or harm you or someone close to you? (3) Insult you or make you feel bad about yourself? Physical abuse: Does your husband: (1) Slap you? (2) Twist your arm or pull your hair? (3) Push you, shake you, or throw something at you? (4) Punch you with his fist or with something that could hurt you? (5) Kick you, drag you, or beat you up? (6) Try to choke you or burn you on purpose? (7) Threaten or attack you with a knife, gun, or any other weapon? (8) Physically force you to have sexual intercourse with him even when you did not want to? (9) Force you to perform any sexual acts you did not want to? | Lifetime | No; N/A |
| Valera and Kucyi, 2017 [125] | Physical | Relationship; partner | Not specified | Not specified | No; N/A |
| Valera et al., 2019 [162] | Physical, sexual | Relationship; partner | Conflict Tactics Scale [CTS]; additional 10 items from the Severity of Violence Against Women Items not listed | Not specified | No; N/A |
| Slayter, 2009 [163] | Control (verbal abuse and coercion), physical violence (includes sexual abuse) | Relationship; partner | Name of scale not specified; Verbal abuse and coercion: "In the past 12 months, when you've had an argument, how often did your husband/boyfriend (interpreted in this text as 'partner') insult you; swear at you (or 'call you out,' in common parlance); refuse to talk about an issue, accuse you of being with another man; stomp out; do or say something just to spite you; try to control your every move; withhold money, make you ask for money, or take yours; prevent you from going to school or work; harass you with phone calls at work or show up at your workplace to harass you?" [response categories: rarely, sometimes, often or very often] Threats: "In the past 12 months, when you've had an argument, how often did your partner threaten to hurt children; threaten to take children away; threaten to turn you in to a government agency; threaten you with a knife/gun; threaten to kill you or threaten to hurt your family/friends" [response categories: rarely, sometimes, often or very often] Physical violence: "In the past 12 months, when you've had an argument, how often did your partner throw things at you; destroy your belongings; push or shove you; slap you; kick or hit you; try to hit you with an object; injure you so that you needed medical treatment; cause you to miss work because of your injuries; beat you up; choke you; force you to have sex; burn you; lock you up; or either cut you with a knife or fire at you with a gun." [response categories: rarely, sometimes, often or very often] | Past 12 months | No; N/A |
| Powers, 2002 [60] | Physical, psychological, verbal, sexual, financial | Disability support; personal assistance–can be family member, friend, paid employee | No name of the scale (developed based on qualitative work); Items include if personal assistant: Makes decisions or choices without asking; touches sexually in unwanted ways; withholds, overdoses or steals medication; hits, kicks, slaps, shoves or is otherwise physically abusive; neglects or abuses children in home; forges checks, uses credit cards, steals money or other things. Also included general screener for any type of abuse: "Have you ever been hit, slapped, kicked or otherwise physically hurt by anyone" and "Either as a child or an adult, has anyone ever touched you in a way you did not want to be touched or forced you into sexual activity you did not want." | Lifetime | Yes; Main items in the survey focused on disability-specific violence measure–if personal assistant: Makes decisions or choices without asking; touches sexually in unwanted ways; withholds, overdoses or steals medication; hits, kicks, slaps, shoves or is otherwise physically abusive; neglects or abuses children in home; forges checks, uses credit cards, steals money or other things. |
| Alangea et al., 2018 [164] | Physical, sexual, emotional, economic | Relationship; Current or previous husband or boyfriend | WHO Multi-Country Study Instrument [MCS]; Physical: How many times has your current or any previous husband or boyfriend slapped you or thrown something at you which could hurt?; How many times has your current or any previous husband or boyfriend pushed or shoved you?; How many times has your current or any previous husband or boyfriend hit you with a fist or something else that could hurt?; How many times has/did your current or any previous husband or boyfriend kick, drag, beat, choke or burnt you?; How many times has your current or any previous husband or boyfriend threatened to use or actually used a gun, knife or other weapon against you? Sexual: How many times has a current or previous husband or boyfriend ever physically forced you to have sex when you did not want to?; How many times has a current or previous husband or boyfriend, husband or partner used threats or intimidation (but not physical force) to get you to have sex when you did not want to?; How many times has a current or previous husband or boyfriend ever forced you to do something else sexual that did not want to do? | 12 months | No; N/A |
| Astbury and Walji, 2014 [46] | Physical, sexual, psychological | Household; family members (not partner) | WHO MCS; Physical: being slapped or having something thrown that could hurt the woman, being pushed or shoved, hit with a fist or something else that could hurt; being kicked, dragged, or beaten; choked or burnt on purpose; and/or threatened with the use or actual use of a gun, knife, or other weapon. Psychological: whether a household member had insulted or made the woman feel bad about herself, had belittled or humiliated her in front of other people, deliberately did things to scare or intimidate her, and threatened to hurt her or someone she cared about. Sexual: being physically forced to have unwanted sexual intercourse, having sexual intercourse she did not want because she was afraid of what the perpetrator might do, and being forced to do something sexual that she found degrading or humiliating. | Lifetime | No; N/A |
| Cannell et al., 2015 [38] | Physical or verbal abuse | Any (not institution); Family member or friend | No scale specified; items were: "Over the past year, were you physically abused by being hit, slapped, pushed, shoved, punched, or threatened with a weapon by a family member or close friend?" and "Over the past year: Were you verbally abused by being made fun of, severely criticized, told you were a stupid or worthless person, or threatened with harm to yourself, your possessions, or your pets, by a family member or close friend?" | Past year | No; N/A |
| Coston, 2019 [111] | Physical, sexual, emotional, stalking | Relationship; current or former (including ex) emotional, romantic, sexual, dating, cohabiting, and/or marital partners | Scale not specified; Physical: measured with a series of questions, including, but not limited to, being slapped, hit with a fist or something hard, being choked, or having a weapon used against you. Sexual violence included coerced or forcible sexual exposure, groping and fondling, and vaginal/oral/anal penetrative sex without consent (or while unable to consent). Emotional: took into account a partner's emotional manipulation, such as anger, put-downs, calling names, or humiliating a person in front of others; Psychological: measured by examining whether a person had ever been kept from seeing friends or family, been forcibly moved to a new home/location, been told what to eat or wear, or been threatened with violence for not behaving in particular ways. Stalking: included being watched or followed, spied on with listening devices/GPS/camera, being approached at places like home, work or school, or being sent unwanted and threatening messages/texts/calls/emails. | Past 12 months, past 3 years | No; N/A |
| Dembo et al., 2018 [39] | Physical and/ or sexual | Any (not institution); intimate partner (spouse, boyfriend, girlfriend, or ex-partner), relative, such as a parent, child, or sibling; known other, such as a friend, neighbor, or colleague; or a stranger. | National Crime Vicitimization Survey; Sexual: including cases of completed and attempted rape, sexual assaults with or without injuries, and unwanted sexual contact; Robbery: including completed or attempted robbery, with or without injuries; Non-sexual assault: including completed aggravated or attempted assault, with or without injuries, with or without a weapon, or simple assault completed with injuries; Verbal: threats of rape, sexual assault, or assault. | Lifetime | No; N/A |

*(Continued)*

**Table 3.** (*Continued*)

| Article | Type(s) of violence | Context/ Perpetrator(s) | Scale, measure, specific items | Time frame(s) | Violence measure(s) specific to women with disabilities (Y/N) If yes, specific items/ measures |
|---|---|---|---|---|---|
| Emerson et al., 2016 [40] | Verbal abuse/ threats, physical abuse | Public places; any perpetrator | Created measure of feelings and experiences of safety in the following public places: (a) on public transport; (b) at or around a bus or train station; (c) in commercial places like shopping centres, shops or petrol stations; (d) in places of entertainment like theatres, cinema, cafes or restaurants; (e) at pubs, nightclubs, discos or clubs; (f) in car parks; (g) outside, such as on the street, in parks or sports grounds.<br>For each of these places, asked: Have you ever "been insulted, called names, threatened or shouted at, in any of these places?" and Have you ever "been physically attacked in any of these places?" | Past 12 months | No; N/A |
| Gibbs et al., 2018 [165] | Physical, psychological | Relationship; husband | WHO MCS;<br>Physical: 5 items, included acts such as whether the woman had been slapped, pushed, hit, threatened with a knife or gun, or had them used on her;<br>Psychological: 7 items, including items about being humiliated, belittled, scared, or threatened. | Past 12 months | No; N/A |
| Guedes et al., 2016 [87] | Physical, psychological | Relationship–partner; family member | Separately asked if partner or family member had: "Hurt; Insulted; Threatened or Screamed at you" | Lifetime; past 6 months | No; N/A |
| Kutin et al., 2017 [133] | Economic | Relationship; partner | Name of scale not specified; Items were 1) stopped or tried to stop you knowing about or having access to household money; 2) stopped or tried to stop you from working or earning money, or studying; 3) deprived you of basic needs e.g. food, shelter, sleep, assistive aids); 4) damaged, destroyed or stole any of your property; and 5) stopped or tried to stop you from using the telephone, Internet or family car. | Lifetime | No; N/A |
| Le et al., 2016 [166] | Physical, sexual | Any (not institution) | Juvenile Victimisation Questionnaire Revised 2 (JVQ R2)—youth self-report lifetime version; examples of items included:<br>Crimes: Assault with weapon: "Sometimes people are attacked with sticks, rocks, guns, knives, or other things that would hurt. At any time in your life, did anyone hit or attack you on purpose with an object or weapon? Somewhere like: at home, at school, at a store, in a car, on the street, or anywhere else? At any time in your life, did anyone hit or attack you without using an object or weapon? At any time in your life, did someone threaten to hurt you when you thought they might really do it?"<br>Peer and sibling assault: "Sometimes groups of kids or gangs attack people. At any time in your life, did a group of kids or a gang hit, jump, or attack you? At any time in your life, did any kid, even a brother or sister, hit you? Somewhere like: at home, at school, out playing, in a store, or anywhere else? At any time in your life, did you get scared or feel really bad because kids were calling you names, saying mean things to you, or saying they didn't want you around? At any time in your life, did a boyfriend or girlfriend or anyone you went on a date with slap or hit you?<br>Sexual assault: At any time in your life, did a grown-up you know touch your private parts when they shouldn't have or make you touch their private parts? Or did a grown-up you know force you to have sex? At any time in your life, did anyone try to force you to have sex; that is, sexual intercourse of any kind, even if it didn't happen? | Lifetime | |
| Platt et al., 2017 [75] | Sexual, physical, economic; disability-specific | Any; any known perpetrator | Name of scale not specified; Items were:<br>Physical: As an adult, has anyone you know made you feel unsafe?<br>As an adult, has anyone you know yelled at you over and over again or hurt your feelings on purpose?<br>As an adult, has anyone you know made you afraid they would hit, kick, slap, shove, or otherwise physically hurt you?<br>As an adult, has anyone you know hit, kicked, slapped, shoved, or otherwise physically hurt you?<br>As an adult, has anyone you know physically handled you in a rough way?<br>As an adult, has anyone you know held or tied you down or made you stay someplace when you did not want to?<br>Disability-specific: As an adult, has anyone you know ignored or refused to help you with an important personal need such as using the bathroom, banking, dressing, eating, communicating, or going out in the community?<br>As an adult, has anyone you know purposely broken or kept you from using things such as a wheelchair, breathing machine, communication device, or service animal?;<br>As an adult, has anyone you know kept you from taking your medicine or given you more medicine than they were supposed to?;<br>Economic: As an adult, has anyone you know stolen or misused your money, bank account, or debit/credit cards?<br>Sexual: As an adult, has anyone you know made you afraid they were going to touch you in a sexual way that you did not want?<br>As an adult, has anyone you know touched you in a sexual way that you did not want?<br>As an adult, has anyone you know made you touch them in a sexual way that you did not want?<br>As an adult, has anyone forced you to have sex? | Lifetime | Yes<br>As an adult, has anyone you know ignored or refused to help you with an important personal need such as using the bathroom, banking, dressing, eating, communicating, or going out in the community?<br>As an adult, has anyone you know purposely broken or kept you from using things such as a wheelchair, breathing machine, communication device, or service animal?<br>As an adult, has anyone you know kept you from taking your medicine or given you more medicine than they were supposed to? |
| Puri et al., 2015 [61] | Physical, sexual, psychological | Any; anyone | WHO MCS;<br>Physical: Slapped you or thrown something at you that could hurt you; Pushed you or shoved you or pulled your hair; Hit you with his fist or with something else that could hurt you; Kicked you, dragged you or beaten you up; Choked or burnt you on purpose; Threatened to use or actually used a gun, knife or other weapon against you; Thrown out from the house.<br>Psychological: Insulted you or made you feel bad about yourself, Belittled or humiliated you in front of other people, Done things to scare or intimidate you on purpose, Threatened to hurt you or someone you care about, Gave mental pressure to earn money.<br>Sexual: Physically forced you to have sexual intercourse when you did not want to; Forced you to have sexual intercourse you did not want to because you were afraid of what your s/he might do; Force you to do something sexual that you found degrading or humiliating; Forced sexual activity like kissing, touching, masturbation, oral sex etc. | Lifetime, past 12 months | No; N/A |
| Slayter et al., 2017 [112] | Physical | Relationship; partner | Name of scale not specified;<br>Less-severe: being pushed, grabbed, or shoved by an intimate partner; having something thrown at them by that partner; or being slapped or hit by that partner.<br>Severe: kicked, bit, or hit with a fist by an intimate partner; beat up, choked, burned, or scalded by a partner; or threatened with a knife or gun by a partner. | Past 12 months | No; N/A |
| Valentine et al., 2019 [97] | Physical, sexual, psychological | Relationship; partner | Demographic and Health Surveys Domestic Violence [DHS DV] module; items not listed | Lifetime; past 12 months | No; N/A |
| Wall et al., 2018 [134] | Experience that resulted in traumatic brain injury [TBI]–type(s) of violence not included | Not specified; not specified | TBI screened using Ohio State University TBI Identification Method [OSU-TBI-ID]; items not specified | Not specified | No; N/A |
| Mirindi 2018 [145] | Rape; physical, verbal, psychological violence experienced prior to rape | Not specified; not specified | Name of scale not stated; items not included | Not specified | No; N/A |

(*Continued*)

**Table 3.** (Continued)

| Article | Type(s) of violence | Context/ Perpetrator(s) | Scale, measure, specific items | Time frame(s) | Violence measure(s) specific to women with disabilities (Y/N) If yes, specific items/ measures |
|---|---|---|---|---|---|
| Anderson et al., 2012 [64] | Psychological, physical, sexual | Relationship; partner | CTS2—Victimization sub-scales; Psychological Aggression: 8 items; Minor items on this subscale specify tactics including insulting and swearing, shouting and yelling, stomping out of an argument, and spiting one's partner. Severe items include name-calling, destruction of property, and threats of physical violence. Physical Assault: 12 items; Minor items query tactics such as pushing, grabbing, and shoving, whereas Severe items query punching, kicking, choking, burning, and use of a weapon. Sexual Coercion: nonconsensual sexual acts, including, but not limited to, imposing unprotected, oral, and anal sex. | Lifetime; past 12 months | No; N/A |
| Anderson et al., 2014 [63] | Psychological, physical, sexual | Relationship; partner | CTS2—Victimization sub-scales; items listed above in [22] | Lifetime; past 12 months | No; N/A |
| Anderson et al., 2011 [62] | Psychological, physical, sexual | Relationship; partner | CTS2—Victimization sub-scales; items listed above in [22] | Lifetime; past 12 months | No; N/A |
| Barrett et al., 2009 [114] | Threats, physical, sexual | Relationship; Current or former intimate partner | Not specified; Items: Has an intimate partner ever threatened you with physical violence?; Has an intimate partner ever hit, slapped, pushed, kicked, or physically hurt you in any way?; or Have you ever experienced any unwanted sex by a current or former intimate partner? | Lifetime | No; N/A |
| Brownridge 2006 [98] | Threats, physical, sexual | Relationship; Current or former intimate partner | Modified CTS; Physical: being pushed, grabbed, or shoved in a way that could hurt; being slapped; being choked; having something thrown that could hurt; being hit with something that could hurt; being threatened with or having a knife or gun used; being kicked, bit, or hit with a fist; being beaten; Threat: being threatened to be hit with a fist or anything else that could hurt; Sexual: being forced into any sexual activity by being threatened, held down, or hurt in some way | Past 5 years; past 12 months | No; N/A |
| Brownridge 2008 [99] | Threats, physical, sexual | Relationship; Current or former intimate partner | Modified CTS; violence items not listed; items related to control: Patriarchal dominance: single item that asked the respondent if her partner prevented her from knowing about or having access to the family income, even if she asked. Possessiveness: single item that asked the respondent if her partner demanded to know who she was with and where she was at all times. Sexual jealousy: single item that asked the respondent if her partner was jealous and did not want her to talk to other men. | Past 5 years | No; N/A |
| Brunnberg et al., 2012 [88] | Sexual | Any; not specified | Name of scale not specified; Items included questions about power relationship between respondent and first intercourse partner, including: When you first had intercourse was it something that. . .you really wanted, you wanted, just happened, you did not want but were not forced into, you were forced into; Whether the first sexual intercourse took place under the influence of alcohol or drugs. | Not specified | No; N/A |
| Curry et al., 2009 [139] | Physical, psychological, economic, sexual, disability-specific | Any; specified in scale that perpetrator could be male or female, paid or unpaid providers, family, friends, or other people | Adapted Abuse Assessment Screen–Disability (AAS-D); Psychological: In the last year, has anyone you know made you feel unsafe? In the last year, has anyone you know: Yelled at you over and over again? Hurt your feelings on purpose? Economic: In the last year, has anyone you know: Stolen your money, important items, or equipment? Signed your checks to take money from you? Used your credit or debit card without your OK? Physical: In the last year, has anyone you know: Made you afraid they would hit, kick, slap, or shove you? Actually hit, kicked, slapped, or shoved you? Handled you roughly? Held or tied you down or made you stay someplace when you didn't want to? Physically hurt you in any way? Sexual: In the last year, has anyone you know: Made you afraid they were going to touch you in a sexual way that you did not want? Actually touched you in a sexual way that you did not want? Taken advantage of you in sexual ways you did not want? Made you look at or took sexual pictures of you Been naked in front of you or made you be naked Asked about your sex life Made you feel bad about your body Also asked about perpetrator characteristics: 1. Is the person someone you depend on for personal care (like dressing, bathing, or using the toilet?) 2. Is the person someone who drinks too much or abuses drugs? 3. Is the person someone who controls whether you get the services and health care you need? 4. Is the person someone who controls most of your daily activities? 5. Is the person someone who gets jealous or has severe fits of anger? 6. Is the person someone who decides whether or not you see your family and friends? 7. Is the person someone who makes you afraid they would or actually has hurt your pet, children, or someone else important to you? 8. Is the person someone who has hurt other people? 9. As time goes by, is the abuse getting worse or happening more often? 10. Has the person ever tried to choke you? 11. Has the person ever used a weapon against you or threatened you with a lethal weapon? 12. If "yes," was the weapon a gun? | Past 12 months | Yes; In the last year, has anyone you know refused or forgotten to help you with an important personal need such as Toileting or going to the bathroom Bathing Helping you move Getting dressed Getting food or water In the last year, has anyone you know broken or kept you from using important things such as a Phone Wheelchair Cane Walker Respirator Communication device Service animal Other assistive devices In the last year, has anyone you know Kept you from taking your medication? Given you too much or too little medication? |
| Du Mont et al., 2013 [113] | Domestic violence, sexual assault, physical assault | Any; Intimate partner, stranger, parent, co-worker, friend, ex-partner, other relative, date, sex trade customer | Not specified; items not listed | Not specified | No; N/A |
| Pollard et al., 2014 [159] | Physical, emotional, sexual | Relationship; intimate partner | DHS DV module questions; items included not listed | Lifetime; past 12 months | No; N/A |
| Powers et al., 2009 [73] | Sexual abuse, physical abuse, multiple types of abuse, low risk of abuse (latent classes) | Relationship; partner | Not specified; not specified | Lifetime; past 12 months | No; N/A |

*(Continued)*

**Table 3.** (Continued)

| Article | Type(s) of violence | Context/ Perpetrator(s) | Scale, measure, specific items | Time frame(s) | Violence measure(s) specific to women with disabilities (Y/N) If yes, specific items/ measures |
|---|---|---|---|---|---|
| Smith et al., 2008 [115] | Sexual, physical | Any; anyone. For some items, relationship/ current or former intimate partner specified | Name of scale not specified; items were: <br> Sexual: In the past 12 months, has anyone exposed you to unwanted sexual situations that did not involve physical touching?; In the past 12 months, has anyone touched sexual parts of your body after you said or showed that you didn't want them to or without your consent?; In the past 12 months, has anyone attempted to have sex with you after you said or showed that you didn't want to or without your consent, but the sex did not occur?; In the past 12 months, has anyone had sex with you after you said or showed that you didn't want to or without your consent? <br> Has anyone ever attempted to have sex with you after you said or showed that you didn't want to or without your consent, but sex did not occur?; Has anyone ever had sex with you after you said or showed that you didn't want them to or without your consent?; Have you ever experienced any unwanted sex by a current or former intimate partner? <br> Physical: Has an intimate partner ever threatened you with physical violence?; This includes threatening to hit, slap, push, kick, or physically hurt you in any way; Has an intimate partner ever hit, slapped, pushed, kicked, or physically hurt you in any way?; Other than what you have already told me about, has an intimate partner ever attempted physical violence against you? | Lifetime; past 12 months | No; N/A |
| Alriksson-Schmidt et al., 2010 [100] | Sexual | Any; anyone | Single item: "Have you ever been physically forced to have sexual intercourse when you did not want to?" | Lifetime | No; N/A |
| Carbone-López et al., 2006 [89] | Physical, sexual, stalking | Relationship; current or former spouse or cohabiting intimate partner | CTS; <br> Physical: Specific items include: pushing/shoving, pulling hair, slapping, kicking, choking, beating up, throwing or hitting with an object, and the threat or use of a weapon (i.e., gun, knife, or other). <br> Sexual assault: forced sex by an intimate partner and included completed vaginal, anal, or oral sex. <br> Stalking: range of items, included an intimate partner who followed or spied on you, sent unsolicited letters or written correspondence, made unsolicited calls, stood outside a home, place of work, or recreation, showed up at places even though he or she had no business being there, left unwanted items for you to find, or tried to communicate against your will. | Lifetime | No; N/A |
| Eberhard-Gran et al., 2007 [101] | Physical, sexual | Any; anyone | Abuse Assessment Screen; <br> Physical: Have you ever, after the age of 18 years, been hit, slapped, kicked, or otherwise physically hurt by someone?; Have you during the last twelve months been hit, slapped, kicked, or otherwise physically hurt by someone?; If yes, how many times? <br> Sexual: Have you ever, as an adult, been coerced into sexual activities?; Have you ever, as an adult, been forced into sexual activities?; If yes, did it happen during the last twelve months? | Lifetime; past 12 months | No; N/A |
| Haydon et al., 2011 [41] | Sexual (coercion, forced) | Any apart from parent or caregiver | Name of scale not specified <br> Coerced sex: Have you ever been forced, in a non-physical way, to have any type of sexual activity against your will? For example, through verbal pressure, threats of harm or by being given alcohol or drugs? <br> Forced sex: Have you ever been physically forced to have any type of sexual activity against your will? | Lifetime | No; N/A |
| Morris et al., 2019 [65] | Physical, sexual, psychological | Relationship; partner | CTS2 –Victimization sub-scale <br> Psychological: items included: My partner called me fat or ugly. <br> Physical aggression: items included: slapped by partner; <br> Sexual: items included: forced to have sex through physical coercion (hitting, holding down or using a weapon) | Lifetime | No; N/A |
| Rasoulian et al., 2014 [126] | Physical | Relationship; husband | Scale developed for this study; <br> Have you ever been intentionally hurt physically by your husband in your lifetime?; If yes, 13 follow-up questions about type and severity of the physical violence. <br> Experience of violence during the past year; If yes, 12 follow-up questions about the type and severity of the physical violence | Lifetime; past 12 months | No; N/A |
| Stockl et al., [102] | Physical, sexual | Relationship; current intimate partner | CTS-revised; included specific acts ranging from being pushed away angrily and slapped in the face to being strangled and injured with a weapon. <br> Sexual: 5 items, reflect German criminal law and include acts from forced or attempted forced sexual intercourse to forced petting and the forced watching and re-enacting of pornographic material. | Not specified | No; N/A |
| Brownridge et al., 2016 [90] | Physical, sexual | Relationship; current or former partner | CTS; <br> Physical: having something thrown at you that could hurt; being pushed, grabbed, or shoved in a way that could hurt; being slapped; being hit with something that could hurt; being kicked, bit, or hit with a fist; being beaten; being choked; being threatened with or having a knife or gun used against the respondent; <br> Physical threat: being threatened to be hit with a fist or anything else that could hurt; <br> Sexual assault: being forced into any sexual activity by being threatened, held down, or hurt in some way | Past 5 years | No; N/A |
| Casteel et al., 2008 [42] | Physical, sexual | Any (not institution); boyfriend or girlfriend, stranger, or acquaintance. | National Violence against Women survey questions; <br> Physical: (12 items) physical contact with a weapon (eg, gun, knife), hands or feet, as well as threats with a weapon, by a boyfriend/girlfriend, stranger, or acquaintance. <br> Sexual: (5 items) included attempted and/or completed forms of sexual contact by force and/or threat of force, by a boyfriend/girlfriend, date, stranger, or acquaintance. | Past 12 months | No; N/A |
| Cimino et al., 2019 [167] | Psychological, physical, injury from partner; forced sex from non-partner | Relationship; partner or any; non-partner (sexual violence) | CTS2; IPV was a summation of yes responses to any item in subscales of the CTS-2: (a) psychological aggression, (b) physical assault, and (c) injury from a partner (sexual violence was not included in the analysis because all participants were exposed to forced sex). <br> History of non-intimate partner forced sex was a dichotomous variable defined as someone who was not a current or former intimate partner ever using force (i.e., hitting, holding down, or using a weapon) or threats of force to make you have sex. | Lifetime | No; N/A |
| Cohen et al., 2005 [168] | Physical, sexual, emotional, financial | Relationship; Current or former intimate partner | Modified CTS; <br> Physical abuse: whether a current or former partner threatened to hit them, threw something at them, pushed, grabbed, shoved or slapped (categorised as nonsevere violence), kicked, bit or hit, hit with something, beat up, choked, burned/scalded, or used or threatened with knife or gun (categorised as severe violence) <br> Sexual abuse: Has your partner or former partner forced you into any unwanted sexual activity by threatening you, holding you down, or hurting you in some way? <br> Emotional abuse: partner or former partner limited contact with family or friends, put you down or called you names to make you feel bad, was being jealous and didn't want you to talk to other men/women, harmed, or threatened to harm, someone close to you, demanded to know with whom you were and where you were at all times, and damaged or destroyed your possessions or property. <br> Financial abuse: Has your partner prevented you from knowing about or having access to the family income, even if you asked? | Past 5 years | No; N/A |

(Continued)

**Table 3.** (Continued)

| Article | Type(s) of violence | Context/ Perpetrator(s) | Scale, measure, specific items | Time frame(s) | Violence measure(s) specific to women with disabilities (Y/N) If yes, specific items/ measures |
|---|---|---|---|---|---|
| Cohen et al., 2006 [91] | Physical, sexual, emotional, financial | Relationship; Current or former intimate partner | Modified CTS;<br>Physical abuse: whether a current or former partner threatened to hit them, threw something at them, pushed, grabbed, shoved or slapped (categorised as nonsevere violence), kicked, bit or hit, hit with something, beat up, choked, burned/scalded, or used or threatened with knife or gun (categorised as severe violence)<br>Sexual abuse: Has your partner or former partner forced you into any unwanted sexual activity by threatening you, holding you down, or hurting you in some way?<br>Emotional abuse: partner or former partner limited contact with family or friends, put you down or called you names to make you feel bad, was being jealous and didn't want you to talk to other men/women, harmed, or threatened to harm, someone close to you, demanded to know with whom you were and where you were at all times, and damaged or destroyed your possessions or property.<br>Financial abuse: Has your partner prevented you from knowing about or having access to the family income, even if you asked? | Past 5 years | No; N/A |
| Curry et al., 2011 [140] | Physical, sexual, emotional | Any; paid or unpaid providers, family, friends, or other people in their lives and may be male or female. | Not specified; Items related to adult abuse were adapted from prior work by Curry et al. (2009) as well as 19 items from the scale developed based on Curry et al., women answered three dichotomous questions about any lifetime experience of emotional, physical, and/or sexual abuse. | Lifetime, past 12 months | No; N/A |
| Elliott Smith et al., 2015 [66] | Sexual | Any; any | CTS2 + some additional questions; additional items addressed sexual assault when the victim was drugged by the assailant or too inebriated to consent to the act. | Lifetime | No; N/A |
| Hahn et al., 2014 [67] | Physical, sexual | Relationship; partner | CTS;<br>6 items: (a) pushed, grabbed, or shoved them; (b) slapped, kicked, bit, or punched them; (c) threatened them with a weapon like a gun or knife; (d) cut or bruised them; (e) forced sex; and/or (f) caused injury requiring medical care | Past 12 months | No; N/A |
| Hasan et al., 2014 [169] | Physical, sexual, emotional | Relationship; current or ex partner | WHO MCS:<br>Physical: Slapped or thrown something at you; Pushed or shoved or pulled hair; Hit with fist or with something else; Kicked/dragged/beaten up; choked or burnt on purpose; Threatened to use a gun, knife, or other weapons against you;<br>Sexual: Physically forced to have sex; Experienced something sexual when you were afraid of what your intimate partner might do; Experienced something sexual that you found degrading or humiliating<br>Emotional: Insulted you or made you feel bad about yourself; Belittled/humiliated you in front of other people; Done things to scare/ intimidate you on purpose; Threatened to hurt someone you care about; | Lifetime, past 12 months | No; N/A |
| Johnston-McCabe et al., 2011 [74] | Physical, sexual, psychological, life-threatening | Relationship; partner | CTS with some revisions; violence items not listed | Not specified | Yes; items added as to whether or not the abusive partner is Deaf, Hard of Hearing, or hearing; the age of onset of partners' hearing loss; the degree of partners' hearing loss; and detailed descriptions of the most recent, worst, and first experiences of abuse from the abusive partner. |
| Krnjacki et al., 2016 [92] | Physical assault; sexual assault; partner violence (includes physical, emotional and sexual violence from a current or previous partner); and stalking and harassment. | Relationship and non-relationship; anyone for some items, for some items current or ex partner | Not specified; items not listed | Lifetime (since age 15), past 12 months | No; N/A |
| Martin et al., 2006 [151] | Physical, sexual | Any; any | Not specified;<br>Physical: whether anyone had "pushed, hit, slapped, kicked, or physically hurt them in any other way" during the past year.<br>Sexual: whether anyone had "forced them to have sex or do sexual things" during the past year. | Past 12 months | No; N/A |
| McFarlane et al., 2001 [141] | Physical, sexual, disability-specific | Relationship, non-relationship, context with care provider; intimate partner, stranger, care provider, health professional, family member | Abuse Assessment Screen–Disability;<br>Physical: Within the last year, have you been hit, slapped, kicked, pushed, shoved, or otherwise physically hurt by someone?;<br>Sexual: Within the last year, has anyone forced you to have sexual activities? | Past 12 months | Yes; Within the last year, has anyone prevented you from using a wheelchair, cane, respirator, or other assistive devices? Within the last year, has anyone you depend on refused to help you with an important personal need, such as taking your medicine, getting to the bathroom, getting out of bed, bathing, getting dressed, or getting food or drink? |
| Mitra et al., 2012 [103] | Physical | Relationship; partner or ex-partner | Not specified;<br>Items: if pushed, hit, slapped, kicked, choked or physically hurt in any way by an ex-husband/partner and whether they were "physically hurt in any way" by their husband/partner during the 12-months before and during their most recent pregnancy. | 12 months before and during most recent pregnancy | No; N/A |
| Nosek et al., 2006 [142] | Physical, sexual, disability-specific | Relationship, non-relationship, context with care provider; intimate partner, stranger, care provider, health professional, family member | Abuse Assessment Screen–Disability;<br>Physical: Within the last year, have you been hit, slapped, kicked, pushed, shoved, or otherwise physically hurt by someone?;<br>Sexual: Within the last year, has anyone forced you to have sexual activities?; | Past 12 months | Yes; Within the last year, has anyone prevented you from using a wheelchair, cane, respirator, or other assistive devices? Within the last year, has anyone you depend on refused to help you with an important personal need, such as taking your medicine, getting to the bathroom, getting out of bed, bathing, getting dressed, or getting food or drink? |
| Rees et al., 2011 [170] | Physical IPV, rape, other forms of sexual assault and stalking | Relationship and non-relationship; partner and non-partner | Physical IPV: whether the respondent was ever badly beaten up by a spouse or romantic partner;<br>Rape: defined as sexual intercourse or penetration with a finger or object against the person's will, or by use of threat or force, or when the person was too young to understand what was happening.<br>Sexual assault: "Other than rape, were you ever sexually assaulted, where someone touched you inappropriately, or when you did not want them to?"<br>Stalking: defined as being followed or kept track of in a manner that led to feelings of serious danger. | Lifetime | No; N/A |
| Smith et al., 2008 [171] | Physical—threat, attempted, physical violence, sexual (unwanted sex) | Relationship; current and/ or former partner | Not specified;<br>Unwanted sex: "Have you ever experienced any unwanted sex by a current or former intimate partner?"<br>Physical: Threat–"Has an intimate partner ever threatened you with physical violence? This includes threatening to hit, slap, push, kick, or physically hurt you in any way."<br>Attempted physical violence–"Other than what you have already told me about, has an intimate partner ever attempted physical violence against you?"<br>Physical violence–"Has an intimate partner ever hit, slapped, pushed, kicked, or physically hurt you in any way?" | Lifetime | No; N/A |
| Sumilo et al., 2012 [138] | Not specified–"force" | Relationship; partner | Not specified; items not listed–variable is "partner has used force in the relationship" | Not specified | No; N/A |

(*Continued*)

**Table 3.** (*Continued*)

| Article | Type(s) of violence | Context/ Perpetrator(s) | Scale, measure, specific items | Time frame(s) | Violence measure(s) specific to women with disabilities (Y/N) If yes, specific items/ measures |
|---|---|---|---|---|---|
| Ward et al., 2010 [172] | Physical, emotional, sexual | Inter-personal–dating; dating partner | Not specified; participants were asked whether they ever had problems in any of their romantic relationships, such as yelling, hitting, unwanted sex, and/or taking things without permission. | Lifetime | No; N/A |
| Yoshida et al., 2011 [68] | Physical, sexual, emotional, disability-specific | Any; any | Not specified; Physical: any form of violence against your body, such as being hit, slapped, kicked, restrained, or denied food or water. Sexual: being forced, threatened, or tricked into sexual activities that range from looking or touching to rape. Emotional: being made to feel badly about yourself because of how you look or act; being controlled by someone or being told that if you do not do something you will be hurt, feeling afraid that someone will hurt you, or being bribed, isolated, or verbally attacked. | Now or ever | Yes; Part of physical violence–"Being restrained or denied food or water;" in sexual violence–"looking" included as women with disabilities may require personal care and the person assisting may look or inappropriately stare in a nonprofessional manner or have someone with them watching who should not be involved with the assistance |
| Young et al., 1997 [104] | Physical, sexual, emotional | Any; any | Not specified; Physical: any form of violence against her body, such as being hit, kicked, restrained, or deprived of food or water; Sexual: being forced, threatened, or deceived into sexual activities ranging from looking or touching to intercourse or rape; Emotional: being threatened, terrorized, corrupted, or severely rejected, isolated, ignored, or verbally attacked | Lifetime | Yes; deprived of food or water (physical abuse); looking (as a component of sexual abuse) |
| Scolese et al., 2020 [105] | Physical, sexual | Relationship; partner | Adapted WHO-MCS Physical: 6 items (items not listed) Sexual: 2 items (items not listed) | Past month | No; N/A |
| Leskosek et al., 2013 [146] | Physical, psychological, sexual, economic and restriction of freedom | Not specified; not specified | Not specified; not specified | Lifetime (since age 15), past 12 months | No; N/A |
| Grossi et al., 2018 [47] | Sexual, physical | Not specified; not specified | Sexual and Physical Abuse History Questionnaire; Insulted by partner; diminished/ humiliated you in front of other people; scared/ intimated you; slapping or throwing objects at you; pushed/ stumbled/ shook you; forced sexual relations; sexual relations by fear; humiliating/ degrading sexual relations | Not specified | No; N/A |
| Li et al., 2000 [76] | Substance abuse-related physical violence | Not specified; not specified | Not specified; whether respondent said yes/ no to having been a victim of substance abuse-related physical violence | Not specified | No; N/A |
| Zilkens et al., 2017 [173] | Physical, sexual | Any; assailant types were categorized as stranger, intimate partner, friend/acquaintance, accidental acquaintance (known <24 h), unknown (no memory), and others (e.g.employer/colleague, carers, relatives, taxi-driver). Intimate partner included current and ex-partners (including husbands, de factos and boyfriends). | Not specified; Physical assault: included a history of blunt force assault, non-fatal strangulation, being bitten and reported weapon use; Indecent assault was a non-consensual sexual act in the absence of completed or attempted penetration; Non-fatal strangulation: included manual, ligature and chokehold methods of neck pressure. Sexual assault: included non-consensual completed or attempted penetration of the patient's vagina or anus by a penis, mouth, finger or other objects or penetration of the patient's mouth by a penis. The nature of the penetration was classified as unknown if the patient suspected sexual assault but had no or incomplete recollection of the incident. | Past 10 days | No; N/A |
| Gunduz et al., 2019 [48] | Physical, sexual, emotional | Relationship; partner | Domestic Violence Against Women Scale (DVAWS)–developed for Turkish population; The scale has nine subscales as physical violence (damaging the integrity of female's body); emotional violence such as insult, contempt; economic violence; social violence and isolation of female; contempt of the sex of the female and threatening behaviors against female; sexual violence against female; negative affectivity of the female towards herself; worries and fears about husband; the use of male privilege and the lack of sharing in marriage | Not specified | No; N/A |
| Leserman et al., 1998 [106] | Physical, sexual | Any; any | Not specified; Physical: items not included Sexual: abuse was defined as either of two types of forced sexual experiences involving contact: sexual touching, and vaginal or anal intercourse (rape). Touch was defined in terms of being touched with hands, mouth, or objects on the breast or genital areas where force or threat of harm was used. Being made by force or threat of harm to touch another person's genitals with mouth or hands. Rape referred to being made by force or threat of harm to have vaginal or anal intercourse. | Ever | No; N/A |
| Martin et al., 2008 [107] | Physical, sexual | Any; any | Not specified; Physical: being pushed, hit, slapped, kicked, or physically hurt in another way Sexual: being forced to have sex or do sexual things | Lifetime (since age 18) | No; N/A |
| Cascardi et al., 1996 [84] | Physical | Relationships and family; partners and family members | CTS; items not specified | Past 12 months | No; N/A |
| Chapple et al., 2004 [174] | Physical, sexual | Not specified; not specified | No scale; specific item was "Have you been physically assaulted, beaten, molested, or otherwise a victim of violence at any time in the last 12 months?" | Past 12 months | No; N/A |
| Goodman et al., 2001 [80] | Physical, sexual | Not specified; not specified | Two sub-scales of Revised CTS; items not specified | Past 12 months | No; N/A |
| Hodgins et al., 2007 [81] | Physical, sexual | Not specified; not specified | Adapted from MacArthur Community Violence Interview; Having been a victim of: serious injury such that the individual had to seek inpatient hospital car; having been injured with a gun, knife or other object; having had an object thrown at you; having been pushed, shoved, grabbed, slapped, kicked, bit, choked or hit; having been physically forced to have sexual relations; having been threatened with a knife, gun or other weapon | Past 6 months | No; N/A |
| Teplin et al., 2005 [82] | Criminal victimization—Physical, sexual | Not specified; not specified | National Crime Victimization Survey, developed by the Bureau of Justice Statistics; items not specified | Past 12 months | No; N/A |
| Walsh et al., 2003 [83] | Any violence | Not specified; not specified | No scale specified; item was "In the last year have you been assaulted, beaten, molested or otherwise the victim of violence?" | Past 12 months | No; N/A |
| Nosek et al., 2001 [160] | Emotional, physical, sexual, disability-specific | Not specified; not specified | No scale specified; Emotional: being threatened, terrorized, corrupted, or severely rejected, isolated, ignored, or verbally attacked. Physical: any form of violence against her body, such as being hit, kicked, restrained, or deprived of food or water. Sexual: being forced, threatened, or deceived into sexual activities ranging from looking or touching to intercourse or rape. If the woman responded positively to the abuse question, she was asked to indicate the type(s) of abuse, who the perpetrator was, and at what age the abuse began and ended. | Lifetime | Yes; asked open ended questions and then developed a measure using the responses– the Abuse Assessment Screen-Disability (AAS-D). This tool adds two questions about 1) preventing the woman with a disability from using a wheelchair, respirator, or other assistive device, and 2) refusal to assist with an essential personal need such as taking medicine, going to the bathroom, getting out of bed, getting dressed, and getting food or drink. |

(*Continued*)

**Table 3.** (Continued)

| Article | Type(s) of violence | Context/ Perpetrator(s) | Scale, measure, specific items | Time frame(s) | Violence measure(s) specific to women with disabilities (Y/N) If yes, specific items/ measures |
|---|---|---|---|---|---|
| Majeed-Ariss et al., 2020 [122] | Sexual | Not specified; any of: 1st degree relative, 2nd-degree relative, Acquaintance, Authority figure, client (sex worker), friend, neighbor, partner and ex-partner, stranger, work colleague | Not specified; forensic medical examination | Not specified | No; N/A |
| Acharya 2019 [147] | Physical, Sexual, Verbal and threats of violence | Sex-work/ trafficking context; Traffickers/pad te/madrote; Owners; Boyfriend; Clients | Not specified; not specified | Not specified | No; N/A |
| Akyazi et al., 2018 [175] | Physical, verbal, sexual, economic and limiting social relations | Relationships; Spouse and family members | CTS and other forms used to study domestic violence in Turkey were used; items not specified | Not specified | No; N/A |
| Basile et al., 2016 [93] | Sexual | Not specified; not specified | National Intimate Partner and Sexual Violence Survey (NISVS); The survey includes behaviorally specific questions that assess the multiple forms of sexual violence victimization: i. Rape (completed forced, attempted forced, or alcohol- or drug-facilitated penetration), being made to penetrate a perpetrator, ii. Sexual coercion: unwanted sexual penetration that occurs after a person is pressured in a nonphysical way. iii. Unwanted sexual contact: includes experiences involving unwanted touch but not sexual penetration, such as being kissed in a sexual way, or having sexual body parts fondled or grabbed. iv. Noncontact unwanted sexual experiences: someone exposing his or her sexual body parts, flashing, or masturbating in front of the victim, or someone harassing the victim in a public place in a way that made the victim feel unsafe. | Past 12 months | No; N/A |
| Breiding et al., 2015 [94] | Rape, sexual violence other than rape, physical violence, stalking, psychological aggression, and control of reproductive or sexual health | Relationship; intimate partners, including "spouses, boyfriends, girlfriends, people you have dated, people you were seeing, or people you hooked up with" | National Intimate Partner and Sexual Violence Survey (NISVS); The survey includes behaviorally specific questions that assess the multiple forms of intimate partner victimization: i. Rape (completed or attempted forced penetration or alcohol- or drug-facilitated penetration); ii. Sexual violence other than rape which includes being made to penetrate someone, sexual coercion (non physically pressured unwanted penetration), unwanted sexual contact (e.g., kissing or fondling),and noncontact unwanted sexual experiences (e.g., being flashed or forced to view sexually explicit media); iii. Physical violence (e.g., kicked, slammed against something); iv. Stalking (e.g., receiving un-wanted e-mails, instant messages, messages through social media; having someone approach or show up in the victim's home, workplace, or school when it was unwanted); v. Psychological aggression (e.g., called names, threats to harm victim or loved ones); and vi. Control of reproductive or sexual health (refusal to use a condom; attempts to get a partner pregnant against a partner's wishes). | Past 12 months | No; N/A |
| De Waal et al., 2017 [176] | Physical, sexual and threats | Not specified; not specified | Safety Monitor, developed by the Dutch Ministry of Security and Justice; items not specified | Past 12 months | No; N/A |
| Del rio Ferres et al., 2013 [135] | Not specified, disability-specific | Relationships; intimate partner, family member, someone living with you | Screened women with Woman Abuse Screening Tool a short screening test to identify "possible cases" of intimate partner violence; then asked possible cases, "have you ever experienced a situation in which you felt abused by a family member, your intimate partner or somebody living with you?" | Past 12 months; lifetime | Yes; two items of the Abuse Assessment Screen-Disability: (a) Within the last year, has anybody you depend on refused to help you with an important personal need (related to your basic daily activities), such as taking your medicine, getting to the bathroom, getting out of bed, bathing, getting dressed, or getting food and drink? And (b) Within the last year, has anyone prevented you from using any of the technical aids you need in your daily life, such as a wheelchair, cane, respirator, or other assistive devices? Items were responded on a 4-point Likert scale ranging from "not at all" to "yes, continuously". |
| Jonas et al., 2013 [177] | Emotional, physical | Relationship; current or former partner | British Crime Survey; Has a current or previous partner ever: Prevented you from having your fair share of the household money?; Stopped you from seeing friends and (or) relatives?; Frightened you, by threatening to hurt you or someone close to you?; Pushed you, held or pinned you down or slapped you?; Kicked you, bit you, or hit you with a fist or something else, or threw something at you that hurt you?; Choked or tried to strangle you?; Threatened you with a weapon, such as a stick or a knife?; Threatened to kill you?; Used a weapon against you e.g. a knife?; Used some other kind of force against you? | Past 12 months; lifetime | No; N/A |
| Lacey et al., 2016 [116] | Physical (severe) | Relationship; spouse or romantic partner | Scale in this study not specified; single item asked was: "Were you ever badly beaten up by a spouse or a romantic partner?"; these primary data were compared to U.S. National Comorbidity Survey Replication (NCS-R) dichotomously defined severe partner violence from Conflict Tactic Scale (CTS) | Lifetime | No; N/A |
| Le et al., 2015 [178] | Physical, sexual | Any (not institution) | Juvenile Victimisation Questionnaire Revised 2 (JVQ R2)—youth self-report lifetime version; Crimes: Assault with weapon: "Sometimes people are attacked with sticks, rocks, guns, knives, or other things that would hurt. At any time in your life, did anyone hit or attack you on purpose with an object or weapon? Somewhere like: at home, at school, at a store, in a car, on the street, or anywhere else? At any time in your life, did anyone hit or attack you without using an object or weapon? At any time in your life, did someone threaten to hurt you when you thought they might really do it?" Peer and sibling assault: "Sometimes groups of kids or gangs attack people. At any time in your life, did a group of kids or a gang hit, jump, or attack you? At any time in your life, did any kid, even a brother or sister, hit you? Somewhere like: at home, at school, out playing, in a store, or anywhere else? At any time in your life, did you get scared or feel really bad because kids were calling you names, saying mean things to you, or saying they didn't want you around? At any time in your life, did a boyfriend or girlfriend or anyone you went on a date with slap or hit you? Sexual assault: At any time in your life, did a grown-up you know touch your private parts when they shouldn't have or make you touch their private parts? Or did a grown-up you know force you to have sex? At any time in your life, did you get scared or feel really bad because kids were calling you names, saying mean things to you, or saying they didn't want you around? At any time in your life, did a boyfriend or girlfriend or anyone you went on a date with slap or hit you? Sexual assault: At any time in your life, did a grown-up you know touch your private parts when they shouldn't have or make you touch their private parts? Or did a grown-up you know force you to have sex? At any time in your life, did you try to force you to have sex; that is, sexual intercourse of any kind, even if it didn't happen? | Lifetime | No; N/A |
| Macdowall et al., 2013 [123] | Sexual | Any (not institution); refer to as a current or former intimate partner; someone known to you as a family member or friend; someone known to you but not as a family member or friend; someone you didn't know | Third National Survey of Sexual Health Attitudes and Lifestyles (Natsal-3); Experience of sex against their will since the age of 13 years: "Has anyone tried to make you have sex with them, against your will?," if yes: "Has anyone actually made you have sex with them, against your will?" | Lifetime (since age 13) | No; N/A |
| New, 2019 [59] | Sexual | Not specified; not specified | Not specified; "Have you ever been sexually abused?" | Lifetime | No; N/A |
| Olofsson et al., 2015 [95] | Psychological, physical | Not specified; not specified | Not specified; "Have you been verbally offended during the past 12 months?," "Have you been exposed to any threats of violence or other threats that scared you during the past 12 months," "Have you been exposed to physical violence during the past 12 months?" | Past 12 months | No; N/A |

(Continued)

**Table 3.** (Continued)

| Article | Type(s) of violence | Context/ Perpetrator(s) | Scale, measure, specific items | Time frame(s) | Violence measure(s) specific to women with disabilities (Y/N) If yes, specific items/ measures |
|---|---|---|---|---|---|
| Salahi et al., 2018 [152] | Physical, psychological, sexual | Relationship; intimate partner | Women Abuse Screening Tool [WAST] and CTS 2; WAST–two screening questions on tension in relationship and difficulty resolving problems, then items on physical, sexual and psychological violence.<br>Items: Do arguments ever result in result in you feeling put down or bad about yourself?; Do arguments ever result in hitting, kicking or pushing? Do you feel frightened by what your partner says or does? Has your partner ever abused you physically? Has your partner ever abused you emotionally? Has your partner ever abused you sexually? | Lifetime | No; N/A |
| Owens 2007 [179] | Psychological, physical, sexual | Relationship; intimate partner | Items adapted from Women's Experiences with Battering Scale, Abuse Assessment Screen and the CTS;<br>Psychological: "I try not to "rock the boat" because I am afraid of what my partner might do"; "I feel owned and controlled by my partner"; and "My partner can scare me without laying a hand on me"<br>Physical: "As an adult, has a romantic partner, spouse or ex-partner ever hit, slapped, kicked or otherwise physically hurt you? Has this happened in the last 12 months?"<br>Sexual: "As an adult, has a romantic partner or ex-partner ever forced you to have an unwanted sexual act? Has this happened in the last 12 months?" | Lifetime; past 12 months | No; N/A |
| Coker et al., 2002 [180] | Physical, sexual, psychological | Relationship; intimate partner, defined as current or former spouse, or live-in boyfriend or girlfriend | 12-item CTS to measure physical aggression; four-item forced sex questions from the National Women's Study, and the 13-item Power & Control Scale to measure psychological abuse by a partner:<br>Verbal abuse: Your partner: "shouts or swears at you"; "provokes arguments"; "calls you names or puts you down in front of others"; "has a hard time seeing things from your point of view"; "is jealous or possessive"<br>Power and control: Your partner: "frightens you"; "makes you feel inadequate"; "prevents you from knowing about or having access to the family income even when you ask"; "prevents you from working outside the home"; and "insists on changing residence even when you don't need or want to"; "tries to limit your contact with family or friends" and "insists on knowing who you are with at all times" | Not specified | No; N/A |
| Dammeyer et al., 2018 [181] | Physical, sexual, psychological/ economic | Not specified; not specified | Survey of Health, Impairment, and Living Conditions in Denmark;<br>Non-physical: "In the past year, has someone: (1) Threatened you with violence (2) Humiliated, degraded or ridiculed you, or constantly criticized you; (3) Prevented you from accessing your money or bank account, blocked your bank card, or forced you to pay a sum of money or act as guarantor?"<br>Physical: "In the past year, has someone: (1) Shaken you, pushed you or pulled your hair; (2) Hit or kicked you?"<br>Sexual: "In the past year, has someone forced you to: (1) Kiss or hug; (2) Have sexual intercourse or engage in other sexual acts?" | Past 12 months | No; N/A |
| Gibbs et al., 2017 [154] | Physical, sexual | Relationship; intimate partner | WHO MCS adapted;<br>Physical IPV: 4 items, i.e "How many times has your current or any previous boyfriend, husband, or partner threatened to use or actually used a gun, knife, or other weapon against you?"; follow-up—"Has your current or any other boyfriend, husband, or partner done any of these things in the last 12 months?"<br>Sexual IPV: 4 items, not listed | Lifetime; past 12 months | No; N/A |
| Khalifeh et al., 2015 [49] | Physical, sexual | Any (not institution); intimate partner, family member, stranger | Not specified; not specified | Past 12 months | No; N/A |
| Milberger et al., 2003 [153] | Physical, sexual, disability-specific | Any | Not specified;<br>Specific items: 1. Since you were 18 years old, have you been hit, slapped, kicked pushed, shoved or otherwise physically hurt by someone?<br>2. Since you were 18 years old, has anyone forced you to have sexual activities?<br>3. Since you were 18 years old, has anyone prevented you from using a wheelchair, cane, respirator, or other assistive devices?<br>4. Since you were 18 years old, has anyone you depend on refused to help you with an important personal need such as taking your medicine, getting to the bathroom, getting out of bed, bathing, getting dressed or getting food or drink or threatened not to help you with these personal needs? | Lifetime | Yes;<br>i. Since you were 18 years old, has anyone prevented you from using a wheelchair, cane, respirator, or other assistive devices?<br>ii. Since you were 18 years old, has anyone you depend on refused to help you with an important personal need such as taking your medicine, getting to the bathroom, getting out of bed, bathing, getting dressed or getting food or drink or threatened not to help you with these personal needs? |
| Nannini 2006 [118] | Sexual | Any; partner, ex-partner, caregiver/ service provider, family member, friend, stranger | Not specified; not specified | Not specified | No; N/A |
| Weiner et al., 2013 [50] | Bullying | School | Olweus Bullying Questionnaire; "How often have you been bullied at school in the past couple of months?" | Past couple of months | No; N/A |
| Nunes de Oliviera et al., 2013 [77] | Physical | Any; intimate partner, parents, relatives, acquaintances, strangers, health professionals or others | Not specified; experience any kind of physical aggression, including being hit, spanked, or beaten in their lifetime. | Lifetime | No; N/A |
| Ferraro et al., 2017 [117] | Physical, psychological, sexual | Relationship; husband/ partner | WHO MCS survey; 7 types of physical abuse, 4 types of psychological abuse, and 3 types of sexual abuse; items not listed | Past 12 months | No/ N/A |
| Gilchrist et al., 2012 [182] | Sexual, physical, emotional | Relationship; current or most recent partner | Composite Abuse Scale; items not specified | Past 12 months | No/ N/A |
| Golding 1996 [108] | Sexual | Any | Not specified; two surveys asked about sexual violence using different items: "In your lifetime, has anyone ever tried to pressure or force you to have sexual contact?" By sexual contact I mean their touching your sexual parts, your touching their sexual parts, or sexual intercourse."; or "One event which people often report as a serious one in their lives is that of being sexually assaulted. Have you ever been in a situation in which you were pressured into doing more sexually than you wanted to do, that is, a situation in which someone pressured you against your will into forced contact with the sexual parts of your body or their body?" | Lifetime | No/ N/A |
| Siqueira-Campos et al., 2019 [51] | Physical, sexual | Not specified; not specified | Scale not specified; Specific items were: "Have you ever suffered physical abuse?" and "Have you ever suffered sexual abuse?" | Lifetime | No; N/A |
| Sturup et al., 2011 [52] | Physical, sexual | Not specified; not specified | Scale not specified; Specific items were: "Have you been subjected to violence resulting in visible injuries during the last twelve months?" and ''Have you been subjected to violence that required medical attention during the last twelve months?" | Past 12 months | No; N/A |
| Thompson et al., 2019 [53] | Sexual | Not specified; not specified | ASHR Health and Relationships Questionnaire; i. unwanted sexual experience and ii. being forced/frightened into doing something sexually | Lifetime | No; N/A |
| Walker et al., 1997 [54] | Sexual, physical, emotional | Any; boyfriend, husband, acquaintance, stranger | Adapted items from Child Maltreatment Interview; 1. After you were 17 years old did anyone ever force you, threaten you, or take advantage of a time when you had used drugs or alcohol to have (vaginal), anal, or oral intercourse but did not succeed? 2. After you were 17 years old did anyone ever force you, threaten you, or take advantage of a time when you had used drugs or alcohol to have (vaginal), anal or oral intercourse, with any amount of penetration? 3. Was the person who assaulted you ever a boyfriend or husband? 4. After you were 17 years old did anyone ever intentionally injure you in any serious way so that you received bruises, cuts, burns or broken bones, or any injury that led you to be seen by a doctor or go to a hospital? 5. Was the person who assaulted you ever a boyfriend or husband? | Lifetime | No; N/A |

(Continued)

**Table 3.** (Continued)

| Article | Type(s) of violence | Context/ Perpetrator(s) | Scale, measure, specific items | Time frame(s) | Violence measure(s) specific to women with disabilities (Y/N) If yes, specific items/ measures |
|---|---|---|---|---|---|
| Afe et al., 2017 [183] | Physical, verbal, sexual | Relationship; intimate partner | WHO MCS survey adapted based on focus group discussion;<br>Physical: Within the last 12 months, Have you ever been slapped, beaten, hit, kicked, pushed or physically assaulted in any way by your intimate partner/husband? Were you ever injured/hospitalized/wounded as a result of the violence? Did you get hurt? Anyone who witnessed the incident? How frequent was it?<br>Verbal: Within the last 12 months, Have you been abused, shouted at insulted/ threatened/cursed/disrespected/disgraced by your intimate partner/husband? Does he make you feel small in front of others? Are you fearful of your husband? Does he make you feel sad by his attitude to you?<br>Sexual: Within the last 12 months have you ever been forced, hit, threatened/abused/ grabbed/injured in order to have sexual intercourse by your intimate partner/husband? Has he ever forced you to have sexual acts that you never liked? | Past 12 months | No/ N/A |
| Anderson et al., 2016 [78] | Physical, sexual, emotional | Any; a partner, a family member (other than a partner), someone else I knew (other than a partner of family member), a stranger | Modified version of modified version of the Office for National Statistics (ONS) Crime Survey for England and Wales (CSEW) questionnaire:<br>Sexual:<br>Indecent exposure: "Since you were 16, has anyone ever indecently exposed themselves to you (i.e. flashing) in a way that caused you fear, alarm or distress?"<br>Sexual touching: "Since you were 16, has anyone ever touched you in a sexual way (e.g. touching, grabbing, kissing or fondling) when you did not want it?"<br>Sexual intercourse: "Since you were 16, has anyone ever forced you to have sexual intercourse, when you were not capable of consent or when you made it clear you did not want to? By sexual intercourse we mean vaginal, anal or oral penetration."<br>Attempted sexual intercourse: "Apart from anything else you have already mentioned, since you were age 16 has anyone ever ATTEMPTED to force you to have sexual intercourse when you were not capable of consent or when you made it clear you did not want to?';<br>Emotional: Since you were 16 has a partner or ex-partner (/member of your family) EVER done any of the things listed below? Prevented you from having your fair share of the household money; Stopped you from seeing friends and relatives; Repeatedly belittled you to the extent that you felt worthless; Since you were 16 has a partner or ex-partner (/member of your family) EVER threatened you in any way?;<br>Physical: Since you were 16 has a partner or ex-partner (/member of your family) EVER used a force on you? Have you EVER been injured (even if only slightly) as a result of the force used on you by a partner (/member of your family)? | Lifetime; past 12 months | No; N/A |
| Beydoun et al., 2017 [127] | Physical | Relationship; spouse or partner | Physical IPV was determined using International Classification of Disease, 9th Revision, Clinical Modification (ICD-9-CM) external cause of injury code E967.3 (battering by spouse or partner) | Not specified | No; N/A |
| Du Mont et al., 2014 [184] | Physical, sexual | Relationship; current or former partner | Modified version of the 10 item CTS; items on emotional and financial abuse were originally created for use on Statistics Canada's 1993 Violence Against Women Survey, specific items not included | Past 5 years | No; N/A |
| Gil-Llario et al., 2019 [85] | Sexual | Not specified; not specified | Sexual abuse reported by participants with intellectual disabilities or by professionals; items not included | Not specified | No; N/A |
| Gold et al., 2012 [136] | Not specified | Relationship; intimate partner | United States National Violent Death Reporting System (NVDRS); defined IPV as either known history of interpersonal violence within the last month (captured in the Death Reporting System) or variable which is coded positive if friction or conflict with a current or former intimate partner appears to have played a precipitating role in the suicide | Past month | No; N/A |
| Gonzalez Cases et al., 2014 [69] | Physical, psychological, sexual | Relationship; current or former intimate partner | Intimate Partner Violence towards Women Questionnaire and CTS; items not specified | Lifetime and past 12 months | No; N/A |
| Helfrich et al., 2008 [137] | Not specified | Relationship; intimate partner | Not specified; not specified | Not specified | No; N/A |
| Khalifeh et al., 2015 [55] | Emotional, physical or sexual | Any; partner (boyfriend or girlfriend); husband, wife or civil partner) or Family member other than partner; strangers or acquaintances | Crime Survey for England and Wales (CSEW) questionnaire;<br>Emotional abuse: perpetrator did any of the following:(a) Prevented them from having fair share of money; (b) Stopped them from seeing friends or relatives; (c) Repeatedly belittled them so they felt worthless; (d) Threatened to hurt them or someone close to them; (e) Threatened them with a weapon or threatened to kill them.<br>Physical violence: perpetrator did any of the following: (a) Pushed them, held them down or slapped them; (b) Kicked, bit or hit them, or threw something at them; (c) Choked or tried to strangle them; (d) Used some other kind of force against them.<br>Sexual violence: perpetrator did any of the following in a way that caused fear, alarm or distress: (a) Indecently exposed themselves to them; (b) Touched them sexually when they did not want it (e.g. groping, touching of breasts or bottom, unwanted kissing); (c) Forced them to have sexual intercourse, or to take part in some other sexual act, when they made it clear that they did not agree or when they were not capable of consent (serious sexual assault). | Lifetime (since age 16) and past year | No; N/A |
| Kmett et al., 2018 [86] | Sexual | Any; partners, friends, family members, strangers | MacArthur Violence Risk Assessment Study; "Did anyone ever bother you sexually or try to have sex with you against your will?"; if yes, asked specifics of the event (i.e. intercourse, attempted intercourse, sodomy, inappropriate touching, oral sex, hugging or kissing, and other) | Lifetime | No; N/A |
| Lundberg et al., 2015 [56] | Sexual | Any; Current husband/cohabiting partner; stranger, boyfriend, and male friend of family | WHO MCS/ DHS DV module Items: (1)"Did your current husband/cohabiting partner or any previous husband/cohabiting partner ever physically force you to have sexual intercourse when you did not want to?"(2)"Did you ever have sexual intercourse you did not want to because you were afraid of what your current (or previous) husband/ cohabiting partner might do?" and (3)"Did your current (or previous) husband/ cohabiting partner ever force you to do something sexual that you found degrading or humiliating?" We defined sexual violence by non partner as a positive answer to the following question, referring to past 12 months before admission: "Since age 15, has anyone (other than your husband/cohabiting partner) ever forced you to have sex or to perform a sexual act when you did not want to?" | Lifetime (since age 15); past year | No; N/A |
| McPherson et al., 2007 [43] | Physical, sexual | Relationship; intimate partner | CTS; Over the past 12 months did someone you were romantically involved with ever push, grab, or slap you? Did they ever hit you with a fist or an object, kick you, or beat you up? Did they ever choke you, tie you up, or physically restrain you? Did they ever force sexual activity that you didn't want to happen? | Past 12 months | No; N/A |
| Meekers et al., 2013 [57] | Physical, psychological, sexual | Relationship; current partner | CTS;<br>Physical: How often (one time, a few times, or often) in the last 12 months their partner had pushed or pinched them, beaten or kicked them, beaten them with an object, or tried to strangle or burn them.<br>Sexual: Whether their partner forced them to have sexual relations against their will during the year before the survey.<br>Psychological: How often their partner, in the past 12 months, had accused them of being unfaithful, had been jealous after she talked with a man, had attempted to limit her contact with her family, had humiliated or insulted her, had threatened to abandon her, had threatened to take away her children, had threatened to take away economic support, or had broken things inside the house. | Past 12 months | No; N/A |
| Nguyen et al., 2017 [58] | Sexual | Not specified; not specified | No scale; one item: "Was there any time in your lifetime you were forced to have sex when you did not want to?" | Lifetime | No; N/A |

*(Continued)*

**Table 3.** (Continued)

| Article | Type(s) of violence | Context/ Perpetrator(s) | Scale, measure, specific items | Time frame(s) | Violence measure(s) specific to women with disabilities (Y/N) If yes, specific items/ measures |
|---|---|---|---|---|---|
| Racic et al., 2006 [185] | Psychological, physical, neglect, financial exploitation | Not specified | The Hwalek-Sengstock Elder Abuse Screening Test (EAST); specific items not listed | Not specified | No; N/A |
| Riley et al., 2014 [186] | Emotional, physical, sexual | Any; partner, friend, stranger, acquaintance, neighbor, family member | Severity of Violence Against Women Scale; Emotional: experienced threats, harassment, cruelty, aggression, harm to another person, or loss of property from malicious intent) Physical: being hit, slapped, kicked, bitten, choked, shot, stabbed, or struck with an object Sexual: forced to have sex of any kind | Past 6 months | No; N/A |
| Santaularia et al., 2014 [109] | Sexual | Any; not specified | Behavioral Risk Factor Surveillance Survey (BRFSS) sexual violence module: Specific item: "has anyone ever had sex with you after you said or showed that you didn't want them to or without your consent?", defined as "things like putting anything into your vagina, anus, or mouth or making you do these things to them after you said or showed that you didn't want to. It includes times when you were unable to consent, for example, you were drunk or asleep, or you thought you would be hurt or punished if you refused." | Lifetime | No; N/A |
| Schofield et al., 2013 [187] | Elder abuse—vulnerability, coercion, dependence, and dejection | Any; not specified | 12-item Vulnerability to Abuse Screening Scale (VASS)– 4 sub-scales Vulnerability: Has anyone close to you tried to hurt you or harm you recently? Has anyone close to you called you names or put you down or made you feel bad recently? Are you afraid of anyone in your family? Coercion: Does someone in your family make you stay in bed or tell you you're sick when you know you're not? Has anyone forced you to do things you didn't want to do? Has anyone taken things that belong to you without your OK? Dependence: Can you take your own medication and get around by yourself? Do you trust most of the people in your family? Do you have enough privacy at home? Dejection: Are you sad or lonely often? Do you feel uncomfortable with anyone in your family? Do you feel that nobody wants you around? | Not specified | No; N/A |
| Shah et al., 2018 [188] | Physical, sexual | Relationship; intimate partner | Modified version of measures from the National Intimate Partner and Sexual Violence Survey; "Has a romantic or sexual partner ever made threats to physically harm you?" "Has a romantic or sexual partner ever shot at, stabbed, struck, kicked, beaten, punched, slapped, or otherwise physically harmed you?" "Has a romantic or sexual partner ever forced or pressured you to engage in unwanted sexual activity that you did not want to do? Unwanted sexual activity includes vaginal, oral, or anal intercourse or inserting an object or fingers into your anus or vagina." | Lifetime | No; N/A |
| Anderson et al., 1993 [124] | Sexual | Not specified; not specified | Questions asked about 12 different types of sexual abuse; items not listed | Lifetime | No; N/A |
| Institute of Statistics, et al., 2018 [129] | Physical | Relationship; Husband or boyfriend | DHS DV Module; Women were asked if they had ever had a husband or a boyfriend that slapped her, hit her with his fists, kicked her, or did anything to hurt her physically. | Lifetime; past 12 months | No; N/A |
| National Institute of Statistics et al., 2015 [189] | Physical, sexual, emotional, economic | Relationship and other; Current or former partners, family members, teacher, employer, someone at work, police, soldier, stranger | DHS DV Module; Does (did) your (last) husband/partner ever: Physical: (a) Push you, shake you, or throw something at you? (b) Slap you? (c) Twist your arm or pull your hair? (d) Punch you with his/her fist or with something that could hurt you? (e) Kick you, drag you, or beat you up? (f) Try to choke you or burn you on purpose? (g) Threaten or attack you with a knife, gun, or any other weapon? Sexual: (h) Physically force you to have sexual intercourse with him/her even when you did not want to? (i) Physically force you to perform any other sexual acts you did not want to? (j) Force you with threats or in any other way to perform sexual acts you did not want to? Emotional: (a) Say or do something to humiliate you in front of others? (b) Threaten to hurt or harm you or someone close to you? (c) Insult you or make you feel bad about yourself? Economic: Does/did not give her money to cover household expenses; Does/did not trust her with money | Last 12 months and since age 15 for physical violence; lifetime and last 12 months for sexual violence | No; N/A |
| Institut National de la Statistique et al., 2012 [190] | Physical, emotional and sexual | Relationship and other; Physical: Current husband Former husband Current boyfriend Father/stepfather Mother/stepmother Sister/brother Daughter/son Other relative Mother-in-law Father-in-law Other in-law Teacher, Employer/Person at work, Police/soldier Stranger | DHS DV Module; Physical: Does (did) your (last) husband/partner ever: (a) Push you, shake you, or throw something at you? (b) Slap you? (c) Twist your arm or pull your hair? (d) Punch you with his/her fist or with something that could hurt you? (e) Kick you, drag you, or beat you up? (f) Try to choke you or burn you on purpose? (g) Threaten or attack you with a knife, gun, or any other weapon? Sexual: Does (did) your (last) husband/partner ever: (h) Physically force you to have sexual intercourse even when you did not want to? (i) force you to perform any other sexual acts you did not want to? Emotional: Does (did) your (last) husband/partner ever: (j) Say or do something to humiliate you in front of others? (k) Threaten to hurt or harm you or someone close to you? (l) Insult you or demean you? | Last 12 months, since the age of 15, lifetime and last 12 months for sexual; violence during pregnancy | No; N/A |
| Institut National de la Statistique et al., 2015 [191] | Physical, emotional and sexual | Relationship and other; Current husband/partner Former husband/partner Current boyfriend/ Former boyfriend; Father/stepfather Mother/stepmother Sister/brother Daughter/son Other relative; Mother-in-law; Other in-law; Teacher/ Employer/ someone at work; Police/soldier Other | DHS DV Module; Physical: Does (did) your (last) husband/partner ever: (a) Push you, shake you, or throw something at you? (b) Slap you? (c) Twist your arm or pull your hair? (d) Punch you with his/her fist or with something that could hurt you? (e) Kick you, drag you, or beat you up? (f) Try to choke you or burn you on purpose? (g) Threaten or attack you with a knife, gun, or any other weapon? Sexual: Does (did) your (last) husband/partner ever: (h) Physically force you to have sexual intercourse even when you did not want to? (i) Force you to perform any other sexual acts you did not want to? Emotional: Does (did) your (last) husband/partner ever: (j) Say or do something to humiliate you in front of others? (k) Threaten to hurt or harm you or someone close to you? (l) Insult you or demean you? | Last 12 months, since the age of 15 for physical and emotional, lifetime and last 12 months for sexual; violence during pregnancy | No; N/A |

(Continued)

**Table 3.** (Continued)

| Article | Type(s) of violence | Context/ Perpetrator(s) | Scale, measure, specific items | Time frame(s) | Violence measure(s) specific to women with disabilities (Y/N) If yes, specific items/ measures |
|---------|--------------------|------------------------|--------------------------------|---------------|---------|
| Institut Haïtien de l'Enfance (IHE) et al., 2018 [192] | Physical, emotional and sexual | Relationships and strangers; Physical: Current husband Former husband Current boyfriend Father/stepfather Mother/stepmother Sister/brother Daughter/son Other relative Mother-in-law Father-in-law Other in-law Teacher, Employer/Person at work, Police/soldier Stranger | DHS DV Module; Physical: Does (did) your (last) husband/partner ever: (a) Push you, shake you, or throw something at you? (b) Slap you? (c) Twist your arm or pull your hair? (d) Punch you with his/her fist or with something that could hurt you? (e) Kick you, drag you, or beat you up? (f) Try to choke or burn you on purpose? (g) Threaten or attack you with a knife, gun, or any other weapon? Sexual: Does (did) your (last) husband/partner ever: (h) Physically force you to have sexual intercourse even when you did not want to? (i) Force you to perform any other sexual acts you did not want to? Emotional: Does (did) your (last) husband/partner ever: (j) Say or do something to humiliate you in front of others? (k) Threaten to hurt or harm you or someone close to you? (l) Insult you or demean you? | Last 12 months, since the age of 15 for physical and emotional, lifetime and last 12 months for sexual | No; N/A |
| Institut National de la Statistique (INSTAT) et al., 2019 [193] | Physical, emotional and sexual | Relationship and strangers; Physical: Current husband Former husband Current boyfriend Father/stepfather Mother/stepmother Sister/brother Daughter/son Other relative Mother-in-law Father-in-law Other in-law Teacher, Employer/Person at work, Police/soldier Stranger | DHS DV Module; Physical: Does (did) your (last) husband/partner ever: (a) Push you, shake you, or throw something at you? (b) Slap you? (c) Twist your arm or pull your hair? (d) Punch you with his/her fist or with something that could hurt you? (e) Kick you, drag you, or beat you up? (f) Try to choke you or burn you on purpose? (g) Threaten or attack you with a knife, gun, or any other weapon? Sexual: Does (did) your (last) husband/partner ever: (h) Physically force you to have sexual intercourse even when you did not want to? (i) force you to perform any other sexual acts you did not want to? Emotional: Does (did) your (last) husband/partner ever: (j) Say or do something to humiliate you in front of others? (k) Threaten to hurt or harm you or someone close to you? (l) Insult you or demean you? | Last 12 months, since the age of 15 for physical or emotional violence, lifetime and last 12 months for sexual; violence during pregnancy | No; N/A |
| Agence Nationale de la Statistique et de la Démographie (ANSD) et al., 2019 [194] | Physical, sexual, emotional | Relationship and other; Current or last husband/partner, mother/ father's wife, father/ husband of mother, sister/ brother, daughter/ son, other parent, current boyfriend, previous boyfriend, mother in law, father in law, other in-law, friend/ acquaintance, friend of the family, teacher/ employer/ someone at work, police/soldier, priest/ someone religious, unknown | DHS DV module: Emotional IPV: Does (did) your (last) husband/partner ever: (a) Say or do something to humiliate you in front of others? (b) Threaten to hurt or harm you or someone close to you? (c) Insult you or demean you? Physical IPV: Does (did) your (last) husband/partner ever: (d) Push you, shake you, or throw something at you? (e) Slap you? (f) Twist your arm or pull your hair? (g) Punch you with his/her fist or with something that could hurt you? (h) Kick you, drag you, or beat you up? (i) Try to choke you or burn you on purpose? Cause deep wounds, broken bones, broken teeth or other serious injuries (j) Threaten or attack you with a knife, gun, or any other weapon? Sexual: (k) Physically force you to have sexual intercourse with him/her even when you did not want to? (l) Physically force you to perform any other sexual acts you did not want to? (m) Force you with threats or in any other way to perform sexual acts you did not want to? Non-partner physical violence: Since you were 15, has anyone other than (your/a) (husband/partner) beat, slapped, kicked or kicked you something to hurt you physically? Non-partner sexual violence: At any time in your life, whether you were a child or an adult, did anyone force you to have sex or other sexual acts against your will? In the past 12 months, has anyone other than (your/a) (husband/partner) physically force you to have sex against your will? | Last 12 months, since the age of 15 for physical and emotional, lifetime and past 12 months for sexual | No; N/A |
| National Department of Health et al., 2016 [195] | Physical, sexual, emotional/ control | Relationship and any relationship; any partner and non-partners | DHS Domestic Violence module; Control exercised by partners: Is jealous or angry if she talks to other men; Frequently accuses her of being unfaithful; Does not permit her to meet her female friends; Tries to limit her contact with her family; Insists on knowing where she is at all times Physical violence: Pushed her, shook her, or threw something at her; Kicked her, dragged her, or beat her up; Tried to choke her or burn her on purpose; Threatened her or attacked her with a knife, gun, or other weapon Sexual violence: Physically forced her to have sexual intercourse with him when she did not want to; Physically forced her to perform any other sexual acts she did not want to; Forced her with threats or in any other way to perform sexual acts she did not want to Emotional violence: Said or did something to humiliate her in front of others; Threatened to hurt or harm her or someone she cared about; Insulted her or made her feel bad about herself | Lifetime, past 12 months for IPV; non-partner after age 15 and past 12 months; violence during pregnancy | No; N/A |
| Uganda Bureau of Statistics et al., 2018 [196] | Physical, sexual, emotional | Any; intimate partner Current partner Former partner Current boyfriend/girlfriend Former boyfriend/girlfriend Father/stepfather Mother/stepmother Sister/brother Daughter/son Other relative Mother-in-law Father-in-law Other in-law Teacher Employer/someone at work Police/soldier; other | DHS DV Module; Physical IPV: push you, shake you, or throw something at you; slap you; twist your arm or pull your hair; punch you with his/her fist or with something that could hurt you; kick you, drag you, or beat you up; try to choke you or burn you on purpose; or threaten or attack you with a knife, gun, or any other weapon Sexual IPV: physically force you to have sexual intercourse with him/her even when you did not want to, physically force you to perform any other sexual acts you did not want to, or force you with threats or in any other way to perform sexual acts you did not want to Emotional spousal violence: say or do something to humiliate you in front of others, threaten to hurt or harm you or someone close to you, or insult you or make you feel bad about yourself | Last 12 months, since the age of 15 for physical and emotional, lifetime and past 12 months for sexual; violence during pregnancy | No; N/A |

(*Continued*)

**Table 3.** (Continued)

| Article | Type(s) of violence | Context/ Perpetrator(s) | Scale, measure, specific items | Time frame(s) | Violence measure(s) specific to women with disabilities (Y/N) If yes, specific items/ measures |
|---|---|---|---|---|---|
| General Directorate of Statistics et al., 2018 [197] | Physical, sexual emotional | Relationship; Physical violence: Current or former partners, husbands, boyfriends; family members, teacher, own friend/ acquaintance; Sexual violence: Current or former partners, husbands, boyfriends; family members, teacher, own friend/ acquaintance, employer, someone at work, police, soldier, stranger, other | DHS DV Module:<br>Physical spousal violence: push you, shake you, or throw something at you; slap you; twist your arm or pull your hair; punch you with his/her fist or with something that could hurt you; kick you, drag you, or beat you up; try to choke you or burn you on purpose; or threaten or attack you with a knife, gun, or any other weapon<br>Sexual spousal violence: physically force you to have sexual intercourse with him even when you did not want to; physically force you to perform any other sexual acts you did not want to; force you with threats or in any other way to perform sexual acts you did not want to<br>Emotional spousal violence: say or do something to humiliate you in front of others; threaten to hurt or harm you or someone close to you; insult you or make you feel bad about yourself | Physical and emotional IPV since age 15 and last 12 months; lifetime and past 12 months for sexual; violence during pregnancy | No; N/A |
| The Gambia Bureau of Statistics et al., 2014 [198] | Physical, sexual, psychological | Any; Physical: Former or current partner, family member, teacher, other; Sexual: former or current partner, family member, teacher, employer/someone at work, police/ soldier, priest/religious leader, stranger other | DHS DV Module:<br>Physical IPV: Does (did) your (last) husband/partner ever:<br>Push you, shake you, or throw something at you; Slap you; Twist your arm or pull your hair?; Punch you with his fist or with something that could hurt you?; Kick you, drag you, or beat you up?; Try to choke you or burn you on purpose?; Threaten or attack you with a knife, gun, or any other weapon?<br>Sexual IPV:<br>Physically force you to have sexual intercourse with him when you did not want to?; Physically force you to perform any other sexual acts you did not want to?; Force you with threats or in any other way to perform sexual acts you did not want to?<br>Emotional IPV:<br>Say or do something to humiliate you in front of others?; Threaten to hurt or harm you or someone close to you?; Insult you or make you feel bad about yourself | Physical and emotional IPV since age 15 and last 12 months; lifetime and past 12 months for sexual | No; N/A |
| National Institute of Population Studies et al., 2019 [199] | Physical, sexual, psychological | Any; Physical: Current husband Former husband Current boyfriend Father/stepfather Mother/stepmother Sister/brother Daughter/son Other relative Mother-in-law Father-in-law Other in-law Teacher, Other; Sexual: Current husband Former husband Current/former boyfriend Father/stepfather Other relative Police/soldier Stranger | DHS DV Module:<br>Physical IPV: push you, shake you, or throw something at you; slap you; twist your arm or pull your hair; punch you with his fist or with something that could hurt you; kick you, drag you, or beat you up; try to choke you or burn you on purpose; or threaten or attack you with a knife, gun, or any other weapon<br>Sexual IPV: physically force you to have sexual intercourse with him even when you did not want to, physically force you to perform any other sexual acts you did not want to, or force you with threats or in any other way to perform sexual acts you did not want to<br>Emotional IPV: say or do something to humiliate you in front of others, threaten to hurt or harm you or someone close to you, or insult you or make you feel bad about yourself | Physical and emotional IPV since age 15 and last 12 months; lifetime and past 12 months for sexual; violence during pregnancy | No; N/A |
| Rand et al., 2007 [200] | Physical, sexual | Any; intimate partner, family member/ relative; well-known or casual acquaintance; strange | National Crime Victimization Survey; items not included | Not indicated | No; N/A |
| Carlile 1991 [70] | Not specified | Relationship; Husband | Not specified; not included | Not indicated | No; N/A |
| Post et al., 1980 [130] | Physical | Relationship; spouse | Not specified; defined domestic violence as "significant physical harm inflicted by a partner including slapping, kicking, punching, biting, or attacking with a weapon" | Not indicated | No; N/A |
| Sansone et al., 2007 [71] | Threats; acts | Relationship; intimate partner | Severity of Violence Against Women Scale; eliminated questions on sexual aggression. Threats– 19 items; Acts– 21 items. Items not listed | Lifetime | No; N/A |
| Zanarini et al., 1999 [201] | Physical, sexual | Relationship; partner | Abuse History Interview; items not specified | Lifetime | No; N/A |
| Bengtsson-Tops et al., 2012 [79] | Threats, physical, sexual | Any; stranger, acquaintance, ex-partner, family member, partner, relative, service staff | Composite Abuse Scale;<br>Threats: threats of being injured, threats of being killed<br>Physical victimization: moderate (being smacked, shaken, pushed, punched, kicked or bitten); resulting in physical injury; use of weapons, knife<br>General sexual victimization: groped/ forced to grope another person; forced to look at/ participate in a porno; oral, vaginal or anal rape | Lifetime; past 12 months | No; N/A |
| Ford 2008 [72] | Physical, sexual | Any; domestic violence included, other perpetrators not specified | Traumatic Events Screening Inventory; items included "domestic violence," "traumatic physical assault," "sexual trauma" | Lifetime | No; N/A |
| Friedman et al., 2011 [44] | Threats, physical, sexual | Relationship; intimate partner | Not stated; not included | Not included | No; N/A |
| Goodman et al., 1995 [202] | Physical, sexual | Relationship and stranger; intimate partner, stranger and other | For physical violence–CTS; throwing an object, pushing, shoving, slapping, kicking, biting, hitting with a fist or object, beating, choking, threats with or use of a knife or a gun<br>For sexual violence–Russell's semi-structured interview, series of open-ended questions to assess acts ranging from unwanted touching to rape | Lifetime | No; N/A |
| Leithner et al., 2009 [203] | Physical, sexual, psychological | Any; included parents, other relatives, partner, strangers, | Scale not specified;<br>Physical violence: Physical violence was classified as "minor" or "severe." Minor physical abuse included the perpetrator throwing an object at the woman or pushing, grabbing, shoving, or slapping her. Severe physical violence included repeated kicks or bites, being hit with a fist or an object, as well as the perpetrator choking her or threatening her with a knife or other object.<br>Sexual violence: Sexual violence included all experiences of unwanted sexual advances of someone in authority, sexual harassment or assault, attempted rape, or experienced rape. | Lifetime | No; N/A |
| Lipschitz et al., 2009 [204] | Physical, sexual | Any; partner or stranger | Traumatic Events Questionnaire; items not included, definition of physical is any beating by partner or assault by stranger; adult sexual assault | Lifetime | No; N/A |
| Morgan et al., 2010 [205] | Physical, sexual, emotional | Relationship; intimate partner | Scale not specified;<br>Physical IPV:<br>Grabbed or shoved you; Punched you on body/arms/legs; Punched you in the face; Forced you to have sex; Physically violent to you in other way; Kicked you on the floor; Choked or held hand over your mouth; Used weapon or object to hurt you; Tried to strangle, burn or drown you<br>Threatening behaviour by partner:<br>Punched, kicked or threw things; Threatened you with fist, hand or foot; Threatened you with object or weapon; Threatened to kill you<br>Controlling behaviour by partner:<br>Shouted, screamed or swore at you; Criticised you; Checked up on your movements; Restricted your social life; Tried to control you in any other way not physical violence; Kept you short of money; Locked you in the house<br>Items for sexual violence not included | Lifetime; past 12 months; during pregnancy | No; N/A |

(*Continued*)

**Table 3.** (Continued)

| Article | Type(s) of violence | Context/ Perpetrator(s) | Scale, measure, specific items | Time frame(s) | Violence measure(s) specific to women with disabilities (Y/N) If yes, specific items/ measures |
|---|---|---|---|---|---|
| Surrey et al., 1990 [206] | Physical, sexual | Any; including partner, family member, friend | Life Experiences Questionnaire;<br>Physical: defined as having been "physically hurt or attacked by someone-such as husband, parent, another family member, or friend (for example, have you ever been kicked, bitten, pushed, or otherwise physically hurt by someone)?"<br>Sexual:<br>"Have you ever been pressured into doing more sexually than you wanted to do or were too young to understand? (By sexually we mean being pressured against your will into forced contact with the sexual part of your body or his/her body)." | Lifetime | No; N/A |
| Swett et al., 1991 [207] | Physical, sexual | Any; including husband, parent, another family member or friend | Physical abuse was defined as having been "physically hurt or attacked by someone-such as husband, parent, another family member or friend (for example, have you ever been kicked, bitten, pushed, or otherwise physically hurt by someone)?"<br>The question about sexual abuse was, "Have you ever been pressured into doing more sexually than you wanted to do or were too young to understand? (By sexually we mean being pressured against your will into forced contact with the sexual part of your body or his/her body.)" | Lifetime | No; N/A |
| Briere et al., 1997 [148] | Physical, sexual | Any; partner or other | Scale and specific items not included; "Patients were asked whether they had ever been raped or physically assaulted, in or outside of a relationship as adults." | Lifetime | No; N/A |
| Bengtsson-Tops et al., 2005 [208] | Physical, sexual, psychological, economic | Any; any | Have you in adulthood (after the age of 16 years)/during the last year been exposed to: serious violence such as being threatened with weapons or violence resulting in wounds, bone or tooth injury, or choking; physical violence such as punches, kicks, or thumps against objects, walls or floor; sexual violence; threats of being injured; threats of being killed; verbal harassments/degradation; isolation at home; economical exploitation; or other forms of abuse? | Lifetime; past 12 months | No; N/A |
| Yellowlees et al., 1994 [149] | Sexual, physical | Any; any | Scale not specified; items not included.<br>Definition of sexual violence: exual approach to a patient by a non-family member where the approach was both unwanted and remembered by the patient as being both physically and psychologically significant, and was resisted.<br>Definition of domestic violence: Domestic violence was defined as occurring only in relationships that were perceived by patients to be both significant and longstanding, and where there had been actual physical harm to the patient that was either of a relatively non-serious but regular nature, or an occasional very severe episode of violence, or a combination | Lifetime | No; N/A |
| Coker et al., 2005 [110] | Physical, sexual, psychological | Relationship; current or previous partner | Abuse Assessment Scale, Women's Experience with Battering Scale (WEB)<br>Physical or sexual IPV included women who had ever experienced physical or sexual IPV in any past relationship (based on the modified Abuse Assessment Scale) or in her current or most recent relationship (based on the modified Index of Spouse Abuse–Physical).<br>Psychological IPV: included women who reported emotional abuse in any past relationship or scored as battered on the Women's Experience with Battering Scale (WEB) | Current = current or most recent relationship<br>Past = previous relationships | No; N/A |
| Brownlie et al., 2007 [45] | Sexual | Any; any | Scale not specific;<br>Two items: "Were you ever raped, that is someone had sexual intercourse with you when you did not want to, by threatening you, or using some degree of force?" And "Were you ever sexually molested, that is, someone touched or felt your genitals when you did not want them to?" | Lifetime | No; N/A |
| Diaz-Olavarrieta et al., 1999 [209] | Physical, emotional, sexual | Any; Husband, ex-husband, boyfriend, father, stranger. | Modified version of Abuse Assessment Screen:<br>Have you ever been physically or emotionally abused by your partner or someone important to you? Yes/ No<br>Type of abuse: Hitting, kicking, slapping, strangling<br>When you have been pregnant, have you been hit, slapped, kicked, or otherwise hurt by someone?<br>Have you ever been forced to have sexual intercourse?<br>Are you afraid of your partner or anyone else listed above? | Lifetime; past 12 months; during pregnancy | No; N/A |
| Findley et al., 2016 [143] | Physical, sexual, psychological, financial, and disability-related abuse | Any; an intimate partner, care provider, health professional, family member, or other. | Abuse Assessment Screen-Disability scale;<br>Within the last year, have you been hit, slapped, kicked, pushed, shoved, or otherwise physically hurt by someone (i.e., physical abuse)?<br>Within the last year, has anyone forced you to have sexual activities?<br>Within the last year, has anyone prevented you from using a wheelchair, cane, respirator, or other assistive devices?<br>Within the last year, has anyone you depend on refused to help you with an important personal need, such as taking your medicine, getting to the bathroom, getting out of bed, bathing, getting dressed, or getting food or drink (i.e., disability-related abuse)? And<br>Item added for this study: Within the last year, did someone take your Supplemental Security Income (SSI) or Social Security Disability (SSD) check, a paycheck, or financial aid check without your permission; refuse to allow you to access your bank account; or restrict your use of money, a debit, or credit card (i.e., financial abuse) | Past 12 months | Yes; 3 items:<br>"Within the last year, has anyone prevented you from using a wheelchair, cane, respirator, or other assistive devices?<br>And: "Within the last year, has anyone you depend on refused to help you with an important personal need, such as taking your medicine, getting to the bathroom, getting out of bed, bathing, getting dressed, or getting food or drink (i.e., disability-related abuse)?<br>And: "Within the last year, did someone take your Supplemental Security Income (SSI) or Social Security Disability (SSD) check, a paycheck, or financial aid check without your permission; refuse to allow you to access your bank account; or restrict your use of money, a debit, or credit card (i.e., financial abuse) |
| Giraldo-Rodriguez et al., 2015 [210] | Physical, psychological, financial, sexual | Any; not specified | Scale not specified;<br>Items: "Have you been treated in an aggressive or violent manner?; Has anyone said things to you and made you feel bad?; Has anyone disparaged or disrespected you?; Have you been humiliated in front of others?; Have you been insulted?; Have you been threatened?; Has anyone destroyed your things?; Has anyone made you feel afraid?; Have you been forbidden to go out or be visited?; Have your decisions not been respected about important events?; Has anyone invaded your privacy?; Have you been controlled or not been given money?; Has anyone managed or does anyone manage your money without your consent?; Have you been forced to sign or put your fingerprint on any document?; Have you been forced to sign or put your fingerprint on any document that you do not understand? Has anyone decided the manner in which your money is spent?; Have you been forced to sell any belongings without your consent?; Have you been forced to work even if you did not want to?; Have you been forced to do things against your will?; Has anyone stolen your personal documents (birth certificate, personal identification)?<br>Has anyone touched you sexually or has anyone forced you to do anything sexual without your consent? | Physical, psychological, financial– past 12 months<br>Sexual–lifetime | No; N/A |
| Longobardi et al., 2018 [144] | Physical, sexual, disability-related | Any; intimate partner, care provider, health professional, family member, other | Abuse Assessment Screen-Disability scale;<br>Within the last year, have you been hit, slapped, kicked, pushed, shoved, or otherwise physically hurt by someone (i.e., physical abuse)?<br>Within the last year, has anyone forced you to have sexual activities?<br>Within the last year, has anyone prevented you from using a wheelchair, cane, respirator, or other assistive devices?<br>Within the last year, has anyone you depend on refused to help you with an important personal need, such as taking your medicine, getting to the bathroom, getting out of bed, bathing, getting dressed, or getting food or drink (i.e., disability-related abuse)?<br>In addition, pre-disability violence question:<br>Have you experienced any violence prior to the emergence of your disability? | Past 12 months | Yes–two items:<br>Within the last year, has anyone prevented you from using a wheelchair, cane, respirator, or other assistive devices?<br>Within the last year, has anyone you depend on refused to help you with an important personal need, such as taking your medicine, getting to the bathroom, getting out of bed, bathing, getting dressed, or getting food or drink (i.e., disability-related abuse)? |

*(Continued)*

**Table 3.** (*Continued*)

| Article | Type(s) of violence | Context/ Perpetrator(s) | Scale, measure, specific items | Time frame(s) | Violence measure(s) specific to women with disabilities (Y/N) If yes, specific items/ measures |
|---|---|---|---|---|---|
| Ministry of Labor, Invalids and Social Affairs et al., 2020 [120] | Physical, Sexual, Emotional, Controlling behaviour Economic abuse | Relationship and non-relationship; intimate partner. Non-partner: parent, parent-in-law, sibling, other family member, someone at work, friend/ acquaintance, recent acquaintance, complete stranger, teacher, doctor/ health staff, religious leader, police/ soldier, other | Adapted WHO MCS;<br>Physical: Slapped you or thrown something at you that could hurt you?<br>Pushed you or shoved you or pulled your hair?<br>Hit you with his fist or with something else that could hurt you?<br>Kicked you, dragged you or beaten you up?<br>Choked or burnt you on purpose?<br>Threatened with or actually used a gun, knife or other weapon against you?<br>Sexual: Did your current husband/partner or any other husband/partner ever force you to have sexual intercourse when you did not want to, for example by threatening you or holding you down?<br>Did you ever have sexual intercourse you did not want to because you were afraid of what your partner or any other husband or partner might do if you refused?<br>Did your husband/partner or any other husband or partner ever force you to do anything else sexual that you did not want or that you found degrading or humiliating?<br>Emotional:<br>Insulted you or made you feel bad about yourself?<br>Belittled or humiliated you in front of other people?<br>Done things to scare or intimidate you on purpose (e.g. by the way he looked at you, by yelling and smashing things)?<br>Verbally threatened to hurt you or someone you care about?<br>Controlling behaviours by an intimate partner<br>a) Tried to keep her from seeing friends<br>b) Tried to restrict contact with her family of birth<br>c) Insisted on knowing where she was at all times<br>d) Got angry if she spoke with another man<br>e) Was often suspicious that she was unfaithful<br>e) Expected her to ask permission before seeking health care for herself<br>Economic violence:<br>a) Prohibited her from getting a job, going to work, trading, earning money or participating in income generation projects<br>b) Took her earnings from her against her will<br>c) Refused to give her money needed for household expenses even when he has money for other things (such as alcohol and cigarettes)<br>d) Expected her to be financially responsible for his family and himself<br>e) Expected her to ask his permission before buying anything for herself<br>Physical violence in pregnancy<br>Was pushed, slapped, hit, kicked or beaten while pregnant<br>Was punched or kicked in the abdomen while pregnant<br>Non-partner physical violence:<br>a) Slapped, hit, beaten, kicked or done anything else to hurt her<br>b) Thrown something at her, pushed her or pulled her hair<br>c) Choked or burned her on purpose<br>d) Threatened with or actually used a gun, knife or other weapon against her<br>Non-partner sexual violence:<br>a) Forced her to have sexual intercourse when she did not want to<br>b) Forced to have sexual intercourse when she was too drunk or drugged to refuse<br>c) Forced or persuaded to have sex against her will with more than one man at the same time<br>d) Attempted to force her into sexual intercourse when she did not want to, for example by holding her down or putting her in a situation where she could not say no<br>e) Touched her sexually against her will<br>f) Made her touch their private parts against her will | Physical, sexual, emotional, controlling and economic abuse by partner–all lifetime and past 12 months; Non-partner physical and sexual–after age 15 and past 12 months; violence during pregnancy | No; N/A |
| National Statistics Office et al., 2018 [121] | Physical, sexual, emotional, economic, controlling behaviours. | Relationship and non-relationship; intimate partner. Non-partner: parent, parent-in-law, sibling, other family member, someone at work, friend/ acquaintance, recent acquaintance, complete stranger, teacher, doctor/ health staff, religious leader, police/ soldier, brother/ sister-in-law, step-father, step-brother/ sister, other | Adapted WHO MCS;<br>Physical IPV:<br>Was slapped or had something thrown at her that could hurt her<br>Was pushed or shoved or had her hair pulled<br>Was hit with fist or something else that could hurt<br>Was kicked, dragged, or beaten up<br>Was choked or burnt on purpose<br>Perpetrator threatened to use, or actually used, a gun, knife, or other weapon against her<br>Was chased by a car or motorcycle<br>Was chased by a horse and/or lashed with a whip<br>Was lashed with a belt<br>Controlling behaviours by an intimate partner<br>He tried to keep her from seeing friends<br>He tried to restrict contact with her family of birth<br>He insisted on knowing where she was at all times<br>He got angry if she spoke with another man<br>He was often suspicious that she was unfaithful<br>He expected her to ask permission before seeking health care for herself<br>Sexual IPV:<br>Was physically forced to have sexual intercourse when she did not want to<br>Had sexual intercourse when she did not want to because she was afraid of what partner might do<br>Was forced to do something sexual that she found degrading or humiliating<br>Physical violence in pregnancy<br>Was pushed, slapped, hit, kicked or beaten while pregnant<br>Was punched or kicked in the abdomen while pregnant<br>Emotional IPV<br>Was insulted or made to feel bad about herself<br>Was belittled or humiliated in front of other people<br>Perpetrator had done things to scare or intimidate her on purpose e.g. by the way he looked at her; by yelling or smashing things<br>Perpetrator had threatened to hurt her or someone she cared about<br>Non-partner physical violence:<br>Slapped, hit, beaten, kicked or done anything else to hurt you?<br>Thrown something at you? Pushed you or pulled your hair?<br>Choked or burnt you on purpose?<br>Threatened with or actually used a gun, knife or other weapon against you?<br>Non-partner sexual violence:<br>**a) Since the age of 15 until now**, has anyone (other than your male partner) ever **forced you into sexual intercourse** when you did not want to, for example by threatening you, holding you down, or putting you in a situation where you could not say no. Remember to include people you have known as well as strangers. Please at this point exclude **attempts** to force you.<br>**b)** Has anyone (other than your male partner) ever forced you to have sex when you were too drunk or drugged to refuse?<br>**c)** Have you been forced or persuaded to have sex against your will with more than one man at the same time?<br>*Attempted sexual assault;*<br>Has anyone attempted but not succeeded to force you into sexual intercourse when you did not want to, for example by holding you down or putting you in a situation where you could not say no?<br>Touched you sexually against your will?<br>Made you touch their private parts against your will? | Physical, sexual, emotional, controlling and economic abuse by partner–all lifetime and past 12 months; Non-partner physical and sexual–after age 15 and past 12 months; violence during pregnancy | No; N/A |

(*Continued*)

**Table 3.** (Continued)

| Article | Type(s) of violence | Context/ Perpetrator(s) | Scale, measure, specific items | Time frame(s) | Violence measure(s) specific to women with disabilities (Y/N) If yes, specific items/ measures |
|---|---|---|---|---|---|
| National Commission for Women and Children, 2017 [119] | Physical, sexual, emotional, economic | Relationship and stranger; current or former male intimate partner (e.g. husband, co-habiting partner, fiance, boyfriend). Mother, father, teacher, friend, stranger | Adapted WHO MCS<br>Physical IPV:<br>Was slapped or had something thrown at her that could hurt her<br>Was pushed or shoved or had her hair pulled<br>Was hit with a fist or something else that could hurt<br>Was kicked, dragged or beaten up<br>Was choked or burnt on purpose<br>Threatened to use or actually used a weapon against her<br>Chased out of the house/denied shelter using physical force<br>Sexual IPV:<br>Was forced to have sexual intercourse when she did not want to, for example by being threatened or held down<br>Had sexual intercourse when she did not want to because she was afraid of what the partner might do if she refused<br>Was forced to do anything sexual that she did not want or that she found degrading or humiliating<br>Emotional IPV:<br>Was insulted or made to feel bad about herself<br>Was belittled or humiliated in front of other people<br>Had done things to scare or intimidate her on purpose (e.g., by yelling or smashing things)<br>Threatened verbally to hurt her or someone she cared about<br>Physical violence in pregnancy<br>Was punched or kicked in the abdomen while pregnant<br>Controlling behaviour by an intimate partner<br>Tried to keep her from seeing friends<br>Tried to restrict contact with her family of birth<br>Insisted on knowing where she is at all times<br>Got angry if she spoke with another man<br>Was often suspicious that she is unfaithful<br>Expected her to ask permission before seeking healthcare for herself<br>Economic IPV<br>Prohibited from getting a job, going to work, trading, earning money or participating in income generation projects<br>Had her earnings taken from her against her will<br>Refused to give her money she needed for household expenses even when he had money for other things (such as alcohol and cigarettes)<br>Non-partner physical violence:<br>Slapped, hit, kicked or anything else to hurt her<br>Had something thrown at her, was pushed or had her hair pulled<br>Choked or burnt on purpose<br>Threatened to use or actually used a gun, knife or other weapons against her<br>Non-partner sexual violence:<br>Was forced by a non-partner into sexual intercourse when she did not want to, for example by threatening her, holding her down, or putting her in a situation where she could not say no<br>Forced to have sex when she was too drunk or drugged to refuse<br>Forced or persuaded to have sex against her will with more than one man at the same time<br>*Sexual assault:*<br>Attempted but did not succeed in forcing her into sexual intercourse when she did not want to, for example by holding her down or putting her in a situation where you could not say no<br>Touched her sexually against her will. This includes, for example, touching of breasts or private parts<br>Made her touch their private parts against her will | Physical, sexual, emotional, controlling and economic abuse by partner–all lifetime and past 12 months; Non-partner physical and sexual–after age 15 and past 12 months; violence during pregnancy | No; N/A |
| CREA 2012 [211] | Physical, sexual, emotional | Relationship and stranger; partner, non-partner (not further specified) | Name of scale not specified;<br>Emotional abuse: Insulted you or made you feel bad about yourself<br>Belittled/humiliated you in front of other people<br>Did things to scare/intimidate you on purpose<br>Threatened to hurt someone you care about<br>Physical violence:<br>Slapped or threw something at you<br>Pushed or shoved or pulled hair<br>Hit with fist or with something else<br>Kicked/dragged/beat up<br>Choked or burnt on purpose<br>Threatened to use a gun, knife, or other weapon<br>Sexual violence:<br>Physically forced to have sex<br>Experienced something sexual when you were afraid of what your intimate partner might do<br>Experienced something sexual that you found degrading or humiliating | Lifetime; past 12 months | No; N/A |
| SINTEF 2016a [131] | Physical | Relationship; Family member | Name of scale not specified; Definition is: "beaten or scolded by family member" | Not specified | No; N/A |
| Bureau of Justice Statistics 2017 [150] | Sexual, physical | Any; Intimate partner<br>Other relatives<br>Well known/casual acquaintances<br>Strangers | Name of scale not specified; items not included | Past 6 months | No; N/A |
| SINTEF 2016b [132] | Physical | Relationship; Family member | Name of scale not specified; Definition is: "beaten or scolded by family member" | Not specified | No; N/A |
| Uganda Bureau of Statistics 2018 [212] | Physical, sexual | Any; any | Scale not specified;<br>Items:<br>From the age of 15 years they had been hit, slapped or kicked by someone, or something else had been done to hurt them physically.<br>At any time in your life, as a child or as an adult, has anyone ever forced you in any way to have sexual intercourse or perform any other sexual acts when you did not want to? | Lifetime; past 12 months | No; N/A |
| Schröttle et al 2013 [156] | Physical, sexual violence, sexual harassment, psychological | Any; any | Modified CTS;<br>Physical violence: operationalized by a list of 21 items ranging from less severe forms of violence (like being pushed away angrily or a light slap in the face) to severe and very severe forms (punching, beating up, strangling, severe threat or use of weapons).<br>Sexual violence: six items addressing forced acts like: "somebody has forced me to have sexual intercourse", "somebody has forced me to engage in sexual acts or practices that I did not want".<br>Sexual harassment: 14 items addressing acts ranging from verbal harassment and gazing, up to unwanted touching, kissing and stalking. Psychological violence: 11 items with various acts from verbal aggression and severe insults over severe threat and continued hassling up to psycho terror | Lifetime; past 12 months | No; N/A |

*(Continued)*

**Table 3.** (Continued)

| Article | Type(s) of violence | Context/ Perpetrator(s) | Scale, measure, specific items | Time frame(s) | Violence measure(s) specific to women with disabilities (Y/N) If yes, specific items/ measures |
|---|---|---|---|---|---|
| Instituto Nacional de Estadistica e Informatica, 2014 [213] | Physical, sexual, psychological/ verbal | Relationship or stranger; current or for former husband/partner, current or former boyfriend, Father/ stepfather Mother/stepmother Sister/brother, other | DHS DV Module: <br> Physical spousal violence: push you, shake you, or throw something at you; slap you; twist your arm; punch you with his/her fist or with something that could hurt you; kick you, drag you; try to choke you or burn you; attack you with a knife, gun or other weapon; threaten you with a knife, gun, or any other weapon <br> Sexual spousal violence: physically force you to have sexual intercourse with him even when you did not want to; force you to perform any other sexual acts you did not want to <br> Emotional spousal violence: get jealous or upset if you converse with another man; frequently accuses you of being unfaithway; prevents you from visiting or being visited by your friends; tried to limit visits and contacts with your family; insists on knowing all the places you go; mistrusts you with money; has said or done things to humiliate you in front of others; has threatened to harm you or someone close to you; has threatened to leave the house, take away the children or financial aid? <br> Non-partner physical violence: <br> Hit, slapped, kick or abused you physically | Lifetime; past 12 months | No; N/A |
| Minsalud, Profamilia, 2015 [214] | Physical, sexual, psychological, economic | Relationship; former or current partner | DHS DV module: <br> Psychological IPV: <br> Has your partner or ex-partner ever: <br> 1. Become jealous if you talk to another man? <br> 2. Accused you of being unfaithful? <br> 3. Prevented you from meeting friends? <br> 4. Tried to limit the relationship with your family? <br> 5. Insisted on knowing where you are all the time? <br> 6. Ignored you/ or has not addressed you? <br> 7. Not consider you for social or family gatherings? <br> 8. Not consulted you on important decisions for your family? <br> 9. Threatened you with a knife, gunfire or other weapon? <br> 10. Referred to you in terms such as: "You are useless", "You never do things right", "you are a brute", or "My mom used to do things better"? <br> 11. Threatened to abandon you or leaving with another women? <br> 12. Threatened to take your children from you? <br> Physical IPV: <br> Has your partner or ex-partner ever: <br> 1. Pushed you or shaken you? <br> 2. Hit you with his hand? <br> 3. Hit you with an object? <br> 4. Kicked you or dragged you? <br> 5. Attacked you with a knife, firearm or other weapon? <br> 6. Tried to strangle or burn you? <br> Economic IPV: <br> Has your partner or ex-partner ever <br> 1. Watched the way you spend money? <br> 2. Threatened you with taking away your financial support? <br> 3. Forbidden to work or study? <br> 4. Spent the money that was needed for the house expenses? <br> 5. Taken over or taken away money or goods (land, property, etc.)? <br> Sexual IPV: <br> Has your partner or ex-partner ever: <br> Physically forced you to have intercourse or sexual acts that you did not want? | Lifetime; past 12 months | No; N/A |
| Centro de Estudios Sociales y Demográficos (CESDEM) et al., 2014 [215] | Physical, sexual, psychological | Relationship or stranger; current or former husband/partner, current or former boyfriend, father/stepfather, mother/stepmother, sister/brother, daughter/son, other relative, mother-in-law, other in-law, teacher, employer/someone at work, police/army, other | DHS DV module: <br> Emotional IPV: <br> a. Did he say or do anything to humiliate you in the presence of other people? <br> b. Did he threaten to hurt you or someone close to you? <br> c. Did he insult you and make you feel bad? <br> Physical IPV: <br> d. Did he push you, shake you, or throw anything at you? <br> e. Did he slap you? <br> f. Did he twist your arm or pull your hair? <br> g. Did he punch you or hit you with something that could hurt you? <br> h. Did he kick you or drag you across the ground? <br> i. Did he tried to strangle you or burn you with something? <br> j. Did he threaten or attack you with a knife, firearm, or other weapon? <br> Sexual IPV: <br> k. Did he physically force you to have sex with him even though you did not wanted? <br> l. Did he force you to perform sexual acts that you did not want? <br> Non-partner sexual violence <br> Has anyone ever forced you to have sex or perform sexual acts that you did not want in your life, either as a child or as an adult woman? <br> In the past 12 months, has someone other than her husband physically forced you to have sex when you don't want to? | Lifetime; past 12 months | No; N/A |
| Secretaría de Salud [Honduras] et al., 2013 [216] | Physical, sexual, psychological | Relationship or stranger; current or former husband/partner, current or former boyfriend, father/stepfather, mother/stepmother, sister/brother, daughter/son, other relative, mother-in-law, other in-law, teacher, employer/someone at work, police/army, other | Psychological IPV: <br> Please tell me if these apply to your relationship with your (last) husband (partner): <br> a) Does your husband (partner) get jealous or upset if you talk to another man? <br> b) Does he frequently accuse you of being unfaithful? <br> c) Does he prevent you from visiting or being visited by friends? <br> d) Does he try to limit visits / contacts with your family? <br> e) Does he always insist (insisted) on knowing all the places where you go / went? <br> Your (last) spouse (partner) ever: <br> a) Has he said or done things to her to humiliate her in front of others? <br> b) Has he threatened to harm you or someone close to you? <br> c) Has he insulted her or made her feel bad about herself? <br> Physical IPV: <br> Your (last) spouse (partner) ever: <br> a) Did he push you, shake you, or throw anything at you? <br> b) Did he slap you? <br> c) Did he twist your arm or pull your hair? <br> d) Did he hit you with his fist or something that could hurt you? <br> e) Has he kicked or dragged you? <br> f) Did he try to strangle or burn you? <br> g) Did he threaten or attack you with a pistol knife or other type of weapon? <br> Sexual IPV: <br> h) Has he used physical force to force you to have sexual intercourse? <br> i) Has he forced or threatened you in any other way to perform sexual acts that you did not want? <br> Non-partner violence: <br> Since you turned 15, has anyone hit, slapped, kicked, or physically abused you? <br> Outside of your husband (partner), did someone force you to have sexual acts that you did not want? | Lifetime; past 12 months | No; N/A |

Powers et al., 2002:
If personal assistant:
- Makes decisions or choices without asking;
- Forges checks/ uses credit cards;
- Denies choices;
- Threatens to leave;
- Blocks path/ puts things out of reach;
- Denies time alone;
- Handles roughly;
- Neglects physical needs;
- Pressures for money;
- Threatens physical harm;
- Pushes beyond limits;
- Violates body privacy;
- Physically abusive;
- Withholds, immobilizes or breaks equipment;
- Withholds, steals or overdoses medication;
- Touches sexually in unwanted ways;
- Withholds communication device;
- Threatens to hurt or hurts pets;
- Forces unwanted sexual activity;
- Abuses woman's children

Platt et al., 2017:
- As an adult, has anyone you know ignored or refused to help you with an important personal need such as using the bathroom, banking, dressing, eating, communicating, or going out in the community?
- As an adult, has anyone you know purposely broken or kept you from using things such as a wheelchair, breathing machine, communication device, or service animal?
- As an adult, has anyone you know kept you from taking your medicine or given you more medicine than they were supposed to?
- As an adult, has anyone you know stolen or misused your money, bank account, or debit/credit cards?
- As an adult, has anyone you know physically handled you in a rough way?
- As an adult, has anyone you know held or tied you down or made you stay someplace when you did not want to?

Curry et al., 2009:
- In the last year, has anyone you know refused or forgotten to help you with an important personal need such as: Toileting or going to the bathroom; Bathing; Helping you move; Getting dressed; Getting food or water
- In the last year, has anyone you know broken or kept you from using important things such as a: Phone; Wheelchair; Cane; Walker; Respirator; Communication device; Service animal; Other assistive devices
- In the last year, has anyone you know: Kept you from taking your medication? Given you too much or too little medication?
- In the last year, has anyone you know: Stolen your money, important items, or equipment? Signed your checks to take money from you? Used your credit or debit card without your OK?

Johnston-McCabe et al., 2011:
Items added to CTS:
- Whether or not the abusive partner is Deaf, Hard of Hearing, or hearing;
- The age of onset of partners' hearing loss; and
- The degree of partners' hearing loss

McFarlane et al., 2001; Nosek et al., 2006, Nosek et al., 2001, Longobardi et al., 2018
- Within the last year, has anyone prevented you from using a wheelchair, cane, respirator, or other assistive devices?
- Within the last year, has anyone you depend on refused to help you with an important personal need, such as taking your medicine, getting to the bathroom, getting out of bed, bathing, getting dressed, or getting food or drink?

Yoshida et al., 2011; Young et al., 1997
- Being restrained
- Being denied food or water
- Included "looking" as a form of sexual abuse (personal assistants/ carers may look or inappropriately stare at women with disabilities)

Del rio Ferres et al., 2013:
- Within the last year, has anybody you depend on refused to help you with an important personal need (related to your basic daily activities), such as taking your medicine, getting to the bathroom, getting out of bed, bathing, getting dressed, or getting food and drink?
- Within the last year, has anyone prevented you from using any of the technical aids you need in your daily life, such as a wheelchair, cane, respirator, or other assistive devices?

Milburger et al., 2003:
- Since you were 18 years old, has anyone prevented you from using a wheelchair, cane, respirator, or other assistive devices?
- Since you were 18 years old, has anyone you depend on refused to help you with an important personal need such as taking your medicine, getting to the bathroom, getting out of bed, bathing, getting dressed or getting food or drink or threatened not to help you with these personal needs?

Findley et al., 2016:
- Within the last year, has anyone prevented you from using a wheelchair, cane, respirator, or other assistive devices?
- Within the last year, has anyone you depend on refused to help you with an important personal need, such as taking your medicine, getting to the bathroom, getting out of bed, bathing, getting dressed, or getting food or drink?
- Within the last year, did someone take your Supplemental Security Income (SSI) or Social Security Disability (SSD) check, a paycheck, or financial aid check without your permission; refuse to allow you to access your bank account; or restrict your use of money, a debit, or credit card?

**Fig 2. Disability-specific items.**

item as a screening question (for example, Rasoulian et al. asked "Have you ever been intentionally hurt physically by your husband in your lifetime?") and then followed up with more detailed questions about the violence, for example, timing, location, frequency and perpetrator [86, 126, 135]. n = 13 included only a single item for any form of violence measures [50, 51, 58, 59, 76, 83, 87, 95, 100, 116, 127, 131, 132, 138]. Given missing detail in some manuscripts and reports, it is possible that more studies utilized acts-based questions and gold standard measures. n = 27 studies did not include any information regarding the time frame of violence, i.e. whether respondents were asked about violence experienced over the past year or since a certain age or both.

## Measurement of disability

Table 4 displays measurement of disability within the 174 included manuscripts and reports, indicating types of disability assessed, type of assessment, including name of scale and items if included, if severity of disability was measured and if so, how, and if a time frame of disability was indicated in the study. We categorized types of manuscripts and reports into physical, intellectual, mental and sensory. Of the included manuscripts and reports, n = 87 examined one of these types of disability, i.e. only sensory or only physical, and n = 84 included more than one of these types of disability. n = 104 included physical disability, n = 94 included mental disability, n = 61 included intellectual disability and n = 62 included sensory disability. n = 2 did not specify the type of disability assessed in the study.

In terms of measurement instruments, we found that amongst the included manuscripts and reports, n = 75 were measures of functioning (n = 20 of these were Washington Group questions), n = 15 utilized a single question approach and n = 67 defined participants in the research as having a disability based on a diagnosis or self-report of a health condition. n = 20 did not indicate or include the type of measures utilized to define participants as having a disability.

## Discussion

This scoping review represents a comprehensive overview of measurement of disability and violence against women, including a wide range of studies, drawing on different types of literature (national surveys, grey literature, peer-reviewed literature) and employing inclusion and exclusion criteria that sought to identify a broad body of literature to inform discussions and considerations concerning measurement of violence against women and disability. We included n = 174 manuscripts or reports in this scoping review, and presented a typology of the types of studies and/ or research questions, and analysis of measurement of both disability and violence against women. One of the objectives of this scoping review was to map the field of measurement of violence against women and disability in different bodies of literature. This broad mapping enabled us to identify research gaps and this scoping review will be utilized both as the basis for sub-analyses of the included studies, and to develop research objectives for subsequent systematic reviews.

We identified two over-arching typologies of studies: studies that included only persons with disabilities (n = 42 included only women; n = 22 included women and men) and studies that included persons with disabilities and persons without disabilities (n = 75 included only women; n = 36 included women and men). These different approaches enabled different types of research questions and comparisons. For example, studies that only included women with disabilities focused on questions of prevalence of violence, types of violence experienced, and other risk factors for violence, while studies including women with and without disabilities often assessed differences in prevalence between women with and without disabilities. While

**Table 4. Measurement of disability.**

| Article | Types of disability | Scale/ measure of disability, specific items | Measure of severity of disability? Yes/ No; If Yes, how? | Time frame of disability specified? Yes/ No; If yes, what? |
|---|---|---|---|---|
| Pandey et al., 2012 [96] | Night blindness or any blindness during pregnancy | No scale or specific measure; Single item– respondents asked if experienced night or daytime blindness during pregnancy | No | Yes–pregnancy |
| Valera and Kucyi, 2017 [125] | Traumatic brain injury [TBI] | Diagnostic criteria for Mild TBI by the American Congress of Rehabilitation Medicine Special Interest Group on Mild TBI. Respondents asked about periods of dizziness, seeing stars and spots, being stunned or disoriented, blacking out/loss of consciousness, or sustaining post-traumatic amnesia (memory loss surrounding an incident). If subjects reported AIC [alterations in consciousness], they were asked about the conditions of the incident, duration of AIC, when the last and first times AICs occurred and on how many occasions. Rivermead Post Concussion Symptom Questionnaire– 16 item questionnaire to assess for emotional, cognitive and behavioral symptoms following a TBI. | No | No |
| Valera et al., 2019 [162] | Traumatic brain injury [TBI] | Diagnostic criteria for TBI by the American Congress of Rehabilitation Medicine Special Interest Group on Mild TBI. Using a semi-structured interview, women were asked questions about potential traumas to the head that might have resulted in an alteration in consciousness (AIC); also asked about AIC as a result of strangulation. | Yes; questions asked about duration of AIC. TBI was considered mild if a loss of consciousness was less than 30 min and/or the post-traumatic amnesia was less than 24 h. Brain injury severity score was created based on number, recency (number of weeks since most recent brain injury), and severity of AICs. | No |
| Slayter, 2009 [163] | Any | No scale or measure; Variable operationalized as a report of "yes" in response to: "In the past twelve months, have you received SSI, that is, Supplemental Security Income or financial assistance for disabled people?" | No | No |
| Powers, 2002 [60] | Physical, cognitive, sensory, mental | No scale, measure or items specified. Participants reported if had one/ some of the following disabilities: mobility, learning, mental health, visual, speech, hearing | Yes–respondents asked to self-report if mild, moderate or severe | Yes–respondents asked if disability was since birth, since childhood or since age 21 |
| Alangea et al., 2018 [164] | Not specified | No scale, measure or items specified | No | No |
| Astbury and Walji, 2014 [46] | Significant difficulty with a function such as seeing, hearing, mobility, cognition, self-care, and communication | Washington Group Short Set questions 1. Do you have difficulty seeing, even if wearing glasses? 2. Do you have difficult hearing, even If using a hearing aid? 3. Do you have difficulty walking or climbing stairs? 4. Do you have difficulty remembering or concentrating? 5. Do you have difficulty with self-care, such as washing all over or dressing? 6. Using your usual language, do you have difficulty communicating, for example understanding or being understood? For all questions, response categories are: 1. No difficulty 2. Some difficulty 3. A lot of difficulty 4. Cannot do at allRespondent defined as with disability if reported significant difficulty with at least one activity | No | No |

*(Continued)*

**Table 4.** (Continued)

| Article | Types of disability | Scale/ measure of disability, specific items | Measure of severity of disability? Yes/ No; If Yes, how? | Time frame of disability specified? Yes/ No; If yes, what? |
|---|---|---|---|---|
| Cannell et al., 2015 [38] | Physical disability | Physical function scale (PFS) on the Rand 36-Item Health Survey (SF-36); If health limits respondents' ability to engage in 10 different activities ranging from vigorous physical activity to bathing and dressing, and if so, by how much. | Yes–cut-off of significant impairment vs. no significant impairment | No |
| Coston, 2019 [111] | Limitations due to physical, mental or emotional problems; use of special equipment | Scale not specified; Two items: "Are any of your activities limited in any way because of physical, mental, or emotional problems?" and "Do you now have any health problems that require you to use special equipment, such as a cane, a wheelchair, a special bed, or a special telephone?" | No | Yes; respondents asked "How long have you been limited in this way?" or "How long have you been using this equipment?" Response options were "less than 1 year," "more than 1 year but less than 3 years," or "3 or more years." |
| Dembo et al., 2018 [39] | Deaf, blind, physical disability, self-care or independent living limitation, cognitive disability | American Community Survey Disability screening questions. Respondents were identified as having a disability if they reported that they: were deaf or had serious difficulty hearing; were blind or had serious difficulty seeing even when wearing glasses; had a physical disability, defined as a condition that substantially limited one or more basic activities such as walking, climbing stairs, reaching, lifting, or carrying; had a self-care or independent living limitation, defined as a condition that caused difficulty with dressing, bathing, or getting around inside the home, or a condition that caused difficulty with going outside the home alone to shop or visit a doctor's office; or had a cognitive disability, defined as a condition that caused difficulty with learning, remembering, or concentrating. | No | No |
| Emerson et al., 2016 [40] | General physical or mental impairment | No scale. Single item–"Do you have any longstanding physical or mental impairment, illness or disability? By 'long-standing' I mean anything that has troubled you over a period of at least 12 months or that is likely to trouble you over a period of at least 12 months." | No | Yes; 12 months |
| Gibbs et al., 2018 [165] | Significant difficulty with a function such as seeing, hearing, mobility, cognition, self-care, and communication | Washington Group Short Set questions. Assessed using mean score | No | No |
| Guedes et al., 2016 [87] | Lower extremity functioning, mobility | External report: Short Physical Performance Battery [SPPB]–includes three timed tests of lower body function: a hierarchical test of standing balance, a 4 m walk and five repeated chair stands. An SPPB score lower than 8 was indicative of poor physical performance. Self-report: Mobility disability defined by two self-report questions: "Do you have difficulties climbing 10 stairs without resting?" and "Do you have difficulties walking 400 m?" People who reported difficulty in walking 400 m and/or climbing 10 stairs were defined as having mobility disability | Yes–SPPB cut-off of 8 to indicate significant vs. non-significant poor physical performance | No |
| Kutin et al., 2017 [133] | Non-specific–disability or long-term health condition | Scale not specified; Items not included | No | No |
| Le et al., 2016 [166] | Pain, disability | Duke Health Profile-Adolescent Version (DHP-A); domains on pain and disability. Items not included | No | No |

(*Continued*)

**Table 4.** (*Continued*)

| Article | Types of disability | Scale/ measure of disability, specific items | Measure of severity of disability? Yes/ No; If Yes, how? | Time frame of disability specified? Yes/ No; If yes, what? |
|---|---|---|---|---|
| Platt et al., 2017 [75] | Developmental disability | Scale not specified; Items include: Harder to speak or be understood; harder to see; harder to hear; harder to learn or remember; harder to understand or process; harder to walk; daily personal needs harder; harder to live independently; harder to make decisions and plans; socialize differently; harder to work independently; harder to manage money; needs assistive devices; needs personal assistance | No | No |
| Puri et al., 2015 [61] | Physical or sensory | Not specified | No | Yes; asked respondents age of onset of disability |
| Slayter et al., 2017 [112] | General—activities of daily living [ADLs] | Scale not specified; A woman was counted as having a disability if she had at least one ADL. Items included: Do you have difficulty in: dressing, bathing, or getting around the house; learning, remembering, or concentrating due to condition; getting along with others due to condition; leaving home alone or to go to see a doctor due to condition; working at job or business due to condition; participating in school, housework, or daily activities; or "do you have a condition that substantially limits your physical activity." | No | No |
| Valentine et al., 2019 [97] | Sight, hearing, communicating, remembering (cognitive), walking, washing or dressing | Washington Group Short Set questions Women who reported "a lot of difficulty" or "cannot function at all" to any of the functional areas were classified as having a disability | No | No |
| Wall et al., 2018 [134] | Cognitive | TBI: Assessed using Ohio State University TBI Identification Method (OSU-TBI-ID). 3 scales to assess cognitive impacts of TBI: The Automated Neuropsychological Assessment Metrics (Version 4) Core Battery; Rey 15 Item Test; and Trail Making Test B:A Ratio–all to screen for gross neuropsychological deficits and assess performance validity. | Severity of TBI assessed by number of TBIs | No |
| Mirindi 2018 [145] | Fistula | Demographic and Health Surveys 11-item fistula module Items not included | No | No |
| Anderson et al., 2012 [64] | Deafness | No scale; No items included–respondents self-identify as deaf or hard of hearing | No | No |
| Anderson et al., 2014 [63] | Deafness | No scale; No items included–respondents self-identify as deaf or hard of hearing | No | No |
| Anderson et al., 2011 [62] | Deafness | No scale; No items included–respondents self-identify as deaf or hard of hearing | No | No |
| Barrett et al., 2009 [114] | Limitations due to physical, mental or emotional problems; use of special equipment | Scale not specified; Two items: "Are you limited in any way in any activities because of physical, mental, or emotional problems?" or "Do you now have any health problem that requires you to use special equipment, such as a cane, a wheelchair, a special bed, or a special telephone?" | No | No |
| Brownridge 2006 [98] | Long-term physical or mental condition or health problem that reduce the amount or kind of activities they could do | Scale not specified; Items not included | No | No |
| Brownridge 2008 [99] | Activities of daily living, physical, mental condition or health problem that limited amount of activities that could do | Scale not specified; Items not included | No | No |

(*Continued*)

**Table 4.** (*Continued*)

| Article | Types of disability | Scale/ measure of disability, specific items | Measure of severity of disability? Yes/ No; If Yes, how? | Time frame of disability specified? Yes/ No; If yes, what? |
|---|---|---|---|---|
| Brunnberg et al., 2012 [88] | Hearing loss, visual disability that cannot be corrected with glasses/ contact lenses, motor disability, difficulties reading or writing, any other | Scale not specified; Items were: Do you have any of the following disabilities/ handicaps? Hearing loss (yes, no); visual disability that cannot be corrected with glasses or contact lenses (yes, no); motor disability (yes, no); difficulties reading and writing (yes, no); some other disability (yes, no). | No | No |
| Curry et al., 2009 [139] | General—physical or mental impairment that substantially limits one or more major life activities | Scale not specified; Items not included | No | No |
| Du Mont et al., 2013 [113] | Cognitive (e.g., learning disability, intellectual delay, autism), psychological (e.g., bipolar disorder, schizophrenia, depression), physical (e.g., back problems, epilepsy, cerebral palsy), and sensory (e.g., blind, hearing impaired) disabilities. | Scale not specified; Items not included | No | No |
| Pollard et al., 2014 [159] | Deafness | Scale not specified; Items not included | No | No |
| Powers et al., 2009 [73] | Any (not specified) | Scale not specified; Items not included | No | No |
| Smith et al., 2008 [115] | Physical, mental or emotional problems | No scale; Single item–"Are you limited in any way in any activities because of physical, mental or emotional problems?" | No | No |
| Alriksson-Schmidt et al., 2010 [100] | Physical disability or long-term health problem | No scale; Single item–"Do you have any physical disabilities or long-term health problems?" | No | No |
| Carbone-López et al., 2006 [89] | Injury disability or chronic disease | No scale; Single item, "Have you ever sustained a serious injury, such as a spinal cord, neck, or head injury, that is disabling or interferes with your normal activities?" Identification of chronic disability was based on a similar question and focused on such conditions as heart and circulatory problems, connective tissue disease, lung disease, diabetes, cancer, muscle and nerve disorders, stomach ailments, severe headaches, osteoporosis, chronic pain or fatigue, or HIV | No | No |
| Eberhard-Gran et al., 2007 [101] | Somatic symptoms | The somatic symptom scale, derived from the Primary Care Evaluation of Mental Disorders (PRIME-MD); Included the following somatic symptoms no/yes): stomach pain, back pain, pain in arms/ legs/ joints, menstrual pain/problems, pain/problems during sexual intercourse, headache, chest pain, dizziness, fainting spells, feeling your heart pound or race, shortness of breath, constipation/loose bowels/diarrhea, feeling tired/ having low energy, and trouble sleeping. | No | No |
| Haydon et al., 2011 [41] | Physical disability or cognitive impairment | Physical disability: Physical disability index (PDI): Includes variables assessing difficulties using limbs due to a permanent physical condition, equipment use, personal care assistance, blindness and deafness. Dichotomous indicator of disability status at Wave I that categorized respondents as having a physical disability if they scored a 2 or higher on the PDI Cognitive performance: Add Health Picture Vocabulary Test (AHPVT); 87 items that ask the respondent to match words (read aloud by the interviewer) with pictorial representations. | Yes–cognitive performance, categorized as below 70, between 70 and 90, between 91 and 110, and above 110. | No |

(*Continued*)

**Table 4.** (Continued)

| Article | Types of disability | Scale/ measure of disability, specific items | Measure of severity of disability? Yes/ No; If Yes, how? | Time frame of disability specified? Yes/ No; If yes, what? |
|---|---|---|---|---|
| Morris et al., 2019 [65] | Intellectual disability | Scale not specified; Items not included | No | No |
| Rasoulian et al., 2014 [126] | Physical illness | Scale not specified; Items not included | No | No |
| Stockl et al., 2011 [102] | Physical disability or severe, chronic illness | Scale not specified; Items not included | No | No |
| Brownridge et al., 2016 [90] | Activity limitations | World Health Organization's [WHO] International Classification of Functioning, Disability and Health [ICF]; Respondents were asked if their daily activities at home, work, school, or any other area were limited by one or more of the following: (a) physical condition; (b) psychological, emotional, or mental condition; (c) learning difficulties; and (d) any other health condition. | No | No |
| Casteel et al., 2008 [42] | Disabling injury, chronic disease or health condition, chronic mental health condition | Scale not specified; A woman was considered to have a disability if she reported one of the following: (1) sustaining a serious injury, such as a spinal cord, neck, or head injury, that is disabling or interferes with normal activities; (2) having a chronic disease or health condition that is disabling or interferes with normal activities; (3) having a chronic mental health disease or condition, such as chronic depression or schizophrenia, that is disabling or interferes with normal activities. | Yes; women were asked about the extent to which these disabilities interfered with normal activities during the past week, with scores ranging from: 0 = 'not at all" to 5 = "extremely". No disability = 0 Moderate disability = 1–3 Severe disability = 4–5 | No |
| Cimino et al., 2019 [167] | TBI | One item from revised Conflict Tactics Scale-2 –becoming unconscious because of a blow to the head; Second item– included in screening survey–"Has your partner ever choked you until you became unconscious?" | No | No |
| Cohen et al., 2005 [168] | Activity limitations–physical or mental | Scale not specified; Single item was: "Does a long term physical or mental condition or heath problem reduce the amount or the kind of activity that you can do at home, at school, at work or in other activities? | No | Yes; "long-term" defined as at least 6 months |
| Cohen et al., 2006 [91] | Activity limitations–physical or mental | Scale not specified; Single item was: "Does a long term physical or mental condition or heath problem reduce the amount or the kind of activity that you can do at home, at school, at work or in other activities? | No | Yes; "long-term" defined as at least 6 months |
| Curry et al., 2011 [140] | Mental or physical impairment that limits life activities | Scale not specified; Items not included | No | No |
| Elliott Smith et al., 2015 [66] | Deafness | No scale; Self-identify as deaf or hard of hearing | No | No |
| Hahn et al., 2014 [67] | Physical health impairments that included physical functioning (e.g., limitations to moderate activities), role physical functioning (e.g., impairment due to health condition), bodily pain, and vitality | Short Form 12, v2.–designed to generate an overall physical component score to assess physical health impairments that included physical functioning (e.g., limitations to moderate activities), role physical functioning (e.g., impairment due to health condition), bodily pain, and vitality; Specific items not included | No | No |
| Hasan et al., 2014 [169] | Physical or sensory disability | Scale not specified; Items not included | Yes–but how it is defined/ measured not specified | No |
| Johnston-McCabe et al., 2011 [74] | Deafness | No scale; Self-identify as deaf or hard of hearing | Yes–mild, moderate, severe, or profound hearing loss. | No |

(*Continued*)

**Table 4.** (*Continued*)

| Article | Types of disability | Scale/ measure of disability, specific items | Measure of severity of disability? Yes/ No; If Yes, how? | Time frame of disability specified? Yes/ No; If yes, what? |
|---|---|---|---|---|
| Krnjacki et al., 2016 [92] | Any disability or long term health condition | The Short Disability Module (SDM); used the SDM to identify whether a person had a disability or long-term health condition. Participants were defined as having a disability if they had a limitation, impairment or restriction in everyday activities that had lasted, or was likely to last, for six months or more. | No | Yes– 6 months |
| Martin et al., 2006 [151] | Physical, mental or emotional problems limiting activity; cognitive limitations; use of special equipment; self-perception as disabled | Scale not specified; Items: One item on activity limitations: asking women whether they felt that "physical, mental, or emotional problems limited their activities in any way"; One items on cognitive limitations, asking the women if they "had trouble learning, remembering, or concentrating"; One item on the use of special equipment, asking women if they "used devices such as a cane, wheelchair, etc"; One item on women's perceptions regarding their own disability status, asking women whether they "considered themselves to have a disability." | No | Yes; women reporting yes to any of the 4 items asked age at which disability began |
| McFarlane et al., 2001 [141] | Physical disability–diagnosed with a physical disability that limited one or more major life activities, including mobility and self-care/home management | Scale not specified; Items not included | No | No |
| Mitra et al., 2012 [103] | Physical, mental or emotional problems | No scale; Single item—"Are you limited in any way in any activities because of physical, mental, or emotional problems?" | No | No |
| Nosek et al., 2006 [142] | Physical disability that impacted one or more major life activity | Disability status assessed according to the paradigm of the WHO ICF: impairment, activity, and participation. Impairment: age at onset and duration of disability; pain scale from the Medical Outcomes Study Short Form-36 [SF-36]; the two-item pain scale asks about severity of pain and the extent to which it interferes with functioning. Activity: three measures of physical functioning: (a) the physical functioning subscale of the SF-36, (b) the mobility subscale of the Craig Handicap Assessment and Reporting Technique, and (c) a two-item measure of the need for personal assistance with activities of daily living (ADLs) and instrumental activities of daily living (IADLs). Participation: measured in terms of social isolation, which was assessed by three items asking about contacts with friends and relatives. Respondents were asked to indicate how many close friends and relatives they have, and how many of these they see at least once a month. | No | Yes–asked age of onset, duration of disability |
| Rees et al., 2011 [170] | Functioning | 12-item version World Health Organization Disability Assessment Schedule and the Short-Form Disability Module; Items not included | No | No |
| Smith et al., 2008 [171] | Physical, mental or emotional problems | No scale; Single item—"Are you limited in any way in any activities because of physical, mental, or emotional problems?" | No | No |

(*Continued*)

**Table 4.** (*Continued*)

| Article | Types of disability | Scale/ measure of disability, specific items | Measure of severity of disability? Yes/ No; If Yes, how? | Time frame of disability specified? Yes/ No; If yes, what? |
|---|---|---|---|---|
| Sumilo et al., 2012 [138] | Longstanding illness, disability or infirmity | No scale specified; Items were: "Do you have a longstanding illness, disability or infirmity. By longstanding I mean anything that has troubled you over a period of time or that is likely to affect you over a period of time?" If "Yes": "What is the matter with you?" and "Does this illness or disability limit your activities in any ways?" | No | No |
| Ward et al., 2010 [172] | Developmental disability | Scale not specified; Items not included | No | No |
| Yoshida et al., 2011 [68] | Any | Scale not specified; Variables included: Born with disability (Y/N); Disability result of abuse (Y/N), Activity prevented by pain (none or a few, some, most); activity limitation (Y/N), needs assistance (Y/N); Health conditions (one, two or more) | No | Yes–asks if born with disability or not |
| Young et al., 1997 [104] | Physical | Scale not specified; Self-reported injuries: spinal cord injury, polio, muscular dystrophy, cerebral palsy, multiple sclerosis, and joint and connective tissue. | Yes–used functional scale from Medical Outcome Study used to categorise into severe, moderate and mild | No |
| Scolese et al., 2020 [105] | Significant difficulty with a function such as seeing, hearing, mobility, cognition, self-care, and communication | Washington Group Short Set questions | A three-level categorical variable was created to assess severity: woman was coded as having no disability if she selected 'no-no difficulty' to all six questions, mild disability if she selected 'yes–some difficulty' or 'yes–a lot of difficulty' to any of the six questions and living with severe disability if she selected 'cannot do at all' to any of the six questions. | No |
| Leskosek et al., 2013 [146] | Injuries–any | Scale not specified; Items not included | No | No |
| Grossi et al., 2018 [47] | Temporomandibular disorders | Research Diagnostic Criteria for Temporomandibular disorders Axes I and II; items not included | No | No |
| Li et al., 2000 [76] | Any | Scale not specified; Items asked about disability onset, presence of multiple disabilities and presence of chronic pain | No | Yes–asked respondents about timing of disability onset |
| Zilkens et al., 2017 [173] | Injury, physical disability, intellectual disability | Scale not specified; Items not included | Yes–for injury: none, mild/ moderate and severe | No |
| Gunduz et al., 2019 [48] | Fibromyalgia; pain related to fibromyalgia | Pain measured using Visual Analog Scale [VAS]; respondents asked to determine level of pain | No | No |
| Leserman et al., 1998 [106] | 6 summary concepts: pain, non-GI symptoms; number of days spent in bed, number of medical visits, overall functional disability, psychological distress | Pain: Visual Analog Scale, pain severity averaged over 14 days (visual analog scale (0 to 100) indicating the amount of pain felt each day); b) Number of non-GI medical symptoms for the previous 6 months: derived from a list of 32 symptoms Number of days spent in bed (more than half the day) because of illness during the previous 3 months; Number of visits to see the doctor for medical problems in the past 6 months; Overall sickness-related functional disability: from the summary scale of the Sickness Impact Profile; Psychological distress: from the raw score of the Global Symptom Index of the SCL-90 (49). | No | Yes–previous 6 months |

(*Continued*)

**Table 4.** (*Continued*)

| Article | Types of disability | Scale/ measure of disability, specific items | Measure of severity of disability? Yes/ No; If Yes, how? | Time frame of disability specified? Yes/ No; If yes, what? |
|---|---|---|---|---|
| Martin et al., 2008 [107] | Functional impairment | Scale not specified; Women were asked about their functional status during the past month, in particular, whether "poor physical or mental health kept you from doing your usual activities, such as self-care, work, or recreation." | No | Yes–ask about past month time frame |
| Cascardi et al., 1996 [84] | Severe psychiatric disorders | DSM-III-R chart diagnosis No items | No | No |
| Chapple et al., 2004 [174] | Psychosis | Screening instrument: Derived from psychosis screening items of the Composite International Diagnostic Interview (CIDI) and the Psychosis Screening Questionnaire From the screen-positive individuals, selected for full assessment with Diagnostic Interview for Psychosis (DIP), which is a modified version of the Schedule for the Assessment of Clinical Neuropsychiatry (SCAN) Social and Occupational Functioning Assessment Scale (SOFAS) Items not included | Yes–Social and Occupational Functioning Assessment Scale (SOFAS) (American Psychiatric Association 1994). For the purposes of this study, the scores were dichotomised into "slight or no impaired social and occupational functioning" (SOFAS>69) and "poor social and occupational functioning" (SOFAS<70). Also, chronicity measured, Similarly, after using all information available, illness chronicity was coded according to five major categories: (a) single episode good recovery, (b) multiple episodes with good recovery, (c) multiple episodes with partial recovery, (d) continuous chronic illness with little/no deterioration, and (e) continuous chronic illness with clear deterioration. For the purposes of the present study, the categories were collapsed into two categories (non-chronic = a, b, c: chronic = d, e). | No |
| Goodman et al., 2001 [80] | Severe mental illness: schizophrenia spectrum disorders, major depression, bipolar disorder | Obtained psychiatric diagnoses using the Structured Clinical Interview for DSM-IV for 19.3% of the sample, and chart diagnosis for the remaining 80.7% of sample; No items | No | No |
| Hodgins et al., 2007 [81] | Schizophrenia, schizoaffective disorder, bipolar disorder, major depression or alcohol- or drug-induced psychosis | Two modules (Conduct Disorder and Antisocial Personality Disorder) of the Structured Clinical Interview for DSM–IV No items | No | No |
| Teplin et al., 2005 [82] | Severe mental illness: schizophrenia spectrum disorders, major depression, bipolar disorder | CIDI version 2.1, which provides DSM-IV and International Classification of Diseases, 10th Revision No items | No | No |
| Walsh et al., 2003 [83] | Psychosis | Comprehensive Psychopathological Rating Scale and Disability Assessment Scale Operational Criteria Checklist for Psychotic Illness Personality Assessment Schedule, Rapid version No items | No | No |
| Nosek et al., 2001 [160] | Physical disability that limits mobility and/or self care | Scale not specified; Items not included | No | No |
| Majeed-Ariss et al., 2020 [122] | Learning disability | Learning Disability Screening Questionnaire; items not specified | No | No |
| Acharya 2019 [147] | (1) physical disabilities, (2) mental disabilities, and (3)social disabilities | No scale Single item:: "Have you experienced any kind of injuries or disabilities after entering into the trafficking network?"; categorized responses into: physical disabilities (Mobility impairment, Hearing impairment, Visual impairment, Speaking impairment, Brain injury); Mental disabilities (Depression, Poor emotional condition) Social disabilities (Discrimination, Stigma) | No | No |

(*Continued*)

**Table 4.** (Continued)

| Article | Types of disability | Scale/ measure of disability, specific items | Measure of severity of disability? Yes/ No; If Yes, how? | Time frame of disability specified? Yes/ No; If yes, what? |
|---|---|---|---|---|
| Akyazi et al., 2018 [175] | Cognitive impairment | Diagnostic structured clinical interview DSM-IV TR axis 1 disorders (SCID-I); No items | No | No |
| Basile et al., 2016 [93] | Non-specific impairment: physical, mental or emotional impairment | No scale; Respondents were identified as having a disability if they answered "yes" to either of the following questions: "Are you limited in any way in any activities because of physical, mental, or emotional problems?" and "Do you now have any health problem that requires you to use special equipment, such as a cane, a wheelchair, a special bed, or a special telephone?" | No | Yes–more than one year |
| Breiding et al., 2015 [94] | Non-specific impairment: physical, mental or emotional impairment | No scale; Respondents were identified as having a disability if they answered "yes" to either of the following questions: "Are you limited in any way in any activities because of physical, mental, or emotional problems?" and "Do you now have any health problem that requires you to use special equipment, such as a cane, a wheelchair, a special bed, or a special telephone?" | No | Yes–more than one year |
| De Waal et al., 2017 [176] | Psychiatric disorders | Psychopathology was measured with the Brief Psychiatric Rating Scale-Expanded (BPRS-E) Items not included | No | No |
| Del rio Ferres et al., 2013 [135] | Visual, physical | 9 items on issues related to the disability, such as level of dependence and need of technical aids. Items obtained from previous studies on persons with disabilities and/or gender-based violence, from indicators proposed by the Observatorio Estatal de Violencia contra la Mujer (2007), local and national organizations of persons with disabilities | No | No |
| Jonas et al., 2013 [177] | Psychiatric disorders | Clinical Interview Schedule (Revised) and screening questionnaires for CMDs Procedure for identifying cases of psychosis involved two phases: in phase-one, respondents were screened for psychosis using the Psychosis Screening Questionnaire (PSQ) together with other criteria indicative of a psychotic episode (such as use of antipsychotic medication, receipt of a diagnosis and a stay in a psychiatric ward or hospital). Screen positive individuals were then interviewed with the Schedules for Clinical Assessment in Neuropsychiatry (SCAN) | No | No |
| Lacey et al., 2016 [116] | Mental disorders | World Mental Health Composite International Diagnostic Interview (WMH-CIDI) | No | No |
| Le et al., 2015 [178] | Not specified–chronic disease or disability | Not specified, "experience of a chronic disease or disability" asked in section of survey on socio-demographics | No | No |
| Macdowall et al., 2013 [123] | Not specified–longstanding illness or disability | Not specified, "longstanding illness or disability" assessed as a variable | No | No |
| New, 2019 [59] | Spinal cord-related disability | Spinal Functional Ability Scale; Items not included | No | No |

(*Continued*)

**Table 4.** (*Continued*)

| Article | Types of disability | Scale/ measure of disability, specific items | Measure of severity of disability? Yes/ No; If Yes, how? | Time frame of disability specified? Yes/ No; If yes, what? |
|---------|--------------------|--------------------------------------------|--------------------------------------------------------|-----------------------------------------------------------|
| Olofsson et al., 2015 [95] | Sensory (hearing and vision); physical | Seven questions were used to define persons with sensory or mobility impairments: <u>Visual and hearing impairments:</u> "Can you see and pick out normal text in a daily newspaper without difficulty?" and "Can you hear what is being said in a conversation between several people without difficulty?". <u>Mobility impairment:</u> "Can you run a fairly short distance?," "Can you climb stairs without difficulty?," "Can you take a fairly short walk?," and "Do you need aids or someone's help to move about outdoors? Another prerequisite or classification as having impairment in this study was to have answered "Yes, to a great extent" to the question: "Do these problems mean that your work capacity is diminished or hinder you in your other daily activities?" | No | No |
| Salahi et al., 2018 [152] | Mental disorders: Schizophrenia and related disorders, Bipolar affective disorder, Depressive disorder, Personality disorder | The Structured Clinical Interview for DSM-IV axis I disorders (SCID-I) and axis II disorders (SCID-II) used by psychiatrist to diagnose | No | No |
| Owens 2007 [179] | Psychiatric (including bipolar disorder and schizophrenia) | Determined by diagnosis at presentation at psychiatric emergency department | No | No |
| Coker et al., 2002 [180] | Chronic mental illness including schizophrenia; past injuries | Scale or measure not specified Items: <u>Past injuries:</u> "Have you ever sustained a serious injury, such as a spinal cord, neck, or head injury, that is disabling or interferes with your normal activities?"; <u>Chronic mental illness:</u> "Do you have a chronic mental health disease or condition, such as chronic depression or schizophrenia, that is disabling or interferes with your normal activities?"; <u>Impact of health conditions on daily life:</u> "To what extent did this disability or condition interfere with your normal activities in the past week?" | No | No |
| Dammeyer et al., 2018 [181] | Physical and mental | No specific scale Items: Participants were asked if they had "a long-term physical health problem or disability" and/or "one or more mental disorders". They were then asked to categorize their most serious physical and/or mental disability. They were also asked: "Would a stranger recognize within five minutes that you have a disability/health problem/mental disorder?" The response categories for the latter were coded by this study as "always" and "sometimes/never" | Yes—In the questionnaire, participants were asked if their main physical and/or mental disability was "minor or major". They were also asked: "Would a stranger recognize within five minutes that you have a disability/ health problem/mental disorder?" The response categories for the latter were coded by this study as "always" and "sometimes/ never". | No |
| Gibbs et al., 2017 [154] | Functional limitations in cognition, mobility, self-care, getting along, participation, and managing life activities | 12 item WHO Disability Assessment Schedule [WHODAS] Example items: "In the past 30 days, how much difficulty did you have in taking care of your household responsibilities?" | No | Last 30 days |
| Khalifeh et al., 2015 [49] | Severe mental illness: Schizophrenia and related disorders, Bipolar affective disorder, Depression & other mood disorders, Personality disorders | No specific scale Chronic mental disorder requiring ongoing secondary mental healthcare | No | 1 year or more |

(*Continued*)

**Table 4.** (*Continued*)

| Article | Types of disability | Scale/ measure of disability, specific items | Measure of severity of disability? Yes/ No; If Yes, how? | Time frame of disability specified? Yes/ No; If yes, what? |
|---|---|---|---|---|
| Milberger et al., 2003 [153] | Physical disabilities | Scale or items not included. Definitions: Physical disabilities were defined as those disabilities that result in functional impairment such as cerebral palsy, postpolio, spina bifida, amputation (bilateral upper limb, unilateral lower limb), rheumatic conditions (including rheumatoid arthritis and systemic lupus erythematosus), multiple sclerosis, spinal cord injury, traumatic brain injury, visual impairment, hearing impairment, and stroke. | No | No |
| Nannini 2006 [118] | Physical, visual, hearing, mental, emotional impairment | No specific scale Item: "Is victim physically challenged?" If the response was affirmative or the survivor volunteers the presence of a disability then staff asked if the survivor had any of the following disabilities: "visual impaired", "hearing impaired", "physically disabled", "mentally retarded", "emotional/mentally impaired", or "other, unspecified" | No | No |
| Weiner et al., 2013 [50] | Hearing | No scale specified; Drew sample from schools for deaf students | No | No |
| Nunes de Oliviera et al., 2013 [77] | Severe mental illness: 1) psychosis (schizophrenia and other non-bipolar psychosis); 2) bipolar disorders; 3) depressive disorders; 4) anxiety disorders; 5) substance abuse disorders; and 6) others (epilepsy, mental retardation, personality disorders or unknown). | For the psychiatric diagnosis, data were obtained from the medical charts and classified according to the International Classification of Diseases (ICD-10). | No | No |
| Ferraro et al., 2017 [117] | Mental disorders | Mini International Neuropsychiatric Interview (MINI); Items not included | No | No |
| Gilchrist et al., 2012 [182] | Mental disorders–major depressive disorder (independent and substance-induced), PTSD, antisocial personality disorder and borderline personality | Spanish Psychiatric Research Interview for Substance and Mental Disorders (PRISM); Items not included | No | No |
| Golding 1996 [108] | Physical functioning | Functional status assessed in two ways: number of days spent in bed because of illness and number of days of restricted activity Specific item: "During past 2 weeks/ 3 months, how many days did you stay in bed all or most of the day because of feeling physical pain or illness?" The Los Angeles interview asked: "During the last 2 weeks, how many days did physical illness or your physical condition make you cut down on things you would like to do, such as getting around or having visitors?" The North Carolina interview asked, "How many days altogether in the last 3 months were you kept from your usual activities because you weren't feeling well?' | No | No |
| Siqueira-Campos et al., 2019 [51] | Chronic pelvis pain | Scale/ items not included; measure of pain severity included (visual analogue scale) | Yes—intensity of pelvic pain was evaluated using a 10 point visual analogue scale, with zero representing the absence of pain and 10 the worst pain imaginable. | Item on duration of pain but not included |
| Sturup et al., 2011 [52] | Mental disorder–mood, psychosis, personality, dependence disorders | ICD-10 diagnosis Items not included | No | No |

(*Continued*)

**Table 4.** (Continued)

| Article | Types of disability | Scale/ measure of disability, specific items | Measure of severity of disability? Yes/ No; If Yes, how? | Time frame of disability specified? Yes/ No; If yes, what? |
|---|---|---|---|---|
| Thompson et al., 2019 [53] | Mental disorders–borderline personality disorder | Structured Clinical Interview for DSM-IV Axis I disorders (SCID-I/P) Bi-polar disorder [BPD] module of the Structured Clinical Interview for DSM-IV Axis II disorders (SCID-II) Items not included | No | No |
| Walker et al., 1997 [54] | Functional impairment—Fibromyalgia and rheumatoid arthritis—functional impairment | MHAQ—modified health assessment questionnaire; SF-36 physical functioning subscale; Items not included | No | No |
| Afe et al., 2017 [183] | Schizophrenia | Structured Clinical Interview for DSM- IV (SCID) Items not included | No | No |
| Anderson et al., 2016 [78] | Severe mental illness | ICD-10 diagnosis | Yes–hospitalization used as a marker for severity | No |
| Beydoun et al., 2017 [127] | Mental disorders–including psychosis and schizophrenia | International Classification of Disease, 9th Revision, Clinical Modification (ICD-9-CM) | No | No |
| Du Mont et al., 2014 [184] | Activity limitations due to mental health condition | No scale specified Items: "Are your daily activities at home, work, school or any other area limited by a psychological, emotional or mental health condition?" Response categories included always/often, sometimes, or no. | Yes—The severity of mental health-related activity limitations; responses always/ often = severe activity limitations, sometimes indicating less severe. | No |
| Gil-Llario et al., 2019 [85] | Intellectual disabilities | No scale/ items Intellectual disability defined by professional report regarding intellectual age of respondent | Yes–mild and moderate, as rated by professional | No |
| Gold et al., 2012 [136] | Mental health disorders—severe (related to suicide); mood disorders (unipolar or bipolar disorder, and dysthymia), anxiety disorders (anxiety, posttraumatic stress disorder, obsessive–compulsive disorder), schizophrenia, other mental health disorders (such as eating disorders, attention deficit hyperactivity disorder, mental retardation and autism, among others) and unknown disorder. | Data drawn from United States National Violent Death Reporting System (NVDRS)–mental health is measured where a current prescription for an antidepressant or other psychiatric medication is evidence for both current mental health problem and current treatment. | No | No |
| Gonzalez Cases et al., 2014 [69] | Severe mental illness | Global Assessment of Functioning (GAF) Items not included | No | Previous 6 months |
| Helfrich et al., 2008 [137] | Impairments and functional limitations due to mental disorder | Selected items from National Health Information Survey Some items included: Identifies as having a disability, mental disorder Interfere with attending work or school or managing daily activities; Trouble finding or keeping job or doing job tasks due to disorder; Unable to work or limited ability to work due to disorder | No | No |
| Khalifeh et al., 2015 [55] | Severe mental illness | ICD-10 Items not | No | No |
| Kmett et al., 2018 [86] | Severe mental illness | DSM-III-R (psychiatrist report) and participant report–respondents asked "if they felt "psychiatrically unwell," with yes/ no responses dichotomously coded. | Yes—Brief Psychiatric Rating Scale (BPRS; Overall & Gorham, 1962) was administered. The BPRS is an 18-item clinical assessment that rates the severity of psychiatric symptoms from the previous week, using a scale from 1 (no symptoms present) to 7 (high level of symptoms) | No |
| Lundberg et al., 2015 [56] | Severe mental illness | Clinical diagnosis | No | No |
| McPherson et al., 2007 [43] | Severe mental illness—affective disorders (major depression and bipolar disorder without psychotic features) and psychotic disorders (schizophrenia, schizoaffective disorder, major depression with psychotic features, and bipolar disorder with psychotic features) (coded as 0, psychotic disorders, or 1, affective disorders). | Diagnostic Interview Schedule, DSM-IV; Psychiatric symptomatology was measured by using the Colorado Symptom Index; items not included | No | No |

(*Continued*)

**Table 4.** (Continued)

| Article | Types of disability | Scale/ measure of disability, specific items | Measure of severity of disability? Yes/ No; If Yes, how? | Time frame of disability specified? Yes/ No; If yes, what? |
|---|---|---|---|---|
| Meekers et al., 2013 [57] | Mental health disorders—depression, anxiety, psychogenic seizures, and psychotic disorders. | Self Reporting Questionnaire 20 (SRQ-20), a mental health screening tool developed by WHO. The SRQ-20 has also been validated as a screening tool for psychological morbidity. The BDHS Also includes selected mental health questions from the extended version of the Self-Reported Questionnaire, the SRQ-25; four additional questions aimed at identifying probable psychosis and one question about psychogenic non-epileptic seizures, or convulsions. | No | No |
| Nguyen et al., 2017 [58] | Severe mental illness | Items or measure not included; need for services taken as indication of severe mental illness | No | No |
| Racic et al., 2006 [185] | Mental illness–depression, anxiety and dementia | Scale not specified; Items not included | No | No |
| Riley et al., 2014 [186] | Psychiatric disorders–including 39 psychiatric diagnoses, including anxiety disorders (panic attack, specific phobia, social phobia, agoraphobia, generalized anxiety disorder, PTSD), mood disorders (major depressive episode, dysthymia, hypomanic episode, manic episode), psychotic disorders (schizophrenia, schizophreniform disorder), substance-related disorders | Diagnostic Interview Schedule for DSM-IV Items not included | No | No |
| Santaularia et al., 2014 [109] | Physical, mental or emotional | Scale not specified Items: "Are you limited in anyway in any activities because of physical, mental, or emotional problems?" and "Do you now have any health problem that requires you to use special equipment, such as a cane, a wheelchair, a special bed, or a special telephone? (Include occasional use or use in certain circumstances)". | No | No |
| Schofield et al., 2013 [187] | Non-specific impairment | Item: "Do you regularly need help with daily tasks because of long-term illness, disability, or frailty (for example, personal care, getting around, preparing meals)?" | No | No |
| Shah et al., 2018 [188] | Psychosis | Psychotic experiences were assessed through the World Health Organization (WHO) Composite International Diagnostic Interview (CIDI) psychosis screen; Items: Participants were asked if any of the following had happened in the past four months: (1) "A feeling something strange and unexplainable was going on that other people would find hard to believe?", (2) "A feeling that people were too interested in you or that there was a plot to harm you?", (3) "A feeling that your thoughts were being directly interfered or controlled by another person, or your mind was being taken over by strange forces?", and (4) "An experience of seeing visions or hearing voices that others could not see or hear when you were not half asleep, dreaming, or under the influence of alcohol or drugs?" | No | Last 12 months |
| Anderson et al., 1993 [124] | Dissociative disorders | Dissociative Experience Scale (DES) and Dissociative Disorders Interview Schedule (DDIS) Included not included | No | No |

*(Continued)*

**Table 4.** (*Continued*)

| Article | Types of disability | Scale/ measure of disability, specific items | Measure of severity of disability? Yes/ No; If Yes, how? | Time frame of disability specified? Yes/ No; If yes, what? |
|---|---|---|---|---|
| Institute of Statistics, et al., 2018 [129] | Chronic physical disabilities, Congenital deformities, Hearing impairment Motor disabilities, Mobility impairment, Polio, Speech impairment, Visual impairment | No scale; single question: "Do you suffer from a chronic disability?" (Yes/ No) "What type of chronic disability do you have?" | No | No |
| National Institute of Statistics et al., 2015 [189] | Non-specific impairment: difficulties with seeing, hearing, walking or climbing stairs, remembering or concentrating, performing self-care, or communicating | Washington Group Short-Set questions Disability defined as some difficulty, a lot of difficulty, cannot do for any domains | No | No; N/A |
| Institut National de la Statistique et al., 2012 [190] | Motor impairment Absence of limbs (or parts of limbs)–lower or upper, Distortion/difficulty using limbs lower or upper, Visual impairment, Hearing impairment, Language or speech impairment, Loss of certain extremities of the body, behavioural disorders | Scale not included; Items: Is there someone in your household who is missing a part of the body, for example, a hand, an arm, a foot or a leg? Is there anyone in your household who suffers from deformity of the upper or lower limbs and which cannot or has difficulty walking and / or standing? Is there anyone in your household who hardly sees or who is blind? Is there anyone in your household who can hardly hear or who is deaf? Is there anyone in your household who has serious difficulty speaking or who is mute? Is there someone in your household who is missing certain extremities of the body, such as fingertips, toe, nose or ear? Is there someone in your household who has behavioural difficulties? | Yes–Partial or full disability determined by questions such as: "Does (NAME) have only difficulties in using his arms or legs, or is it (NAME) cannot use his arms or legs at all?" | No; N/A |
| Institut National de la Statistique et al., 2015 [191] | Absence of limbs (or parts of limbs)–lower or upper, Distortion/difficulty using limbs lower or upper, Visual impairment, Hearing impairment, Language or speech impairment, Loss of certain extremities of the body, behavioural disorders | Scale not included; Items: Is there someone in your household who is missing a part of the body, for example, a hand, an arm, a foot or a leg? Is there anyone in your household who suffers from deformity of the upper or lower limbs and which cannot or has difficulty walking and / or standing? Is there anyone in your household who hardly sees or who is blind? Is there anyone in your household who can hardly hear or who is deaf? Is there anyone in your household who has serious difficulty speaking or who is mute? Is there someone in your household who is missing certain extremities of the body, such as fingertips, toe, nose or ear? Is there someone in your household who has behavioural difficulties? | Yes Partial or full disability determined by questions such as: "Does (NAME) have only difficulties in using his arms or legs, or is it (NAME) cannot use his arms or legs at all?" | No; N/A |
| Institut Haïtien de l'Enfance (IHE) et al., 2018 [192] | Vision, hearing, communication, cognitive functions, walking and having the autonomy for washing or dressing one's self | Washington Group Short-Set questions | No | No; N/A |
| Institut National de la Statistique (INSTAT) et al., 2019 [193] | Difficulty seeing, difficulty hearing, difficulty to communicate, difficulty remembering or concentrating, difficulty walking or climbing stairs, difficulty in washing or getting dressed | Washington Group Short-Set questions | No | No; N/A |
| Agence Nationale de la Statistique et de la Démographie (ANSD) et al., 2019 [194] | Difficulty seeing, difficulty hearing, difficulty to communicate, difficulty remembering or concentrating, difficulty walking or climbing stairs, difficulty in washing or getting dressed | Washington Group Short-Set questions | No | No; N/A |

(*Continued*)

**Table 4.** (*Continued*)

| Article | Types of disability | Scale/ measure of disability, specific items | Measure of severity of disability? Yes/ No; If Yes, how? | Time frame of disability specified? Yes/ No; If yes, what? |
|---|---|---|---|---|
| National Department of Health et al., 2016 [195] | Difficulty seeing, difficulty hearing, difficulty to communicate, difficulty remembering or concentrating, difficulty walking or climbing stairs, difficulty in washing or getting dressed | Washington Group Short-Set | No | No; N/A |
| Uganda Bureau of Statistics et al., 2018 [196] | Difficulty seeing, difficulty hearing, difficulty to communicate, difficulty remembering or concentrating, difficulty walking or climbing stairs, difficulty in washing or getting dressed | Washington Group Short-Set | No | No; N/A |
| General Directorate of Statistics et al., 2018 [197] | Difficulty seeing, difficulty hearing, difficulty to communicate, difficulty remembering or concentrating, difficulty walking or climbing stairs, difficulty in washing or getting dressed | Washington Group Short-Set In addition: Have you ever been told by a doctor or other health worker that you have any other chronic disease, that is, any other disease that is long lasting? Are you receiving any treatment for [CHRONIC DISEASE]? | No | No; N/A |
| The Gambia Bureau of Statistics et al., 2014 [198] | Difficulty seeing, hearing, using legs | If any household member(s) age 7 to age 69 had any form of disability and, if so, what type of disability: Eyesight—Does (NAME) wear glasses? Does (NAME) have difficulty seeing during the day (even if she / he is wearing glasses)? Hearing—Does (NAME) use a hearing aid? Does (NAME) have difficulty hearing (even if she / he is using the hearing aid)? Legs—Does (NAME) have any difficulty using his / her legs even for simple activities such as walking or climbing up the stairs? Does (NAME) use a cane or crutches or wheelchair? | No | No; N/A |
| National Institute of Population Studies et al., 2019 [199] | Difficulty seeing, difficulty hearing, difficulty to communicate, difficulty remembering or concentrating, difficulty walking or climbing stairs, difficulty in washing or getting dressed | Washington Group Short-Set questions | No | No; N/A |
| Rand et al., 2007 [200] | Cognitive functioning limitation; Sensory limitation; Physical limitation; Self-care limitation; Going-outside-home limitation; Employment limitation | National Crime Victimization Survey adapted disability questions from U.S. Census American Community Survey; Do you have any of the following long-lasting conditions: (a) Blindness, deafness, or a severe vision or hearing impairment? (b) A condition that substantially limits one or more basic physical activities such as walking, climbing stairs, reaching, lifting, or carrying? Because of a physical, mental, or emotional condition lasting 6 months or more, do you have any difficulty in doing any of the following activities: (a) Learning, remembering, or concentrating? (b) Dressing, bathing, or getting around inside the home? (c) Going outside the home alone to shop or visit a doctor's office? (d) Working at a job or business? | No | Yes; more than 6 months |
| Carlile 1991 [70] | Psychiatric diagnoses: major affective disorder, schizophrenias, personality, anxiety and somatoform disorders | DSM-III diagnosis; diagnoses made by multidisciplinary team | No | No |
| Post et al., 1980 [130] | Psychiatric disorders | Psychiatric diagnosis; scale not specified Items not included | No | No |

(*Continued*)

**Table 4.** (Continued)

| Article | Types of disability | Scale/ measure of disability, specific items | Measure of severity of disability? Yes/ No; If Yes, how? | Time frame of disability specified? Yes/ No; If yes, what? |
|---|---|---|---|---|
| Sansone et al., 2007 [71] | Psychiatric disorders | Psychiatric diagnosis; scale not specified Items not included | No | No |
| Zanarini et al., 1999 [201] | Borderline personality disorder; other Axis 2 disorders | The Structured Clinical Interview for DSM-III-R Axis I Disorders (SCID-I; the Revised Diagnostic Interview for Borderlines; and the Diagnostic Interview for DSM-III-R Personality Disorders (DIPD-R) Items not included | No | No |
| Bengtsson-Tops et al., 2012 [79] | Psychosis | Psychiatric diagnosis; scale not specified Items not included | No | No |
| Ford 2008 [72] | Severe mental illness—schizophrenia, schizoaffective disorder, psychosis NOS, bipolar disorder, or major depressive disorder with psychotic features. | Chart diagnoses; Structured Clinical Interview for the Diagnostic and Statistical Manual of Mental Disorders Items not included | No | No |
| Friedman et al., 2011 [44] | Severe mental illness | Psychiatric diagnosis confirmed based on the Structured Clinical Interview for the Diagnostic and Statistical Manual of Mental Disorders Items not included | No | No |
| Goodman et al., 1995 [202] | Severe mental illness | Diagnosed–type of diagnosis or method not noted | No | No |
| Leithner et al., 2009 [203] | Gynecological problems with psychosomatic reason | Women are self-referred, referred by outpatient or inpatient gynaecological units, or referred by their gynaecologists in private practice; no items listed | No | No |
| Lipschitz et al., 2009 [204] | Psychiatric diagnoses–including mood disorders, anxiety disorders, schizophrenia | Chart diagnoses No items included | No | No |
| Morgan et al., 2010 [205] | Psychiatric disorders | Symptom Checklist 90-Revised (SCL90-R) normalized for outpatients. After the routine assessment interview, DSM-III-R diagnosis was given to each patient by a psychiatric resident who was blind to the information contained in the packet. No items included | No | No |
| Surrey et al., 1990 [206] | Psychiatric disorders | DSM III-R diagnoses from charts No items included | No | No |
| Swett et al., 1991 [207] | Psychiatric disorders | DSM III-R diagnoses from charts No items included | No | No |
| Briere et al., 1997 [148] | Psychiatric disorders—depressive disorders (e.g., major depression, dysthymic disorder), anxiety disorders (e.g., generalized anxiety disorder, adjustment disorders), non-manic psychotic disorders (e.g., schizophrenia, psychosis NOS), and manic disorders. | DSM-III-R diagnoses from charts No items included | No | No |
| Bengtsson-Tops et al., 2005 [208] | Psychiatric disorders | Scale not specified; Items not included | No | No |
| Yellowlees et al., 1994 [149] | Psychiatric disorders | DSM III-R diagnoses No items included | No | No |
| Coker et al., 2005 [110] | Physical or mental conditions—disabilities were grouped in the following ways: generalized chronic pain; disabilities associated with the nervous system; disabilities associated with brain or head trauma; asthma or respiratory condition; mental illnesses; chronic disease disabilities; stroke, thrombosis, diabetes, cancer, chronic kidney or bladder infection or diseases; blindness or glaucoma; and autoimmune diseases | Modification of the National Health Interview Survey to ascertain whether women have ever been told they had a range of mental and physical health conditions. Participants also asked if they had ever had a chronic disability that prevented them from working outside the home or from doing housework if they were a homemaker. | No | No |

(*Continued*)

**Table 4.** (Continued)

| Article | Types of disability | Scale/ measure of disability, specific items | Measure of severity of disability? Yes/ No; If Yes, how? | Time frame of disability specified? Yes/ No; If yes, what? |
|---|---|---|---|---|
| Brownlie et al., 2007 [45] | Language impairment | At age 5, language impairment was defined as one or more of the following: (i) Test of Language Development [TOLD]—2, Spoken Language Quotient, 1 SD below the mean; (ii) any TOLD language subtest (not including Word Articulation and Word Discrimination), 2 SD below the mean; (iii) Peabody Picture Vocabulary Test-Revised, 1 SD below the mean; (iv) Goldman-Fristoe-Woodcock Auditory Memory Tests, 1 SD below the mean on both memory for content and memory for sequence subtests. | No | No |
| Diaz-Olavarrieta et al., 1999 [209] | Chronic neurologic disorders | Scale not specified; Items not included | No | No |
| Findley et al., 2016 [143] | Hard of hearing/deaf; Blind/limited vision; Learning disability with and without ADD/ADHD; Neurological; Psychological/psychiatric; Orthopedic/ mobility; Chronic illness | Scale not specified; Items not included Participants were students registered with university Office of Disability Services | No | No |
| Giraldo-Rodriguez et al., 2015 [210] | Limitations in activities of daily living— (walking or mobility, visual, hearing, speech or communication, attention or learning, self-care) | Washington Group Short Set questions Reported if participants reported difficulty on none, one, two or more activities | No | No |
| Longobardi et al., 2018 [144] | Physical, motor, sensory, intellectual | Scale not specified; Items not included Participants attended disability centre | No | Participants asked if born with disability or not |
| Ministry of Labor, Invalids and Social Affairs et al., 2020 [120] | Limitations in activities of daily living— (walking or mobility, visual, hearing, speech or communication, attention or learning, self-care) | Washington Group Short Set questions | 3 categories of disability created: Disability level 1: one domain/ question is coded some difficulty or a lot of difficulty or cannot do at all Disability level 2: 2 domains/ questions are coded some difficulty or any 1 domain is coded a lot of difficulty or cannot do at all Disability level 3: 1 domain/question is coded a lot of difficulty or cannot do at all | No |
| National Statistics Office et al., 2018 [121] | Visual, hearing, walking, mental, motor, communication | Washington Group Short Set questions | Yes–moderate defined as some difficulty, severe defined as a lot of difficulty or cannot do at all | No |
| National Commission for Women and Children, 2017 [119] | Visual, hearing, walking, mental, motor, communication | Washington Group Short Set questions | No | No |
| CREA 2012 [211] | Physical or sensory impairment | Scale not specified; Items not included Women self-identified as disabled | No | No |
| SINTEF 2016a [131] | Visual, hearing, walking, mental, motor, communication | Washington Group Short Set questions | No | No |
| Bureau of Justice Statistics 2017 [150] | Hearing (deafness or serious difficulty hearing), vision (blindness or serious difficulty seeing, even when wearing glasses), cognitive (serious difficulty in concentrating, remembering, or making decisions because of a physical, mental, or emotional condition), ambulatory (difficulty walking or climbing stairs), self-care (a condition that causes difficulty dressing or bathing), and independent living (physical, mental, or emotional condition that impedes doing errands alone, such as visiting a doctor or shopping) | Adopted survey questions from the U.S. Census Bureau's American Community Survey (ACS) to identify crime victims with disabilities. Items were:Are you deaf or do you have serious difficulty hearing? Are you blind or do you have serious difficulty seeing, even when wearing glasses?Because of a physical, mental, or emotional condition, do you have serious difficulty: • concentrating, remembering, or making decisions • walking or climbing stairs • dressing or bathing? Because of a physical, mental, or emotional condition, do you have difficulty doing errands alone, such as visiting a doctor's office or shopping? | No | No |

(*Continued*)

**Table 4.** (Continued)

| Article | Types of disability | Scale/ measure of disability, specific items | Measure of severity of disability? Yes/ No; If Yes, how? | Time frame of disability specified? Yes/ No; If yes, what? |
|---|---|---|---|---|
| SINTEF 2016b [132] | Visual, hearing, walking, mental, motor, communication | Washington Group Short Set questions | No | No |
| Uganda Bureau of Statistics 2018 [212] | Visual, hearing, walking, mental, motor, communication | Washington Group Short Set questions | No | No |
| Schröttle et al 2013 [156] | Severe long-term movement, hearing, visual, cognitive, and/or psychological impairments, and/or long-term-impairments resulting from chronic diseases Purposively selected participants who were blind, deaf or severely disabled | Scale not specified; Items not included Self-report or living in institutions for persons with disabilities | No | No |
| Instituto Nacional de Estadistica e Informatica, 2014 [213] | Non-specific impairment, including vision, hearing, communication | Name of scale not specified Items: Do you have any permanent limitations or difficulties: 1. To move, walk, use your arms and / or legs? 2. To see, despite wearing glasses? 3. To hear, even when wearing headphones? 4. To talk or communicate? 5. To understand or learn (focus and remember)? 6. Do you have any other permanent limitations? | No | No |
| Minsalud, Profamilia, 2015 [214] | Non-specific impairment, including vision, hearing, communication | Scale not listed, indicates that adapted from Washington Group questionnaire: Items: You would say that given your physical and mental condition can you: 1. Hear a voice or sounds? 2. Talk or converse? 3. See up close, far or around? 4. Move the body, walk, go up or down stairs? 5. Grasp or move objects with your hands? 6. Understand, remember or make decisions for yourself? 7. Eat, dress or bathe by yourself? 8. Relate or interact with other people? 9.Do daily tasks without showing cardiac, respiratory or renal problems? 10. Of the above difficulties, which is the one that most affects your daily performance | No | No |
| Estudios Sociales y Demográficos (CESDEM) et al., 2014 [215] | Non-specific impairment, including vision, hearing, communication | Scale not listed Items: Do you have any of the following impairments or disabilities: a. Total blindness b. Partial blindness c. Total deafness d. Partial deafness e. Can't speak or make any sound (muteness) f. Speak with difficulty g. Cannot walk or does it with great difficulty (with the help of devices) h. Cannot or has difficulty grasping objects i. Other impairment / disability | Yes–total or partial included | No |
| Secretaría de Salud [Honduras] et al., 2013 [216] | Non-specific impairment, including vision, hearing, communication | Scale not listed Any of the following health problems: a) Difficulties in vision? b) Problems moving or walking? c) Hearing or hearing problems? d) Mental retardation? | No | No |

the body of literature included covered a wide range of settings, research questions, and types of violence and disability, there were some notable gaps. Very few studies compared how women with different types of disabilities experience violence differently; studies often either focused on one type of disability or assessed a number of different types of disability as one group for the purposes of analysis. This represents a significant gap, as forms and perpetrators of violence and risk and protective factors for violence may vary based on different types of disability [11].

Studies that did include analysis of different types of disability identified important relationships with implications for policy and programmatic response. For example, a study of IPV, disability and depression amongst post-partum women in South Africa identified different relationships between different types of functional limitations and IPV, with mobility limitations being the only specific type of functioning limitation associated with increased IPV [154]. Nannini's study provided in-depth analysis of perpetrator and context of, and response to violence, for women with physical, visual, hearing, mental and cognitive disabilities, identifying different patterns in seeking services amongst women with cognitive disabilities [118]. Olofsson identified women with auditory or visual disability as at higher risk of violence-exposure compared to non-disabled women [95]. Measurement of and policy and programmatic response to address violence against women with disabilities needs not only to recognize that women with disabilities are at increased risk for violence victimization, but also respond to how specific disabilities relate to violence experiences and barriers to accessing services and reporting violence.

In addition, we identified very few studies that explored violence risk for women with disabilities alongside other vulnerabilities, for example, migration status or age. One study that focused on women from Puerto Rico in the USA did not include a comparison group [44], while a study of black Caribbean women in the USA examined the influence of generational status on physical and mental health outcomes associated with severe IPV [116]. Intersectionality is an important perspective on violence against women, positing that women are differently situated and therefore experience inequalities–including exposure to violence– differently. Gender, class, disability, ethnic status and age can all intersect and influence women's experiences of violence [7], and the current evidence-base does not adequately shed light on these intersections.

We found that more than a third of included manuscripts or reports focused only on IPV, and within these studies, research that compared women with and without disabilities consistently found elevated risk for IPV amongst women with disabilities. However, this finding does not indicate that women with disabilities face the greatest risk for or severity of violence within intimate partner relationships; rather, that the focus of study design, research questions and measurement instruments has been IPV. Women with disabilities likely face significant violence risks from family members apart from intimate partners and caregivers (paid and/ or family members), and within institutions. One included study indicated that women with disabilities faced increased risk of violence perpetrated by caregivers and decreased risk of violence perpetrated by intimate partners compared to women without disabilities [118]. In a study that included a measure of violence specific to women with disabilities, that form of violence was equally likely to be perpetrated by an intimate partner, a care provider, or a health professional [141]. The question of whether violence perpetrated by an intimate partner is the most prevalent or pervasive form of violence for women with disabilities needs further exploration.

The results of our scoping review did not shed light on duration and severity of violence experienced by women with disabilities. Researchers have hypothesized that women with disabilities may be more likely to experience violence for longer given barriers to leaving an

intimate partner relationship or reliance on an abusive care-giver [13]. Within our included reports and manuscripts, data are not available to support this hypothesis and this is an important gap in the evidence-base. While we found a large set of studies that addressed the intersection of disability and violence against women, there are numerous evidence gaps remaining that need to be addressed through high-quality qualitative and quantitative primary research and analyses of existing data.

We found that violence was assessed using a range of measurement instruments. Three widely used instruments–the Conflict Tactics Scale, the WHO Multi-Country Study Instrument, and the DHS Domestic Violence Module–were utilized in a total of n = 61 included manuscripts or reports. Overall, n = 114 reports or manuscripts used acts-based questions to assess all forms of violence measured, which is the gold-standard approach for research on violence against women. Some studies utilized a single item to assess sexual violence and acts-based questions to assess physical violence. Given how stigmatized sexual violence is, and low disclosure rates, this is likely to result in significant underestimation of sexual violence [1]. The design and focus of the studies that only included single item measures of violence varied (n = 13 used a single item for any measure of violence) and we cannot conclude that higher quality violence measurement was included in specific types of research approaches. The snapshot of violence measurement that includes or is included in disability research that this scoping review provides is mixed. While a majority of studies included acts-based measures, this proportion was not as high as might be expected given the overall advances in quality measurement in the field of research on violence against women.

It is evident that dimensions of the different violence experiences of women with disabilities may be overlooked in dominant research approaches to violence against women [10]. Women with disabilities may be excluded from research for reasons pertaining to methodology–for example, household surveys exclude women with disabilities who may be living in institutions or group housing [155] or data collection methods, e.g., telephone surveys may exclude women with hearing disabilities [156]; reasons pertaining to research ethics–for example, that women with cognitive disabilities may be unable to give informed consent [155]; and stereotypes and misconceptions about women with disabilities, including perceptions of asexuality, influencing the types of research projects planned, funded and implemented [13]. We found few studies that took specific measures to ensure inclusion of women with disabilities by developing or adapting specific data collection methods. Two manuscripts identified in our review that did not meet inclusion criteria reported on development and utilization of a survey of violence against women with disabilities using audio computer-assisted self-interviews [ACASI] specifically designed, based on extensive community consultations, to be accessible for women with disabilities [157, 158]. One included study utilized a computerized sign language survey to ensure accessibility for women with hearing disabilities [159]. In addition, few included studies took measures to adapt ethics procedures, such as delivery or design of informed consent processes, to ensure that women with disabilities for whom these procedures could form a barrier to participation in research (analysis not shown). Overall, data collection and ethics procedures need to be developed, piloted and implemented, with significant input from women with various types of disabilities, to ensure inclusion of women with disabilities in research on violence against women.

The prevalence of violence against women with disabilities may be underestimated in the studies we explored as violence specific to women with disabilities was only assessed in a small number (n = 11) of manuscripts or reports that we reviewed. One study indicated that 20% of violence prevalence would have been excluded had their violence instrument not included disability-specific violence items [160]; beyond this finding, there is limited evidence indicating

the extent to which violence against women with disabilities is underestimated due to lack of disability-specific violence items in the majority of violence against women research.

Our results on disability measurement in this body of literature indicate a very broad range of measurement instruments utilized. Despite the variety of available instruments, however, there is still no gold standard instrument that is easy to administer and can comprehensively capture the experience of disability. Different measurement approaches found in this review were measures of functioning (n = 75), a single item such as "Do you have a disability?" (n = 15) and definition based on a diagnosed or self-reported health condition (n = 67), all of which have certain limitations. For example, using a single item to define who has disability can lead to underreporting because the term "disability" is often associated with an assumption of a severe condition. Relying on a medical diagnosis of a health condition or impairment can also lead to under-reporting as those without access to health services may not have been diagnosed by a professional. Functioning instruments can assess only certain domains and exclude other relevant everyday functioning areas. The Washington Group Questions, which are widely used to assess disability, for example, do not address mental health functioning, and tend to generally identify individuals with more significant levels of disability and miss those with less severe disabilities.

There are several approaches towards research on violence against women with disabilities that may serve to improve quality and availability of evidence. Integration of disability measures into existing population-based violence against women studies, or leveraging population-based household surveys by disaggregating violence against women findings by disability status, may be feasible approaches. Integration of disability measures into violence against women surveys does have some limitations, such as excluding women not living on households or unable to provide informed consent for reasons relating to disability. Another approach is to integrate violence against women measures within disability surveys. In our grey literature search, we did not identify a large number of disability specific surveys that included women's violence experiences; some surveys that did include violence exposure did not provide adequate sex disaggregation, as required for inclusion in this scoping review. Integration of violence measurement within disability surveys may be a more cost-effective and time-efficient way of collecting representative data on these topics. However, challenges include needing to ensure that appropriate ethics and safety measures for violence against women research are incorporated into the disability surveys [8].

Lack of sex disaggregation within disability-focused research stymies further understanding of violence against women with disabilities within the disability-focused research. A large proportion of national disability surveys were not included in the scoping review as while some did include short measures of violence, the data were not disaggregated by sex. One of our inclusion criteria for studies including men and women was that "studies including men and women with disability were included if sex-specific analyses were done," and one of the primary reasons for exclusion when we screened at the full-text level was lack of sex disaggregation of data. This indicates that there are considerable bodies of evidence and datasets available that could further shed light on the experiences of women with disabilities. However, without adequate sex-disaggregation of data, these studies were excluded from this current scoping review. An important step in improving the evidence-base, even prior to designing and conducting further primary data collection, is to conduct adequate sex-disaggregation of available data, which is recommended in United Nations guidance on gender mainstreaming policies [161]. In addition, disability-disaggregated data is fundamental for our understanding on how the inequalities that people with disability face globally can be addressed. Disaggregated data can reveal increased risks persons with disabilities may face as well as root causes of exclusion of persons with disabilities from various areas of life or highlight where inequalities

exist [21]. Such disaggregation is essential for countries to develop evidence-based policies to monitor how existing barriers are addressed, to measure progress towards national targets and the SDGs, and to plan future policy priorities.

Our findings indicate limitations in measurement and evidence-base, which impact policy-makers and programmers given the overall prevalence of violence against women with disabilities is unknown, which limits development and implementation of effective policies and support services. In addition, the lack of evidence concerning how different types of disabilities operate as risk factors compared to others, and can create different barriers and enablers for women who experience violence seeking support, hampers effective programming tailored to specific needs.

## Strengths and limitations

One of the strengths of this scoping review is its breadth and inclusiveness. In seeking to develop a snapshot of the field of measurement on disabilities and violence against women, we conducted a broad literature search, including national statistics, grey literature and published surveys (DHS and national VAW surveys). Based on our inclusion criteria, we identified a wide range of quantitative evidence. This resulted in a comprehensive overview of the existing literature, yet given the inclusiveness of the scoping review, the picture is of several disparate and distinct bodies of literature. We plan to undertake sub-analyses of the data identified in this study, for example, comparing types of violence and perpetrators in studies that explored multiple types of violence and a sub-analysis of prevalence of violence identified in studies that assessed disability based on functioning measure. One of the limitations at this stage is that we focus on descriptive analysis and do not present quantitative assessment of the relationship between violence and disability. We did not identify a clear and comprehensive way to categorise disability measurement beyond the descriptive analysis we present, given the purpose of disability measures within the included studies was so broad. Therefore at this stage we cannot address questions such as whether disability measurement approach was correlated with levels of violence identified. Several key questions–such as how and why different vulnerabilities intersect with disability and result in violence victimization, the perceptions and experiences of women with disabilities about violence prevention and response programs, and the role of disability discrimination in driving levels of violence–are not adequately addressed in the quantitative literature. A scoping or systematic review of qualitative literature is an important complement to this work.

## Conclusion

Globally, violence against women remains unacceptably high, and women with disabilities may be at higher risk of exposure to violence, as well as being exposed to different forms of violence. Disability as a risk factor for exposure to violence is poorly understood, and this scoping review provides a first step towards understanding the current status of measurement of both violence and disability within relevant bodies of literature. Our findings indicate several gaps in evidence, including lack of comparison of how women with different types of disabilities experience violence differently, and indicate future directions for research and analysis.

## Supporting information

**S1 Appendix. Preferred Reporting Items for Systematic reviews and Meta-Analyses extension for Scoping Reviews (PRISMA-ScR) checklist.**
(DOCX)

**S2 Appendix. Data extraction variables.**
(DOCX)

## Acknowledgments

We appreciate Dr. Ana Ortega-Avila's support in extracting data in Spanish language reports.

## Author Contributions

**Conceptualization:** Sarah R. Meyer, Claudia García-Moreno.

**Data curation:** Sarah R. Meyer, Heidi Stöckl, Cecilia Vorfeld, Kaloyan Kamenov.

**Formal analysis:** Sarah R. Meyer, Heidi Stöckl, Kaloyan Kamenov.

**Funding acquisition:** Claudia García-Moreno.

**Investigation:** Sarah R. Meyer.

**Methodology:** Sarah R. Meyer, Claudia García-Moreno.

**Supervision:** Claudia García-Moreno.

**Writing – original draft:** Sarah R. Meyer.

**Writing – review & editing:** Heidi Stöckl, Cecilia Vorfeld, Kaloyan Kamenov, Claudia García-Moreno.

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
