## [Decision Letter · Decision Letter 0]

15 Dec 2021

PONE-D-21-26391

A scoping review of measurement of violence against women and disability

PLOS ONE

Dear Dr. Meyer,

Thank you for submitting your manuscript to PLOS ONE. After careful consideration, we feel that it has merit but does not fully meet PLOS ONE’s publication criteria as it currently stands. Therefore, we invite you to submit a revised version of the manuscript that addresses the points raised during the review process.

**The Reviewer greatly enjoyed the manuscript and suggested only a few corrections. I invite, therefore, the authors to address the points raised by the Reviewer to make the manuscript suitable for publication. **

We look forward to receiving your revised manuscript.

Kind regards,

Stefano Federici, Ph.D.

Academic Editor

PLOS ONE

“This work was funded under the UNWomen-WHO Joint Programme on Violence against Women Data, funded by the Foreign, Commonwealth and Development Office. The funders had no role in study design, data collection and analysis, decision to publish, or preparation of the manuscript.”

“This work was funded under the UNWomen-WHO Joint Programme on Violence against Women Data, funded by the Foreign, Commonwealth and Development Office. The funders had no role in study design, data collection and analysis, decision to publish, or preparation of the manuscript.”

Additional Editor Comments:

The Reviewer greatly enjoyed the manuscript and suggested only a few corrections. I invite, therefore, the authors to address the points raised by the Reviewer to make the manuscript suitable for publication.

Reviewers' comments:

Reviewer's Responses to Questions

**Comments to the Author**

1. Is the manuscript technically sound, and do the data support the conclusions?

Reviewer #1: Yes

2. Has the statistical analysis been performed appropriately and rigorously? 

Reviewer #1: Yes

3. Have the authors made all data underlying the findings in their manuscript fully available?

Reviewer #1: Yes

4. Is the manuscript presented in an intelligible fashion and written in standard English?

Reviewer #1: Yes

5. Review Comments to the Author

Reviewer #1: The study focuses on a significant problem-violence against women with disabilities. Excellent work. The paper is well-written. I have few suggestions for strengthening the manuscript.

A general definition of disability or how authors defined disability early on in the paper could be included. On page 13 studies were excluded if they focused only on common mental health disorders—rationale could be provided as to why these were excluded.

The authors cover three bodies of literature and define two of these-disability focused research and intersection of disability and violence on page 10, could also elaborate more on the third-“measurement of disability in the context of research focused on violence against women”.

Page 17 says, 419 selected for full text review and an additional 269 were excluded after that. The count is 150 after exclusion and not 174. The sentence “A final 174 reports or manuscripts met the inclusion criteria” could be deleted from that para—that para is focusing only on peer reviewed literature. The next para could then be rephrased for clarity and the final count can be mentioned in that para.

Discussion/conclusion—The authors could elaborate more on how the gap in literature is impacting practice and policies for disabled women with exposures to violence and how addressing this gap is important for practitioners and policy makes

6. PLOS authors have the option to publish the peer review history of their article (what does this mean?). If published, this will include your full peer review and any attached files.

Reviewer #1: No

---

## [Author Response · Author response to Decision Letter 0]

22 Dec 2021

Authors’ response to reviewers

Manuscript title: A scoping review of measurement of violence against women and disability

To the Editors, PLoS One

Thank you for the recognition of the contribution of our manuscript, “A scoping review of measurement of violence against women and disability.” In response to the reviewer’s comments, some changes have been made to the manuscript. We appreciate the reviewer’s positive comments and feel that our responses to these helpful suggestions helped improve this manuscript. The reviewers’ comments, as well as journal requirements listed, are addressed point-by-point in turn below.

Editorial comments:

1. Please ensure that your manuscript meets PLOS ONE's style requirements

We have formatted the manuscript according to these style requirements.

2. Grant information: We have removed the funding information from the manuscript. The funding information that is now in the online submission form reads as follows: 

“This work was funded under the UNWomen-WHO Joint Programme on Violence against Women Data, funded by the Foreign, Commonwealth and Development Office. The funders had no role in study design, data collection and analysis, decision to publish, or preparation of the manuscript.”

Reviewer comments: 

1. A general definition of disability or how authors defined disability early on in the paper could be included. 

We have added the following definition on Page 7: 

Disability is defined as “the interaction between individuals with a health condition…with personal and environmental factors including negative attitudes, inaccessible transportation and public buildings, and limited social support,” [18]; the Convention on the Rights of Persons with Disabilities also emphasizes social participation, such that “disability results from the interaction between persons with impairments and attitudinal and environmental barriers that hinders their full and effective participation in society on an equal basis with others” [19]. 

2. On page 13 studies were excluded if they focused only on common mental health disorders—rationale could be provided as to why these were excluded.

We have added the following rationale. 

Common mental disorders were excluded as there is a robust evidence-base on VAW and common mental disorders. This evidence-base includes several systematic reviews and meta-analyses [33-35], and therefore we focused this review on an area with less well-developed measurement and methodology.

3. The authors cover three bodies of literature and define two of these-disability focused research and intersection of disability and violence on page 10, could also elaborate more on the third-“measurement of disability in the context of research focused on violence against women”.

We have added the following definition on Page 11: “We define measurement of disability in VAW research as research that focuses on questions of prevalence of violence that measure disability as a specific risk factor or variable within study objectives focusing on understanding VAW in a population or specific group.” 

4. Page 17 says, 419 selected for full text review and an additional 269 were excluded after that. The count is 150 after exclusion and not 174. The sentence “A final 174 reports or manuscripts met the inclusion criteria” could be deleted from that para—that para is focusing only on peer reviewed literature. The next para could then be rephrased for clarity and the final count can be mentioned in that para.

We have edited as suggested, and the second paragraph now reads as follows: 

The grey literature search was conducted separately, and identified 316 reports, of which 5 were selected for inclusion. In addition, 16 Demographic and Health Surveys and 3 reports of national violence against women studies met the inclusion criteria. With the 150 peer-reviewed manuscripts, a final 174 reports or manuscripts met the inclusion criteria.

5. Discussion/conclusion—The authors could elaborate more on how the gap in literature is impacting practice and policies for disabled women with exposures to violence and how addressing this gap is important for practitioners and policy makes

We have added the following text to the Discussion section: “Our findings indicate limitations in measurement and evidence-base, which impact policy-makers and programmers given the overall prevalence of violence against women with disabilities is unknown, which limits development and implementation of effective policies and support services. In addition, the lack of evidence concerning how different types of disabilities operate as risk factors compared to others, and can create different barriers and enables for women who experience violence seeking support, hampers effective programming tailored to specific needs.”

---

## [Decision Letter · Decision Letter 1]

11 Jan 2022

A scoping review of measurement of violence against women and disability

PONE-D-21-26391R1

Dear Dr. Meyer,

We’re pleased to inform you that your manuscript has been judged scientifically suitable for publication and will be formally accepted for publication once it meets all outstanding technical requirements.

Kind regards,

Stefano Federici, Ph.D.

Academic Editor

PLOS ONE

Additional Editor Comments (optional):

Reviewers' comments:

Reviewer's Responses to Questions

**Comments to the Author**

1. If the authors have adequately addressed your comments raised in a previous round of review and you feel that this manuscript is now acceptable for publication, you may indicate that here to bypass the “Comments to the Author” section, enter your conflict of interest statement in the “Confidential to Editor” section, and submit your "Accept" recommendation.

Reviewer #1: All comments have been addressed

2. Is the manuscript technically sound, and do the data support the conclusions?

Reviewer #1: Yes

3. Has the statistical analysis been performed appropriately and rigorously? 

Reviewer #1: Yes

4. Have the authors made all data underlying the findings in their manuscript fully available?

Reviewer #1: Yes

5. Is the manuscript presented in an intelligible fashion and written in standard English?

Reviewer #1: Yes

6. Review Comments to the Author

Reviewer #1: I do not have additional comments. The authors addressed my suggested changes. The only minor edit is in the text added in the discussion--do you mean barriers and enablers? Enables seem like an error

7. PLOS authors have the option to publish the peer review history of their article (what does this mean?). If published, this will include your full peer review and any attached files.

Reviewer #1: No

---

## [Editor Report · Acceptance letter]

18 Jan 2022

PONE-D-21-26391R1 

A scoping review of measurement of violence against women and disability 

Dear Dr. Meyer:

I'm pleased to inform you that your manuscript has been deemed suitable for publication in PLOS ONE. Congratulations! Your manuscript is now with our production department. 

Kind regards, 

on behalf of

Prof. Stefano Federici 

Academic Editor

PLOS ONE